# How to Boost Any Loss Function

**Richard Nock**
Google Research
richardnock@google.com

**Yishay Mansour**
Tel Aviv University
Google Research
mansour@google.com

## Abstract

Boosting is a highly successful ML-born optimization setting in which one is required to computationally efficiently learn arbitrarily good models based on the access to a weak learner oracle, providing classifiers performing at least slightly differently from random guessing. A key difference with gradient-based optimization is that boosting's original model does not requires access to first order information about a loss, yet the decades long history of boosting has quickly evolved it into a first order optimization setting – sometimes even wrongfully *defining* it as such. Owing to recent progress extending gradient-based optimization to use only a loss' zeroth ($0^{th}$) order information to learn, this begs the question: what loss functions can be efficiently optimized with boosting and what is the information really needed for boosting to meet the *original* boosting blueprint's requirements?

We provide a constructive formal answer essentially showing that *any* loss function can be optimized with boosting and thus boosting can achieve a feat not yet known to be possible in the classical $0^{th}$ order setting, since loss functions are not required to be be convex, nor differentiable or Lipschitz – and in fact not required to be continuous either. Some tools we use are rooted in quantum calculus, the mathematical field – not to be confounded with quantum computation – that studies calculus without passing to the limit, and thus without using first order information.

## 1   Introduction

In ML, zeroth order optimization has been devised as an alternative to techniques that would otherwise require access to $\geqslant 1$-order information about the loss to minimize, such as gradient descent (stochastic or not, constrained or not, etc., see Section 2). Such approaches replace the access to a so-called *oracle* providing derivatives for the loss at hand, operations that can be consuming or not available in exact form in the ML world, by the access to a cheaper function value oracle, providing loss values at queried points.

Zeroth order optimization has seen a considerable boost in ML over the past years, over many settings and algorithms, yet, there is one foundational ML setting and related algorithms that, to our knowledge, have not yet been the subject of investigations: boosting [32, 31]. Such a question is very relevant: boosting has quickly evolved as a technique requiring first-order information about the loss optimized [6, Section 10.3], [41, Section 7.2.2] [53]. It is also not uncommon to find boosting reduced to this first-order setting [9]. However, originally, the boosting model did not mandate the access to any first-order information about the loss, rather requiring access to a weak learner providing classifiers at least slightly different from random guessing [31]. In the context of zeroth-order optimization gaining traction in ML, it becomes crucial to understand not just whether differentiability is necessary for boosting, but more generally what are loss functions that can be boosted with a weak learner and *in fine* where boosting stands with respect to recent formal progress on lifting gradient descent to zeroth-order optimisation.

38th Conference on Neural Information Processing Systems (NeurIPS 2024).

**In this paper**, we settle the question: we design a formal boosting algorithm for any loss function whose set of discontinuities has zero Lebesgue measure. With traditional floating point encoding (e.g. float64), any stored loss function would *de facto* meet this condition; mathematically speaking, we encompass losses that are not necessarily convex, nor differentiable or Lipschitz. This is a key difference with classical zeroth-order optimization results where the algorithms are zeroth-order *but* their proof of convergence makes various assumptions about the loss at hand, such as convexity, differentiability (once or twice), Lipschitzness, etc. . Our proof technique builds on a simple boosting technique for convex functions that relies on an order-one Taylor expansion to bound the progress between iterations [45]. Using tools from quantum calculus$^*$, we replace this progress using $v$-derivatives and a quantity related to a generalisation of the Bregman information [7]. The boosting rate involves the classical weak learning assumption's advantage over random guessing and a new parameter bounding the ratio of the expected weights (squared) over a generalized notion of curvature involving $v$-derivatives. Our algorithm, which learns a linear model, introduces notable generalisations compared to the AdaBoost / gradient boosting lineages, chief among which the computation of acceptable *offsets* for the $v$-derivatives used to compute boosting weights, offsets being zero for classical gradient boosting. To preserve readability and save space, all proofs and additional information are postponed to an Appendix.

## 2 Related work

Over the past years, ML has seen a substantial push to get the cheapest optimisation routines, in general batch [14], online [27], distributed [3], adversarial [20, 18] or bandits settings [2] or more specific settings like projection-free [26, 28, 51] or saddle-point optimisation [25, 38]. We summarize several dozen recent references in Table 1 in terms of assumptions for the analysis about the loss optimized, provided in Appendix, Section A. *Zeroth-order* optimization reduces the information available to the learner to the "cheapest" one which consists in (loss) function values, usually via a so-called function value *oracle*. However, as Table 1 shows, the loss itself is always assumed to have some form of "niceness" to study the algorithms' convergence, such as differentiability, Lipschitzness, convexity, etc. . Another quite remarkable phenomenon is that throughout all their diverse settings and frameworks, not a single one of them addresses boosting. Boosting is however a natural candidate for such investigations, for two reasons. First, the most widely used boosting algorithms are first-order information hungry [6, 41, 53]: they require access to derivatives to compute examples' weights and classifiers' leveraging coefficients. Second and perhaps most importantly, unlike other optimization techniques like gradient descent, the original boosting model *does not* mandate the access to a first-order information oracle to learn, but rather to a weak learning oracle which supplies classifiers performing slightly differently from random guessing [32, 31]. Only few approaches exist to get to "cheaper" algorithms relying on less assumptions about the loss at hand, and to our knowledge do not have boosting-compliant convergence proofs, as for example when alleviating convexity [16, 46] or access to gradients of the loss [54]. Such questions are however important given the early negative results on boosting convex potentials with first-order information [37] and the role of the classifiers in the negative results [39].

Finally, we note that a rich literature has developed in mathematics as well for derivative-free optimisation [34], yet methods would also often rely on assumptions included in the three above (*e.g.* [42]). It must be noted however that derivative-free optimisation has been implemented in computers for more than seven decades [24].

## 3 Definitions and notations

The following shorthands are used: $[n] \doteq \{1, 2, ..., n\}$ for $n \in \mathbb{N}_*$, $z \cdot [a, b] \doteq [\min\{za, zb\}, \max\{za, zb\}]$ for $z \in \mathbb{R}, a \leqslant b \in \mathbb{R}$. In the batch supervised learning setting, one is given a training set of $m$ examples $S \doteq \{(\boldsymbol{x}_i, y_i), i \in [m]\}$, where $\boldsymbol{x}_i \in \mathcal{X}$ is an observation ($\mathcal{X}$ is called the domain: often, $\mathcal{X} \subseteq \mathbb{R}^d$) and $y_i \in \mathcal{Y} \doteq \{-1, 1\}$ is a label, or class. We study the empirical convergence of boosting, which requires fast convergence on training. Such a setting is standard in zeroth order optimization [42]. Also, investigating generalization would entail specific design choices about the loss at hand and thus would restrict the scope of our result (see *e.g.* [8]). The objective is to

---

$^*$Calculus "without limits" [30] (thus without using derivatives), not to be confounded with calculus on quantum devices.

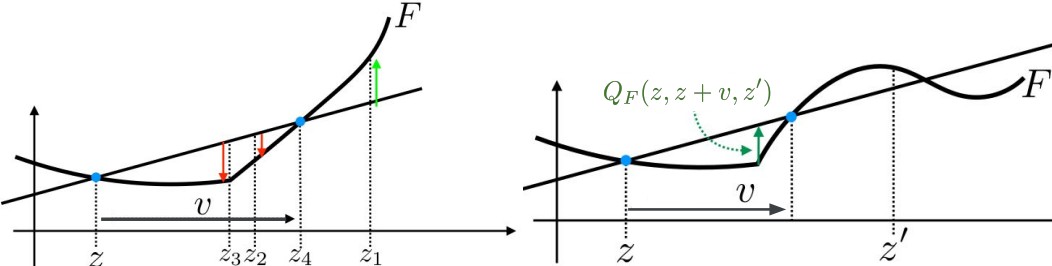

Figure 1: *Left*: value of $S_{F|v}(z'\|z)$ for convex $F$, $v \doteq z_4 - z$ and various $z'$ (colors), for which the Bregman Secant distortion is positive ($z' = z_1$, green), negative ($z' = z_2$, red), minimal ($z' = z_3$) or null ($z' = z_4, z$). *Right*: depiction of $Q_F(z, z + v, z')$ for non-convex $F$ (Definition 4.6).

learn a *classifier*, *i.e.* a function $h : \mathcal{X} \to \mathbb{R}$ which belongs to a given set $\mathcal{H}$. The goodness of fit of some $h$ on $S$ is evaluated from a given function $F : \mathbb{R} \to \mathbb{R}$ called a loss function, whose expectation on training is sought to be minimized:

$$F(S, h) \doteq \mathsf{E}_{i \sim [m]}[F(y_i h(\boldsymbol{x}_i))].$$

The set of most popular losses comprises convex functions: the exponential loss ($F_{\text{EXP}}(z) \doteq \exp(-z)$), the logistic loss ($F^{\text{LOG}}(z) \doteq \log(1 + \exp(-z))$), the square loss ($F_{\text{SQ}}(z) \doteq (1 - z)^2$), the Hinge loss ($F_{\text{H}}(z) \doteq \max\{0, 1 - z\}$). These are surrogate losses because they all define upperbounds of the 0/1-loss ($F_{0/1}(z) \doteq 1_{z \leqslant 0}$, "1" being the indicator variable).

Our ML setting is that of boosting [31]. It consists in having primary access to a weak learner WL that when called, provides so-called weak hypotheses, weak because barely anything is assumed in terms of classification performance relatively to the sample over which they were trained. Our goal is to devise a so-called "boosting" algorithm that can take any loss $F$ *as input* and training sample $S$ and a target loss value $F_*$ and after some $T$ calls to the weak learner crafts a classifier $H_T$ satisfying $F(S, H_T) \leqslant F_*$, where $T$ depends on various parameters of the ML problem. Our boosting architecture is a linear model: $H_T \doteq \sum_t \alpha_t h_t$ where each $h_t$ is an output from the weak learner and leveraging coefficients $\alpha_t$ have to be computed during boosting. Notice that this is substantially more general than the classical boosting formulation where the loss would be fixed or belong to a restricted subset of functions.

## 4 $v$-derivatives and Bregman secant distortions

Unless otherwise stated, in this Section, $F$ is a function defined over $\mathbb{R}$.

**Definition 4.1.** *[30] For any $z, v \in \mathbb{R}$, we let $\delta_v F(z) \doteq (F(z+v) - F(z))/v$ denote the $v$-derivative of $F$ in $z$.*

This expression, which gives the classical derivative when the *offset* $v \to 0$, is called the *h-derivative* in quantum calculus [30, Chapter 1]. We replaced the notation for the risk of confusion with classifiers. Notice that the $v$-derivative is just the slope of the secant that passes through points $(z, F(z))$ and $(z + v, F(z + v))$ (Figure 1). Higher order $v$-derivatives can be defined with the same offset used several times [30]. Here, we shall need a more general definition that accommodates for variable offsets.

**Definition 4.2.** *Let $v_1, v_2, ..., v_n \in \mathbb{R}$ and $\mathcal{V} \doteq \{v_1, v_2, ..., v_n\}$ and $z \in \mathbb{R}$. The $\mathcal{V}$-derivative $\delta_\mathcal{V} F$ is:*

$$\delta_\mathcal{V} F(z) \doteq \begin{cases} F(z) & \text{if} \quad \mathcal{V} = \varnothing \\ \delta_{v_1} F(z) & \text{if} \quad \mathcal{V} = \{v_1\} \\ \delta_{\{v_n\}}(\delta_{\mathcal{V} \setminus \{v_n\}} F)(z) & \text{otherwise} \end{cases}.$$

*If $v_i = v, \forall i \in [n]$ then we write $\delta_v^{(n)} F(z) \doteq \delta_\mathcal{V} F(z)$.*

In the Appendix, Lemma B.1 computes the unravelled expression of $\delta_\mathcal{V} F(z)$, showing that the order of the elements in $\mathcal{V}$ does not matter; $n$ is called the order of the $\mathcal{V}$-derivative.

We can now define a generalization of Bregman divergences called *Bregman Secant distortions*.

**Definition 4.3.** *For any $z, z', v \in \mathbb{R}$, the Bregman Secant distortion $S_{F|v}(z'\|z)$ with generator $F$ and offset $v$ is:*

$$S_{F|v}(z'\|z) \doteq F(z') - F(z) - (z' - z)\delta_v F(z).$$

Even if $F$ is convex, the distortion is not necessarily positive, though it is lowerbounded (Figure 1). There is an intimate relationship between the Bregman Secant distortions and Bregman divergences. We shall use a definition slightly more general than the original one when $F$ is differentiable [11, eq. (1.4)], introduced in information geometry [5, Section 3.4] and recently reintroduced in ML [10].

**Definition 4.4.** *The Bregman divergence with generator $F$ (scalar, convex) between $z'$ and $z$ is $D_F(z'\|z) \doteq F(z') + F^\star(z) - z'z$, where $F^\star(z) \doteq \sup_t tz - F(t)$ is the convex conjugate of $F$.*

We state the link between $S_{F|v}$ and $D_F$ (proof omitted).

**Lemma 4.5.** *Suppose $F$ strictly convex differentiable. Then $\lim_{v \to 0} S_{F|v}(z'\|z) = D_F(z'\|F'(z))$.*

Relaxed forms of Bregman divergences have been introduced in information geometry [43].

**Definition 4.6.** *For any $a, b, \alpha \in \mathbb{R}$, denote for short $\mathbb{I}_{a,b} \doteq [\min\{a, b\}, \max\{a, b\}]$ and $(uv)_\alpha \doteq \alpha u + (1 - \alpha)v$. The Optimal Bregman Information (OBI) of $F$ defined by triple $(a, b, c) \in \mathbb{R}^3$ is:*

$$Q_F(a, b, c) \doteq \max_{\alpha:(ab)_\alpha \in \mathbb{I}_{a,c}} \{(F(a)F(b))_\alpha - F((ab)_\alpha)\}. \tag{1}$$

As represented in Figure 1 (right), the OBI is obtained by drawing the line passing through $(a, F(a))$ and $(b, F(b))$ and then, in the interval $\mathbb{I}_{a,c}$, look for the maximal difference between the line and $F$. We note that $Q_F$ is non negative because $a \in \mathbb{I}_{a,c}$ and for the choice $\alpha = 1$, the RHS in (1) is 0. We also note that when $F$ is convex, the RHS is indeed the maximal Bregman information of two points in [7, Definition 2], where maximality is obtained over the probability measure. The following Lemma follows from the definition of the Bregman secant divergence and the OBI. An inspection of the functions in Figure 1 provides a graphical proof.

**Lemma 4.7.** *For any $F$,*

$$\forall z, v, z' \in \mathbb{R}, S_{F|v}(z'\|z) \geqslant -Q_F(z, z + v, z'). \tag{2}$$

*and if $F$ is convex,*

$$\forall z, v \in \mathbb{R}, \forall z' \notin \mathbb{I}_{z,z+v}, S_{F|v}(z'\|z) \geqslant 0,$$
$$\forall z, v, z' \in \mathbb{R}, S_{F|v}(z'\|z) \geqslant -Q_F(z, z + v, z + v). \tag{3}$$

We shall abbreviate the two possible forms of OBI in the RHS of (2), (3) as:

$$Q_F^*(z, z', v) \doteq \begin{cases} Q_F(z, z + v, z + v) & \text{if } F \text{ convex} \\ Q_F(z, z + v, z') & \text{otherwise} \end{cases}. \tag{4}$$

## 5 Boosting using only queries on the loss

We make the assumption that predictions of so-called "weak classifiers" are finite and non-zero on training without loss of generality (otherwise a simple tweak ensures it without breaking the weak learning framework, see Appendix, Section B.2). Excluding 0 ensures our algorithm does not make use of derivatives.

**Assumption 5.1.** *$\forall t > 0, \forall i \in [m], |h_t(\boldsymbol{x}_i)| \in (0, +\infty)$ (we thus let $M_t \doteq \max_i |h_t(\boldsymbol{x}_i)|$).*

For short, we define two *edge* quantities for $i \in [m]$ and $t = 1, 2, ...$,

$$e_{ti} \doteq \alpha_t \cdot y_i h_t(\boldsymbol{x}_i), \quad \tilde{e}_{ti} \doteq y_i H_t(\boldsymbol{x}_i), \tag{5}$$

where $\alpha_t$ is a leveraging coefficient for the weak classifiers in an ensemble $H_T(.) \doteq \sum_{t \in [T]} \alpha_t h_t(.)$. We observe

$$\tilde{e}_{ti} = \tilde{e}_{(t-1)i} + e_{ti}.$$

**Algorithm 1** SECBOOST$(S, T)$ // red boxes pinpoint substantial differences with "classical" boosting

**Input** sample $S = \{(\boldsymbol{x}_i, y_i), i = 1, 2, ..., m\}$, number of iterations $T$, initial $(h_0, v_0)$ (constant classification and offset).

Step 1 : let $H_0 \leftarrow 1 \cdot h_0$ and $\boxed{\boldsymbol{w}_1 = -\delta_{v_0} F(h_0) \cdot \mathbf{1}}$ ; $\qquad$ // $h_0, v_0 \neq 0$ chosen s. t. $\delta_{v_0} F(h_0) \neq 0$

Step 2 : **for** $t = 1, 2, ..., T$

$\qquad$ Step 2.1 : let $h_t \leftarrow \text{WL}(S_t, |\boldsymbol{w}_t|)$ $\qquad$ //weak learner call, $\boxed{S_t \doteq \{(\boldsymbol{x}_i, y_i \cdot \text{sign}(w_{ti}))\}}$

$\qquad$ Step 2.2 : let $\eta_t \leftarrow (1/m) \cdot \sum_i w_{ti} y_i h_t(\boldsymbol{x}_i)$ $\qquad\qquad\qquad\qquad$ //unnormalized edge

$\qquad$ Step 2.3 :

| If bound on $\overline{W}_{2,t}$ available (Section 5.3) | otherwise \| general procedure |
|---|---|
| pick $\varepsilon_t > 0, \pi_t \in (0, 1)$ and $\alpha_t \in \dfrac{\eta_t}{2(1+\varepsilon_t)M_t^2\overline{W}_{2,t}} \cdot [1 - \pi_t, 1 + \pi_t]$; $\quad$ (6) | $\alpha_t \leftarrow \text{SOLVE}_\alpha(S, \boldsymbol{w}_t, h_t)$ // $\overline{W}_{2,t} > 0, \varepsilon_t > 0, \pi_t \in (0, 1)$ // Theorem 5.8 |

$\qquad$ Step 2.4 : let $H_t \leftarrow H_{t-1} + \alpha_t \cdot h_t$ $\qquad\qquad\qquad\qquad\qquad\qquad$ //classifier update

$\qquad$ $\boxed{\text{Step 2.5}}$ : **if** $\mathbb{I}_{ti}(\varepsilon_t \cdot \alpha_t^2 M_t^2 \overline{W}_{2,t}) \neq \varnothing, \forall i \in [m]$ **then** $\qquad\qquad$ //new offsets

$\qquad\qquad\qquad$ **for** $i = 1, 2, ..., m$, let

$$v_{ti} \leftarrow \text{OO}(t, i, \varepsilon_t \cdot \alpha_t^2 M_t^2 \overline{W}_{2,t}) \ ;$$

$\qquad\qquad\qquad$ **else return** $H_t$;

$\qquad$ Step 2.6 : **for** $i = 1, 2, ..., m$, let $\qquad\qquad\qquad\qquad\qquad\qquad$ //weight update

$$\boxed{w_{(t+1)i} \leftarrow -\delta_{v_{ti}} F(y_i H_t(\boldsymbol{x}_i))} \ ; \tag{7}$$

$\qquad$ $\boxed{\text{Step 2.7}}$ : **if** $\boldsymbol{w}_{t+1} = \mathbf{0}$ **then** break;

**Return** $H_T$.

---

## 5.1 Algorithm: SECBOOST

### 5.1.1 General steps

Without further ado, Algorithm SECBOOST presents our approach to boosting without using derivatives information. The key differences with traditional boosting algorithms are red color framed. We summarize its key steps.

**Step 1** This is the initialization step. Traditionally in boosting, one would pick $h_0 = 0$. Note that $\boldsymbol{w}_1$ is not necessarily positive. $v_0$ is the initial offset (Section 4).

**Step 2.1** This step calls the weak learner, as in traditional boosting, using variable "weights" on examples (the coordinate-wise absolute value of $\boldsymbol{w}_t$, denoted $|\boldsymbol{w}_t|$). The key difference with traditional boosting is that examples labels can switch between iterations as well, which explains that the training sample, $S_t$, is indexed by the iteration number.

**Step 2.3** This step computes the leveraging coefficient $\alpha_t$ of the weak classifier $h_t$. It involves a quantity, $\overline{W}_{2,t}$, which we define as any strictly positive real satisfying

$$\mathbb{E}_{i \sim [m]}\left[\delta_{\{e_{ti}, v_{(t-1)i}\}} F(\tilde{e}_{(t-1)i}) \cdot \left(\frac{h_t(\boldsymbol{x}_i)}{M_t}\right)^2\right] \leqslant \overline{W}_{2,t}. \tag{8}$$

For boosting rate's sake, we should find $\overline{W}_{2,t}$ as small as possible. We refer to (5) for the $e_{\cdot}, \tilde{e}_{\cdot}$ notations; $v_{\cdot}$ is the current (set of) offset(s) (Section 4 for their definition). The second-order $\mathcal{V}$-derivative in the LHS plays the same role as the second-order derivative in classical boosting rates, see for example [45, Appendix, eq. 29]. As offsets $\to 0$, it converges to a second-order derivative; otherwise, they still share some properties, such as the sign for convex functions.

**Lemma 5.2.** *Suppose $F$ convex. For any $a \in \mathbb{R}, b, c \in \mathbb{R}_*$, $\delta_{\{b,c\}} F(a) > 0$.*

(Proof in Appendix, Section B.3) We can also see a link with weights variation since, modulo a slight abuse of notation, we have $\delta_{\{e_{ti}, v_{(t-1)i}\}} F(\tilde{e}_{(t-1)i}) = \delta_{e_{ti}} w_{ti}$. A substantial difference with traditional boosting algorithms is that we have two ways to pick the leveraging coefficient $\alpha_t$; the first one can be used when a convenient $\overline{W}_{2,t}$ is directly accessible from the loss. Otherwise, there is a simple algorithm that provides parameters (including $\overline{W}_{2,t}$) such that (8) is satisfied. Section 5.3

details those two possibilities and their implementation. In the more favorable case (the former one), $\alpha_t$ can be chosen in an interval, furthermore defined by flexible parameters $\varepsilon_t > 0, \pi_t \in (0, 1)$. Note that fixing beforehand these parameters is not mandatory: we can also pick *any*

$$\alpha_t \in \eta_t \cdot \left(0, \frac{1}{M_t^2 \overline{W}_{2,t}}\right), \tag{9}$$

and then compute choices for the corresponding $\varepsilon_t$ and $\pi_t$. $\varepsilon_t$ is important for the algorithm and both parameters are important for the analysis of the boosting rate. From the boosting standpoint, a smaller $\varepsilon_t$ yields a larger $\alpha_t$ and a smaller $\pi_t$ reduces the interval of values in which we can pick $\alpha_t$; both cases tend to favor better convergence rates as seen in Theorem 5.3.

**Step 2.4** is just the crafting of the final model.

**Step 2.5** is new to boosting, the use of a so-called offset oracle, detailed in Section 5.1.2.

**Step 2.6** The weight update does not rely on a first-order oracle as in traditional boosting, but uses only loss values through $v$-derivatives. The finiteness of $F$ implies the finiteness of weights.

**Step 2.7** Early stopping happens if all weights are null. While this would never happen with traditional (*e.g.* strictly convex) losses, some losses that are unusual in the context of boosting can lead to early stopping. A discussion on early stopping and how to avoid it is in Section 6.

### 5.1.2  The offset oracle, OO

Let us introduce notation

$$\mathbb{I}_{ti}(z) \doteq \left\{v : Q_F^*(\tilde{e}_{ti}, \tilde{e}_{(t-1)i}, v) \leqslant z\right\}, \forall i \in [m], \forall z > 0. \tag{10}$$

(see Figure 3 below to visualize $\mathbb{I}_{ti}(z)$ for a non-convex $F$) The offset oracle is used in Step 2.5, which is new to boosting. It requests the offsets to carry out weight update in (7) to an *offset oracle*, which achieves the following, for iteration #$t$, example #$i$, limit OBI $z$:

$$\mathrm{OO}(t, i, z) \text{ returns some } v \in \mathbb{I}_{ti}(z) \tag{11}$$

Note that the offset oracle has the freedom to pick the offset in a whole set. Section 5.4 investigates implementations of the offset oracle, so let us make a few essentially graphical remarks here. OO does not need to build the whole $\mathbb{I}_{ti}(z)$ to return some $v \in \mathbb{I}_{ti}(z)$ for Step 2.5 in SECBOOST. In the construction steps of Figure 3, as soon as $\mathcal{O} \neq \varnothing$, one element of $\mathcal{O}$ can be returned. Figure 4 presents more examples of $\mathbb{I}_{ti}(z)$. One can remark that the sign of the offset $v_{ti}$ in Step 2.5 of SECBOOST is the same as the sign of $\tilde{e}_{(t-1)i} - \tilde{e}_{ti} = -y_i \alpha_t h_t(\boldsymbol{x}_i)$. Hence, unless $F$ is derivable or all edges $y_i h_t(\boldsymbol{x}_i)$ are of the same sign ($\forall i$), the set of offsets returned in Step 2.5 always contain at least two different offsets, one non-negative and one non-positive (Figure 4, (a-b)).

### 5.2  Convergence of SECBOOST

The offset oracle has a technical importance for boosting: $\mathbb{I}_{ti}(z)$ is the set of offsets that limit an OBI for a training example (Definition 4.6). The importance for boosting comes from Lemma 4.7: upperbounding an OBI implies lowerbounding a Bregman Secant divergence, which will also guarantee a sufficient slack between two successive boosting iterations. This is embedded in a blueprint of a proof technique to show boosting-compliant convergence which is not new, see *e.g.* [45]. We now detail this convergence.

Remark that the expected edge $\eta_t$ in Step 2.2 of SECBOOST is not normalized. We define a normalized version of this edge as:

$$[-1, 1] \ni \tilde{\eta}_t \doteq \sum_i \frac{|w_{ti}|}{W_t} \cdot \tilde{y}_{ti} \cdot \frac{h_t(\boldsymbol{x}_i)}{M_t}, \tag{12}$$

with $\tilde{y}_{ti} \doteq y_i \cdot \mathrm{sign}(w_{ti})$, $W_t \doteq \sum_i |w_{ti}| = \sum_i |\delta_{v_{(t-1)i}} F(\tilde{e}_{(t-1)i})|$. Remark that the labels are corrected by the weight sign and thus may switch between iterations. In the particular case where the loss is non-increasing (such as with traditional convex surrogates), the labels do not switch. We need also a quantity which is, in absolute value, the expected weight:

$$\overline{W}_{1,t} \doteq \left|\mathbb{E}_{i \sim [m]}\left[\delta_{v_{(t-1)i}} F(\tilde{e}_{(t-1)i})\right]\right| \quad \text{(we indeed observe } \overline{W}_{1,t} = |\mathbb{E}_{i \sim [m]}[w_{ti}]|). \tag{13}$$

In classical boosting for convex decreasing losses[†], weights are non-negative and converge to a minimum (typically 0) as examples get the right class with increasing confidence. Thus, $\overline{W}_{1,t}$ can be an indicator of when classification becomes "good enough" to stop boosting. In our more general setting, it shall be used in a similar indicator. We are now in a position to show a first result about SECBOOST.

**Theorem 5.3.** *Suppose assumption 5.1 holds. Let $F_0 \doteq F(S, h_0)$ in SECBOOST and $z^*$ any real such that $F(z^*) \leqslant F_0$. Then we are guaranteed that classifier $H_T$ output by SECBOOST satisfies $F(S, H_T) \leqslant F(z^*)$ when the number of boosting iterations $T$ yields:*

$$\sum_{t=1}^{T} \frac{\overline{W}_{1,t}^2 (1 - \pi_t^2)}{\overline{W}_{2,t}(1 + \varepsilon_t)} \cdot \tilde{\eta}_t^2 \geqslant 4(F_0 - F(z^*)), \tag{14}$$

*where parameters $\varepsilon_t, \pi_t$ appear in Step 2.3 of SECBOOST.*

(proof in Appendix, Section B.4) We observe the tradeoff between the freedom in picking parameters and convergence guarantee as exposed by (14): to get more freedom in picking the leveraging coefficient $\alpha_t$, we typically need $\pi_t$ large (Step 2.3) and to get more freedom in picking the offset $v_t \neq 0$, we typically need $\varepsilon_t$ large (Step 2.5). However, allowing more freedom in such ways reduces the LHS and thus impairs the guarantee in (14). Therefore, there is a subtle balance between "freedom" of choice and convergence. This balance becomes more clear as boosting compliance formally enters convergence requirement.

**Boosting-compliant convergence** We characterize convergence in the boosting framework, which shall include the traditional weak learning assumption.

**Assumption 5.4.** *($\gamma$-**Weak Learning Assumption**, $\gamma$-WLA) We assume the following on the weak learner: $\exists \gamma > 0$ such that $\forall t > 0$, $|\tilde{\eta}_t| \geqslant \gamma$.*

As is usually the case in boosting, the weights are normalized in the weak learning assumption (12). So the minimization "potential" of the loss does not depend on the absolute scale of weight. This is not surprising because the loss is "nice" in classical boosting: a large $\gamma$ guarantees most examples' edges moving to the right of the $x$-axis after the classifier update which, because the loss is strictly decreasing (exponential loss, logistic loss, etc.), is sufficient to yield a smaller expected loss. In our case it is not true anymore as for example there could be a local bump in the loss that would have it increase after the update. This is not even a pathological example: one may imagine that instead of a single bump the loss jiggles a lot locally. How can we keep boosting operating in such cases ? A sufficient condition takes the form of a second assumption that also integrates weights, ensuring that the *variation* of weights is locally not too large compared to (unnormalized) weights, which is akin to comparing local first- and second-order variations of the loss in the differentiable case. We encapsulate this notion in what we call a weight regularity assumption.

**Assumption 5.5.** *($\rho$-**Weight Regularity Assumption**, $\rho$-WRA) Let $\rho_t \doteq \overline{W}_{1,t}^2 / \overline{W}_{2,t}$. We assume there exists $\rho > 0$ such that $\forall t \geqslant 1$, $\rho_t > \rho$.*

In Figure 2 we present a(n overly) simplified depiction of the cases where $\overline{W}_{2,t}$ is large for "not nice" losses, and two workarounds on how to keep it small enough for the WRA to hold. Keep in mind that $\overline{W}_{1,t}$ is an expected local variation of the loss (13), (5), so as it goes to zero, boosting converges to a local minimum and it is reasonable to expect that the WRA breaks. Otherwise, there are two strategies that keep $\overline{W}_{2,t}$ relatively small enough for WRA to hold: either we pick small enough offsets, which essentially works for most losses but make us converge in general to a local minimum (this is in essence our experimental choice) *or* we optimize the offset oracle so that it sometimes "passes" local jiggling (Figure 2 (d)). While this eventually requires to tune the weak learner jointly with the offset oracle and fine-tune that latter algorithm on a loss-dependent basis, such a strategy can be used to eventually pass local minima of the loss. To do so, "larger" offsets directly translate into corresponding requests for larger magnitude classification for the next weak classifier, for the related examples. We are now in a position to state a simple corollary to Theorem 5.3.

---

[†]This is an important class of losses since it encompasses the convex surrogates of symmetric proper losses [44, 49]

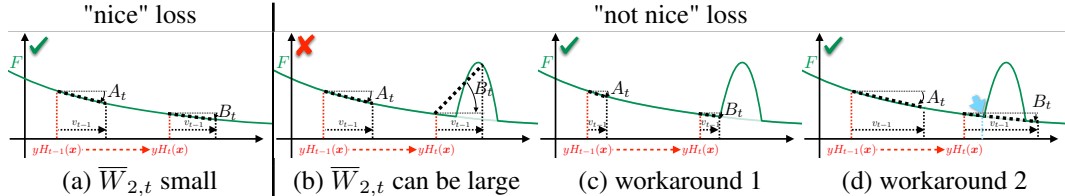

| (a) $\overline{W}_{2,t}$ small | (b) $\overline{W}_{2,t}$ can be large | (c) workaround 1 | (d) workaround 2 |

Figure 2: Simplified depiction of $\overline{W}_{2,t}$ "regimes" (Assumption 5.5). We only plot the components of the $v$-derivative part in (8): removing index $i$ for readability, we get $\delta_{\{e_t, v_{t-1}\}} F(\tilde{e}_{t-1}) = (B_t - A_t)/(yH_t(\boldsymbol{x}) - yH_{t-1}(\boldsymbol{x}))$ with $A_t \doteq \delta_{v_{t-1}} F(yH_{t-1}(\boldsymbol{x})) = -w_t$ and $B_t \doteq \delta_{v_{t-1}} F(yH_t(\boldsymbol{x}))$ ($= -w_{t+1}$ iff $v_{t-1} = v_t$). If the loss is "nice" like the exponential or logistic losses, we always have a small $\overline{W}_{2,t}$ (a). Place a bump in the loss (b-d) and the risk happens that $\overline{W}_{2,t}$ is too large for the WRA to hold. Workarounds include two strategies: picking small enough offsets (b) or fit offsets large enough to pass the bump (c). The blue arrow in (d) is discussed in Section 6.

---

**Algorithm 2** $\text{SOLVE}_\alpha(S, \boldsymbol{w}, h)$

---

**Input** sample $S = \{(\boldsymbol{x}_i, y_i), i = 1, 2, ..., m\}$, $\boldsymbol{w} \in \mathbb{R}^m$, $h : \mathcal{X} \to \mathbb{R}$.
Step 1 : find any $a > 0$ such that

$$\frac{|\eta(\boldsymbol{w}, h) - \eta(\tilde{\boldsymbol{w}}(\text{sign}(\eta(\boldsymbol{w}, h)) \cdot a), h)|}{|\eta(\boldsymbol{w}, h)|} < 1. \tag{16}$$

**Return** $\text{sign}(\eta(\boldsymbol{w}, h)) \cdot a$.

---

**Corollary 5.6.** *Suppose assumptions 5.1, 5.5 and 5.4 hold. Let $F_0 \doteq F(S, h_0)$ in SECBOOST and $z$ any real such that $F(z) \leqslant F_0$. If SECBOOST is run for a number $T$ of iterations satisfying*

$$T \geqslant \frac{4(F_0 - F(z))}{\gamma^2 \rho} \cdot \frac{1 + \max_{t \in [T]} \varepsilon_t}{1 - \max_{t \in [T]} \pi_t^2}, \tag{15}$$

*then $F(S, H_T) \leqslant F(z)$.*

We remark that the dependency in $\gamma$ is optimal [4].

## 5.3 Finding $\overline{W}_{2,t}$

There is lots of freedom in the choice of $\alpha_t$ in Step 2.3 of SECBOOST, and even more if we look at (9). This, however, requires access to some bound $\overline{W}_{2,t}$. In the general case, the quantity it upperbounds in (8) also depends on $\alpha_t$ because $e_{ti} \doteq \alpha_t \cdot y_i h_t(\boldsymbol{x}_i)$. So unless we can obtain such a "simple" $\overline{W}_{2,t}$ that does *not* depend on $\alpha_t$, (6) – and (9) – provide a *system* to solve for $\alpha_t$.

$\overline{W}_{2,t}$ **via properties of** $F$ Classical assumptions on loss functions for zeroth-order optimization can provide simple expressions for $\overline{W}_{2,t}$ (Table 1). Consider smoothness: we say that $F$ is $\beta$-smooth if it is derivable and its derivative satisfies the Lipschitz condition $|F'(z') - F'(z)| \leqslant \beta|z' - z|, \forall z, z'$ [12]. Notice that this implies the condition on the $v$-derivative of the derivative: $|\delta_v F'(z)| \leqslant \beta, \forall z, v$. This also provides a straightforward useful expression for $\overline{W}_{2,t}$.

**Lemma 5.7.** *Suppose that the loss $F$ is $\beta$-smooth. Then we can fix $\overline{W}_{2,t} = 2\beta$.*

(Proof in Appendix, Section B.5) What the Lemma shows is that a bound on the $v$-derivative of the derivative implies a bound on order-2 $\mathcal{V}$-derivatives (in the quantity that $\overline{W}_{2,t}$ bounds (8)). Such a condition on $v$-derivatives is thus weaker than a condition on derivatives, and it is strictly weaker if we impose a strictly positive lowerbound on the offset's absolute value, which would be sufficient to characterize the boosting convergence of SECBOOST.

**A general algorithm for** $\overline{W}_{2,t}$ If we cannot make any assumption on $F$, there is a simple way to *first* obtain $\alpha_t$ and then $\overline{W}_{2,t}$, from which all other parameters of Step 2.3 can be computed. We first need a few definitions. We first generalize the edge notation appearing in Step 2.2:

$$\eta(\boldsymbol{w}, h) \doteq \mathbb{E}_{i \sim [m]} [w_i y_i h(\boldsymbol{x}_i)],$$

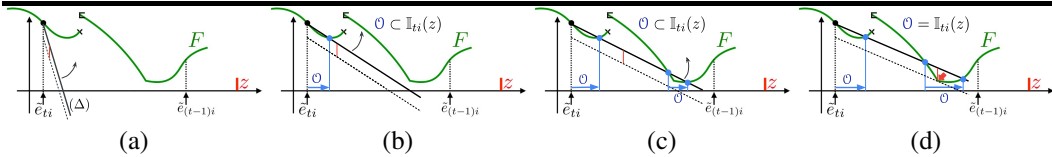

(a)          (b)          (c)          (d)

Figure 3: A simple way to build $\mathbb{I}_{ti}(z)$ for a discontinuous loss $F$ ($\tilde{e}_{ti} < \tilde{e}_{(t-1)i}$ and $z$ are represented), $\mathcal{O}$ being the set of solutions as it is built. We rotate two half-lines, one passing through $(\tilde{e}_{ti}, F(\tilde{e}_{ti}))$ (thick line, $(\Delta)$) and a parallel one translated by $-z$ (dashed line) (a). As soon as $(\Delta)$ crosses $F$ on any point $(z', F(z'))$ with $z \neq \tilde{e}_{ti}$ while the dashed line stays below $F$, we obtain a candidate offset $v$ for OO, namely $v = z' - \tilde{e}_{ti}$. In (b), we obtain an interval of values. We keep on rotating $(\Delta)$, eventually making appear several intervals for the choice of $v$ if $F$ is not convex (c). Finally, when we reach an angle such that the maximal difference between $(\Delta)$ and $F$ in $[\tilde{e}_{ti}, \tilde{e}_{(t-1)i}]$ is $z$ ($z$ can be located at an intersection between $F$ and the dashed line), we stop and obtain the full $\mathbb{I}_{ti}(z)$ (d).

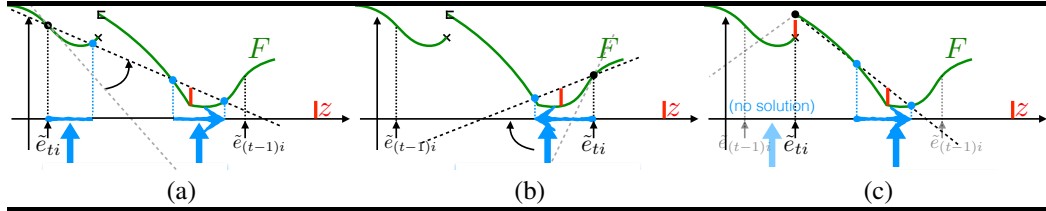

(a)          (b)          (c)

Figure 4: More examples of ensembles $\mathbb{I}_{ti}(z)$ (in blue) for the $F$ in Figure 3. (a): $\mathbb{I}_{ti}(z)$ is the union of two intervals with all candidate offsets non negative. (b): it is a single interval with non-positive offsets. (c): at a discontinuity, if $z$ is smaller than the discontinuity, we have no direct solution for $\mathbb{I}_{ti}(z)$ for at least one positioning of the edges, but a simple trick bypasses the difficulty (see text).

so that $\eta_t \doteq \eta(\boldsymbol{w}_t, h_t)$. Remind the weight update, $w_{ti} \doteq -\delta_{v_{(t-1)i}} F(y_i H_{t-1}(\boldsymbol{x}_i))$. We define a "partial" weight update,

$$\tilde{w}_{ti}(\alpha) \doteq -\delta_{v_{(t-1)i}} F(\alpha y_i h_t(\boldsymbol{x}_i) + y_i H_{t-1}(\boldsymbol{x}_i)) \tag{17}$$

(if we were to replace $v_{(t-1)i}$ by $v_{ti}$ and let $\alpha \doteq \alpha_t$, then $\tilde{w}_{ti}(\alpha)$ would be $w_{(t+1)i}$, hence the partial weight update). Algorithm 2 presents the simple procedure to find $\alpha_t$. Notice that we use $\tilde{\boldsymbol{w}}$ with sole dependency on the prospective leveraging coefficient; we omit for clarity the dependences in the current ensemble ($H_.$), weak classifier ($h_.$) and offsets ($v_{.i}$) needed to compute (17).

**Theorem 5.8.** *Suppose Assumptions 5.1 and 5.4 hold and $F$ is continuous at all abscissae* $\{\tilde{e}_{(t-1)i} \doteq y_i H_{t-1}(\boldsymbol{x}_i), i \in [m]\}$. *Then there are always solutions to Step 1 of* SOLVE$_\alpha$ *and if we let $\alpha_t \leftarrow$ SOLVE$_\alpha(S, \boldsymbol{w}_t, h_t)$ and then compute*

$$\overline{W}_{2,t} \doteq \left| \mathbb{E}_{i \sim [m]} \left[ \frac{h_t^2(\boldsymbol{x}_i)}{M_t^2} \cdot \delta_{\{\alpha_t y_i h_t(\boldsymbol{x})_i\}, v_{(t-1)i}\}} F(\tilde{e}_{(t-1)i}) \right] \right|,$$

*then $\overline{W}_{2,t}$ satisfies (8) and $\alpha_t$ satisfies (6) for some $\varepsilon_t > 0, \pi_t \in (0, 1)$.*

The proof, in Section B.6, proceeds by reducing condition (9) to (16). The Weak Learning Assumption (5.4) is important for the denominator in the LHS of (16) to be non zero. The continuity assumption *at all abscissae* is important to have $\lim_{a \to 0} \eta(\tilde{\boldsymbol{w}}_t(a), h_t) = \eta_t$, which ensures the existence of solutions to (16), also easy to find, *e.g.* by a simple dichotomic search starting from an initial guess for $a$. Note the necessity of being continuous only at abscissae defined by the training sample, which is finite in size. Hence, if this condition is not satisfied but discontinuities of $F$ are of Lebesgue measure 0, it is easy to add an infinitesimal constant to the current weak classifier, ensuring the conditions of Theorem 5.8 and keeping the boosting rates.

### 5.4 Implementation of the offset oracle

Figure 3 explains how to build graphically $\mathbb{I}_{ti}(z)$ for a general $F$. While it is not hard to implement a general procedure following the blueprint (*i.e.* accepting the loss function as input), it would be far

from achieving computational optimality: a much better choice consists in specializing it to the (set of) loss(es) at hand via hardcoding specific optimization features of the desired loss(es). This would not prevent "loss oddities" to get absolutely trivial oracles (see Appendix, Section B.7).

# 6 Discussion

For an efficient implementation, boosting requires specific design choices to make sure the weak learning assumption stands for as long as necessary; experimentally, it is thus a good idea to adapt the weak learner to build more complex models as iterations increase (*e.g.* learning deeper trees), keeping Assumption 5.4 valid with its advantage over random guessing parameter $\gamma > 0$. In our more general setting, our algorithm SECBOOST pinpoints two more locations that can make use of specific design choices to keep assumptions stand for a larger number of iterations.

The first is related to handling local minima. When Assumption 5.5 breaks, it means we are close to a local optimum of the loss. One possible way of escaping those local minima is to adapt the offset oracle to output larger offsets (Step 2.5) that get weights computed outside the domain of the local minimum. Such offsets can be used to inform the weak learner of the specific examples that then need to receive larger magnitude in classification, something we have already discussed in Section 5. There is also more: the sign of the weight indicates the polarity of the next edge ($e_{t\cdot}$, (5)) needed to decrease the loss *in the interval spanned by the last offset*. To simplify, suppose a substantial fraction of examples have an edge $\tilde{e}_{t\cdot}$ in the vicinity of the blue dotted line in Figure 2 (d) so that the loss value is indicated by the big arrow and suppose their current offset $= v_{t-1}$ so that their weight (positive) signals that to minimize further the loss, the weak learner's next weak classifier has to have a positive edge over these examples. Such is the polarity constraint which essentially comes to satisfy the WLA, but there is a magnitude constraint that comes from the WRA: indeed, if the positive edge is too small so that the loss ends up in the "bump" region, then there is a risk that the WRA breaks because the loss around the bump is quite flat, so the numerator of $\rho_t$ in Assumption 5.5 can be small. Passing the bump implies escaping the local minimum at which the loss would otherwise be trapped. Section 5.4 has presented a general blueprint for the offset oracle but more specific implementation designs can be used; some are discussed in the Appendix, Section B.7.

The second is related to handling losses that take on constant values over parts of their domain. To prevent early stopping in Step 2.7 of SECBOOST, one needs $\boldsymbol{w}_{t+1} \neq \boldsymbol{0}$. The update rule of $\boldsymbol{w}_t$ imposes that the loss must then have non-zero *variation* for some examples between two successive edges (5). If the loss $F$ is constant, then clearly the algorithm obviously stops without learning anything. If $F$ is piecewise-constant, this constrain the design of the weak learner to make sure that some examples receive a different loss with the new model update $H_\cdot$. As explained in Appendix, Section B.11, this can be efficiently addressed by specific designs on SOLVE$_\alpha$.

In the same way as there is no "1 size fits all" weak learner for all domains in traditional boosting, we expect specific design choices to be instrumental in better handling specific losses in our more general setting. Our theory points two locations further work can focus on.

# 7 Conclusion

Boosting has rapidly moved to an optimization setting involving first-order information about the loss optimized, rejoining, in terms of information needed, that of the hugely popular (stochastic) gradient descent. But this was not a formal requirement of the initial setting and in this paper, we show that essentially any loss function can be boosted without this requirement. From this standpoint, our results put boosting in a slightly more favorable light than recent development on zeroth-order optimization since, to get boosting-compliant convergence, we do not need the loss to meet any of the assumptions that those analyses usually rely on. Of course, recent advances in zeroth-order optimization have also achieved substantial design tricks for the implementation of such algorithms, something that undoubtedly needs to be adressed in our case, such as for the efficient optimization of the offset oracle. We leave this as an open problem but provide in Appendix some toy *experiments* that a straightforward implementation achieves, hinting that SECBOOST can indeed optimize very "exotic" losses.

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

# Appendix

To differentiate with the numberings in the main file, the numbering of Theorems, etc. is letter-based (A, B, ...).

## Table of contents

| reference | F | | | | | $\nabla F$ | main |
| --- | --- | --- | --- | --- | --- | --- | --- |
| | conv. | diff. | Lip. | smooth | Lb | diff. | ML topic |
| [2] | ✓ | ✓ | ✓ | ✓ | | | online ML |
| [3] | ✓ | | ✓ | | | | distributed ML |
| [1] | ✓ | | ✓ | | | | online ML |
| [14] | ✓ | ✓ | | ✓ | | ✓ | alt. GD |
| [15] | | ✓ | | ✓ | | ✓ | alt. GD |
| [18] | | ✓ | ✓ | | | | alt. GD |
| [17] | | ✓ | | | ✓ | | alt. GD |
| [19] | ✓ | ✓ | ✓ | ✓ | | | alt. GD |
| [20] | ✓ | ✓ | | ✓ | | | alt. GD |
| [25] | | ✓ | ✓ | ✓ | | | saddle pt opt |
| [26] | | ✓ | | ✓ | | | alt. FW |
| [28] | | ✓ | | ✓ | | | alt. FW |
| [22] | ✓ | ✓ | | | | | alt. GD |
| [27] | | ✓ | | | | | online ML |
| [29] | | ✓ | | ✓ | | | deep ML |
| [35] | ✓ | ✓ | | ✓ | | | alt. GD |
| [36] | | | ✓ | | | | saddle pt opt |
| [38] | ✓ | ✓ | ✓ | ✓ | | | saddle pt opt |
| [40] | | ✓ | | ✓ | | ✓ | distributed ML |
| [48] | | ✓ | | ✓ | | | alt. GD |
| [47] | | ✓ | ✓ | ✓ | | | federated ML |
| [50] | | ✓ | ✓ | ✓ | | ✓ | saddle pt opt |
| [51] | | ✓ | | ✓ | | | alt. FW |
| [52] | | ✓ | ✓ | ✓ | | | alt. GD |
| [55] | | ✓ | ✓ | ✓ | | ✓ | saddle pt opt |
| [56] | | ✓ | ✓ | ✓ | | ✓ | saddle pt opt |

Table 1: Summary of formal assumptions about loss $F$ used to prove algorithms' convergence in recent papers on zeroth order optimization, in different ML settings (see text for details). We use "smoothness" as a portmanteau for various conditions on the $\geqslant 1$ order differentiability condition of $F$. "conv." = convex, "diff." = differentiable, "Lip." = Lipschitz, "Lb" = lower-bounded, "alt. GD" = general alternative to gradient descent (stochastic or not), "alt. FW" = idem for Frank-Wolfe. Our paper relies on no such assumptions.

## A  A quick summary of recent zeroth-order optimization approaches

Table 1 summarizes a few dozens of recent approaches that can be related to zeroth-order optimization in various topics of ML. Note that no such approaches focus on boosting.

## B  Supplementary material on proofs

### B.1  Helper results

We now show that the order of the elements of $\mathcal{V}$ does not matter to compute the $\mathcal{V}$-derivative as in Definition 4.2. For any $\boldsymbol{\sigma} \in \{0,1\}^n$, we let $1_{\boldsymbol{\sigma}} \doteq \sum_i \sigma_i$.

**Lemma B.1.** *For any $z \in \mathbb{R}$, any $n \in \mathbb{N}_*$ and any $\mathcal{V} \doteq \{v_1, v_2, ..., v_n\} \subset \mathbb{R}$,*

$$\delta_{\mathcal{V}} F(z) = \frac{\sum_{\boldsymbol{\sigma} \in \{0,1\}^n} (-1)^{n-1_{\boldsymbol{\sigma}}} F(z + \sum_{i=1}^n \sigma_i v_i)}{\prod_{i=1}^n v_i}. \tag{18}$$

*Hence, $\delta_{\mathcal{V}} F$ is invariant to permutations of the elements of $\mathcal{V}$.*

*Proof.* We show the result by induction on the size of $\mathcal{V}$, first noting that

$$\delta_{\{v_1\}}F(z) = \delta_{v_1}F(z) \doteq \frac{F(z+v_1) - F(z)}{v_1} = \frac{1}{\prod_{i=1}^{1} v_i} \cdot \sum_{\sigma \in \{0,1\}} (-1)^{1-1\sigma} F(z + \sigma v_1). \quad (19)$$

We then assume that (18) holds for $\mathcal{V}_n \doteq \{v_1, v_2, ..., v_n\}$ and show the result for $\mathcal{V}_{n+1} \doteq \mathcal{V}_n \cup \{v_{n+1}\}$, writing (induction hypothesis used in the second identity):

$$\delta_{\mathcal{V}_{n+1}} F(z)$$
$$\doteq \frac{\delta_{\mathcal{V}_n} F(z + v_{n+1}) - \delta_{\mathcal{V}_n} F(z)}{v_{n+1}}$$
$$= \frac{\sum_{\boldsymbol{\sigma} \in \{0,1\}^n} (-1)^{n-1\boldsymbol{\sigma}} F(z + \sum_{i=1}^{n} \sigma_i v_i + v_{n+1}) - \sum_{\boldsymbol{\sigma} \in \{0,1\}^n} (-1)^{n-1\boldsymbol{\sigma}} F(z + \sum_{i=1}^{n} \sigma_i v_i)}{\prod_{i=1}^{n+1} v_i}$$
$$= \frac{\sum_{\boldsymbol{\sigma} \in \{0,1\}^n} (-1)^{n-1\boldsymbol{\sigma}} F(z + \sum_{i=1}^{n} \sigma_i v_i + v_{n+1}) + \sum_{\boldsymbol{\sigma} \in \{0,1\}^n} (-1)^{n-1\boldsymbol{\sigma}+1} F(z + \sum_{i=1}^{n} \sigma_i v_i)}{\prod_{i=1}^{n+1} v_i}$$
$$= \frac{\left\{ \begin{array}{l} \sum_{\boldsymbol{\sigma}' \in \{0,1\}^{n+1}: \sigma'_{n+1}=1} (-1)^{n-(1\boldsymbol{\sigma}'-1)} F(z + \sum_{i=1}^{n+1} \sigma'_i v_i) \\ + \sum_{\boldsymbol{\sigma}' \in \{0,1\}^{n+1}: \sigma'_{n+1}=0} (-1)^{n+1-1\boldsymbol{\sigma}'} F(z + \sum_{i=1}^{n+1} \sigma'_i v_i) \end{array} \right.}{v^{n+1}}$$
$$= \frac{\sum_{\boldsymbol{\sigma}' \in \{0,1\}^{n+1}} (-1)^{n+1-1\boldsymbol{\sigma}'} F(z + \sum_{i=1}^{n+1} \sigma'_i v_i)}{\prod_{i=1}^{n+1} v_i}, \quad (20)$$

as claimed. $\square$

We also have the following simple Lemma, which is a direct consequence of Lemma B.1.

**Lemma B.2.** *For all $z, \in \mathbb{R}, v, z' \in \mathbb{R}_*$, we have*

$$\delta_v F(z + z') = \delta_v F(z) + z' \cdot \delta_{\{z', v\}} F(z). \quad (21)$$

*Proof.* It comes from Lemma B.1 that $\delta_{\{z', v\}} F(z) = \delta_{\{v, z'\}} F(z) = (\delta_v F(z + z') - \delta_v F(z))/z'$ (and we reorder terms). $\square$

## B.2 Removing the $\neq 0$ part in Assumption 5.1

Because everything needs to be encoded, finiteness is not really an assumption. However, the non-zero assumption may be seen as limiting (unless we are happy to use first-order information about the loss (Section 5). There is a simple trick to remove it. Suppose $h_t$ zeroes on some training examples. The training sample being finite, there exists an open neighborhood $\mathbb{I}$ in 0 such that $h'_t \doteq h_t + \delta$ does not zero anymore on training examples, for any $\delta \in \mathbb{I}$. This changes the advantage $\gamma$ in the WLA (Definition 5.4) to some $\gamma'$ satisfying (we assume $\delta > 0$ wlog)

$$\gamma' \geqslant \frac{\gamma M_t}{M_t + \delta} - \frac{\delta}{M_t + \delta}$$
$$\geqslant \gamma - \frac{\delta}{M_t} \cdot (1 + \gamma),$$

from which it is enough to pick $\delta \leqslant \varepsilon \gamma M_t / (1 + \gamma)$ to guarantee advantage $\gamma' \geqslant (1 - \varepsilon)\gamma$. If $\varepsilon$ is a constant, this translates in a number of boosting iterations in Corollary 5.6 affected by a constant factor that we can choose as close to 1 as desired.

## B.3 Proof of Lemma 5.2

We reformulate

$$\delta_{\{b,c\}} F(a) = \frac{2}{b} \cdot \frac{1}{c} \cdot \left( \underbrace{\frac{F(a+b+c) + F(a)}{2}}_{\doteq \mu_2} - \underbrace{\frac{F(a+b) + F(a+c)}{2}}_{\doteq \mu_1} \right). \quad (22)$$

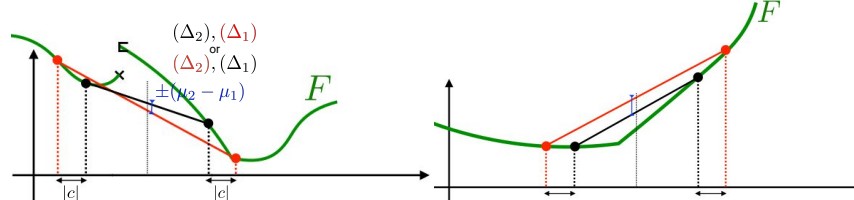

Figure 5: *Left*: representation of the difference of averages in (22). Each of the secants $(\Delta_1)$ and $(\Delta_2)$ can take either the red or black segment. Which one is which depends on the signs of $c$ and $b$, but the general configuration is always the same. Note that if $F$ is convex, one necessarily sits above the other, which is the crux of the proof of Lemma 5.2. For the sake of illustration, suppose we can analytically have $b, c \to 0$. As $c$ converges to 0 but $b$ remains $> 0$, $\delta_{\{b,c\}}F(a)$ becomes proportional to the variation of the average secant midpoint; the then-convergence of $b$ to 0 makes $\delta_{\{b,c\}}F(a)$ converge to the second-order derivative of $F$ at $a$. *Right*: in the special case where $F$ is convex, one of the secants always sits above the other.

Both $\mu_1$ and $\mu_2$ are averages that can be computed from the midpoints of two secants (respectively):

$$(\Delta_1) \doteq [(a + c, F(a + c)), (a + b, F(a + b))],$$
$$(\Delta_2) \doteq [(a, F(a)), (a + b + c, F(a + b + c))].$$

Also, the midpoints of both secants have the same abscissa (and the ordinates are $\mu_1$ and $\mu_2$), so to study the sign of $\delta_{\{b,c\}}F(a)$, we can study the position of both secants with respect to each other. $F$ being convex, we show that the abscissae of one secant are included in the abscissae of the other, this being sufficient to give the position of both secants with respect to each other. We distinguish four cases.

**Case 1**: $c > 0, b > 0$. We have $a + b + c > \max\{a + b, a + c\}$ and $a < \min\{a + b, a + c\}$. $F$ being convex, $(\Delta_2)$ sits above $(\Delta_1)$. So, $\mu_2 \geqslant \mu_1$ and finally $\delta_{\{b,c\}}F(a) \geqslant 0$.

**Case 2**: $c < 0, b < 0$. We now have $a + b + c < \min\{a + b, a + c\}$ while $a > \max\{a + b, a + c\}$, so $(\Delta_2)$ sits above $(\Delta_1)$. Again, $\mu_2 \geqslant \mu_1$ and finally $\delta_{\{b,c\}}F(a) \geqslant 0$.

**Case 3**: $c > 0, b < 0$. We have $a + b < a$ and $a + b < a + b + c$. Also $a + c > \max\{a + b + c, a\}$, so this time $(\Delta_2)$ sits below $(\Delta_1)$ but $cb < 0$, so $\delta_{\{b,c\}}F(a) \geqslant 0$ again.

**Case 4**: $c < 0, b > 0$. So $a + c < a < a + b$ and $a + c < a + b + c$. So $a + c < \min\{a, a + b + c\}$ and $a + b > \max\{a, a + c\}$, so $(\Delta_2)$ sits below $(\Delta_1)$. Since $cb < 0$, so $\delta_{\{b,c\}}F(a) \geqslant 0$ again.

### B.4   Proof of Theorem 5.3

Let us remind key simplified notations about edges, $\forall t \geqslant 0$:

$$\tilde{e}_{ti} \doteq y_i \cdot H_t(\boldsymbol{x}_i), \tag{23}$$
$$e_{ti} \doteq y_i \cdot \alpha_t h_t(\boldsymbol{x}_i) = \tilde{e}_{ti} - \tilde{e}_{(t-1)i}. \tag{24}$$

For short, we also let:

$$Q_{ti}^* \doteq Q_F^*(\tilde{e}_{ti}, \tilde{e}_{(t-1)i}, v_{i(t-1)}), \tag{25}$$
$$\Delta_{ti} \doteq \delta_{v_{i(t-1)}}F(\tilde{e}_{ti}) - \delta_{v_{i(t-1)}}F(\tilde{e}_{(t-1)i}), \tag{26}$$

where $Q_{\ddot{\cdot}}^*$ is defined in (4). We also split the computation of the leveraging coefficient $\alpha_t$ in SECBOOSTin two parts, the first computing a real $a_t$ as:

$$a_t \in \frac{1}{2(1 + \varepsilon_t)M_t^2 \overline{W}_{2,t}} \cdot [1 - \pi_t, 1 + \pi_t], \tag{27}$$

and then using $\alpha_t \leftarrow a_t \eta_t$. We now use Lemma 4.7 (main file) and get

$$\mathbb{E}_{i \sim [m]} \left[ S_{F | v_{ti}}(\tilde{e}_{ti} \| \tilde{e}_{(t+1)i}) \right] \geqslant -\mathbb{E}_{i \sim D} \left[ Q^*_{(t+1)i} \right], \forall t \geqslant 0. \tag{28}$$

If we reorganise (28) using the definition of $S_{F|.}(.\|.)$, we get:

$$\mathbb{E}_{i \sim [m]} \left[ F(\tilde{e}_{(t+1)i}) \right]$$

$$\leqslant \mathbb{E}_{i \sim [m]} \left[ F(\tilde{e}_{ti}) \right] - \mathbb{E}_{i \sim [m]} \left[ (\tilde{e}_{ti} - \tilde{e}_{(t+1)i}) \cdot \delta_{v_{ti}} F(\tilde{e}_{(t+1)i}) \right] + \mathbb{E}_{i \sim [m]} \left[ Q^*_{(t+1)i} \right]$$

$$= \mathbb{E}_{i \sim [m]} \left[ F(\tilde{e}_{ti}) \right] - \mathbb{E}_{i \sim [m]} \left[ -e_{(t+1)i} \cdot \delta_{v_{ti}} F(\tilde{e}_{(t+1)i}) \right] + \mathbb{E}_{i \sim [m]} \left[ Q^*_{(t+1)i} \right] \tag{29}$$

$$= \mathbb{E}_{i \sim [m]} \left[ F(\tilde{e}_{ti}) \right] + \alpha_{t+1} \cdot \mathbb{E}_{i \sim [m]} \left[ y_i h_{t+1}(\boldsymbol{x}_i) \cdot \delta_{v_{ti}} F(\tilde{e}_{(t+1)i}) \right] + \mathbb{E}_{i \sim [m]} \left[ Q^*_{(t+1)i} \right] \tag{30}$$

$$= \mathbb{E}_{i \sim [m]} \left[ F(\tilde{e}_{ti}) \right] + a_{t+1} \eta_{t+1} \cdot \mathbb{E}_{i \sim [m]} \left[ y_i h_{t+1}(\boldsymbol{x}_i) \cdot \delta_{v_{ti}} F(\tilde{e}_{ti}) \right]$$

$$+ a_{t+1} \eta_{t+1} \cdot \mathbb{E}_{i \sim [m]} \left[ y_i h_{t+1}(\boldsymbol{x}_i) \cdot \Delta_{(t+1)i} \right] + \mathbb{E}_{i \sim [m]} \left[ Q^*_{(t+1)i} \right] \tag{31}$$

$$= \mathbb{E}_{i \sim [m]} \left[ F(\tilde{e}_{ti}) \right] - a_{t+1} \eta_{t+1} \cdot \underbrace{\mathbb{E}_{i \sim [m]} \left[ w_{(t+1)i} y_i h_{t+1}(\boldsymbol{x}_i) \right]}_{= \eta_{t+1}} + a_{t+1} \eta_{t+1} \cdot \mathbb{E}_{i \sim [m]} \left[ y_i h_{t+1}(\boldsymbol{x}_i) \cdot \Delta_{(t+1)i} \right]$$

$$+ \mathbb{E}_{i \sim [m]} \left[ Q^*_{(t+1)i} \right]$$

$$= \mathbb{E}_{i \sim [m]} \left[ F(\tilde{e}_{ti}) \right] - a_{t+1} \eta_{t+1}^2 + a_{t+1} \eta_{t+1} \cdot \mathbb{E}_{i \sim [m]} \left[ y_i h_{t+1}(\boldsymbol{x}_i) \cdot \Delta_{(t+1)i} \right] + \mathbb{E}_{i \sim [m]} \left[ Q^*_{(t+1)i} \right]. \tag{32}$$

(29) – (31) make use of definitions (24) (twice) and (26) as well as the decomposition of the leveraging coefficient in (27).

Looking at (32), we see that we can have a boosting-compliant decrease of the loss if the two quantities depending on $\Delta_{(t+1).}$ and $Q^*_{(t+1).}$ can be made small enough compared to $a_{t+1} \eta_{t+1}^2$. This is what we investigate.

**Bounding the term depending on $\Delta_{(t+1).}$** – We use Lemma B.2 with $z \doteq \tilde{e}_{ti}, z' \doteq e_{(t+1)i}, v \doteq v_t$, which yields (also using (24) and the assumption that $h_{t+1}(\boldsymbol{x}_i) \neq 0$):

$$\Delta_{(t+1)i} \doteq \delta_{v_{ti}} F(\tilde{e}_{(t+1)i}) - \delta_{v_{ti}} F(\tilde{e}_{ti})$$

$$= \delta_{v_{ti}} F(\tilde{e}_{ti} + e_{(t+1)i}) - \delta_{v_{ti}} F(\tilde{e}_{ti})$$

$$= e_{(t+1)i} \cdot \delta_{\{e_{(t+1)i}, v_{ti}\}} F(\tilde{e}_{ti})$$

$$= y_i \cdot \alpha_{t+1} h_{t+1}(\boldsymbol{x}_i) \cdot \delta_{\{e_{(t+1)i}, v_{ti}\}} F(\tilde{e}_{ti}), \tag{33}$$

and so we get:

$$a_{t+1} \eta_{t+1} \cdot \mathbb{E}_{i \sim [m]} \left[ y_i h_{t+1}(\boldsymbol{x}_i) \cdot \Delta_{(t+1)i} \right]$$

$$= a_{t+1} \eta_{t+1} \cdot \mathbb{E}_{i \sim [m]} \left[ \alpha_{t+1} (y_i h_{t+1}(\boldsymbol{x}_i))^2 \cdot \delta_{\{e_{(t+1)i}, v_{ti}\}} F(\tilde{e}_{ti}) \right]$$

$$= a_{t+1}^2 \eta_{t+1}^2 \cdot \mathbb{E}_{i \sim [m]} \left[ (h_{t+1}(\boldsymbol{x}_i))^2 \cdot \delta_{\{e_{(t+1)i}, v_{ti}\}} F(\tilde{e}_{ti}) \right]$$

$$\leqslant a_{t+1}^2 \eta_{t+1}^2 M_{t+1}^2 \cdot \overline{W}_{2,t+1}. \tag{34}$$

**Bounding the term depending on $Q^*_{.(t+1)}$** – We immediately get from the value picked in argument of $\mathbb{I}_{t+1}$ in step 2.5 of SECBOOST, the definition of $\mathbb{I}_{ti}(.)$ in (10) and our decomposition $\alpha_t \leftarrow a_t \eta_t$ that $Q^*_{(t+1)i} \leqslant \varepsilon_{t+1} \cdot a_{t+1}^2 \eta_{t+1}^2 M_{t+1}^2 \cdot \overline{W}_{2,t+1}, \forall i \in [m]$, so that:

$$\mathbb{E}_{i \sim [m]} \left[ Q^*_{(t+1)i} \right] \leqslant \varepsilon_{t+1} \cdot a_{t+1}^2 \eta_{t+1}^2 M_{t+1}^2 \cdot \overline{W}_{2,t+1}. \tag{35}$$

**Finishing up with the proof** – Suppose that we choose $\varepsilon_{t+1} > 0$, $\pi_{t+1} \in (0, 1)$ and $a_{t+1}$ as in (27). We then get from (32), (34), (35) that for any choice of $v_{ti}$ in Step 2.5 of SECBOOST,

$$
\begin{aligned}
&\mathbb{E}_{i \sim [m]}\left[F(\tilde{e}_{(t+1)i})\right] \\
&\leqslant \mathbb{E}_{i \sim [m]}\left[F(\tilde{e}_{ti})\right] - a_{t+1}\eta_{t+1}^2 + a_{t+1}^2\eta_{t+1}^2 M_{t+1}^2 \cdot \overline{W}_{2,t+1} + \varepsilon_{t+1} \cdot a_{t+1}^2\eta_{t+1}^2 M_{t+1}^2 \cdot \overline{W}_{2,t+1} \\
&= \mathbb{E}_{i \sim [m]}\left[F(\tilde{e}_{ti})\right] - a_{t+1}\eta_{t+1}^2 \cdot \left(1 - a_{t+1}\left(1 + \varepsilon_{t+1}\right)M_{t+1}^2 \cdot \overline{W}_{2,t+1}\right) \\
&\leqslant \mathbb{E}_{i \sim [m]}\left[F(\tilde{e}_{ti})\right] - \frac{\eta_{t+1}^2(1 - \pi_{t+1}^2)}{4\left(1 + \varepsilon_{t+1}\right)M_{t+1}^2 \cdot \overline{W}_{2,t+1}},
\end{aligned}
\tag{36}
$$

where the last inequality is a consequence of (27). Suppose we pick $H_0 \doteq h_0 \in \mathbb{R}$ a constant and $v_0 > 0$ such that

$$
\delta_{v_0} F(h_0) \neq 0. \tag{37}
$$

The final classifier $H_T$ of SECBOOST satisfies:

$$
\mathbb{E}_{i \sim [m]}\left[F(y_i H_T(\boldsymbol{x}_i))\right] \leqslant F_0 - \frac{1}{4} \cdot \sum_{t=1}^{T} \frac{\eta_t^2(1 - \pi_t^2)}{(1 + \varepsilon_t)M_t^2 \overline{W}_{2,t}}, \tag{38}
$$

with $F_0 \doteq \mathbb{E}_{i \sim [m]}\left[F(\tilde{e}_{i0})\right] \doteq \mathbb{E}_{i \sim [m]}\left[F(y_i H_0)\right] = \mathbb{E}_{i \sim [m]}\left[F(y_i h_0)\right]$. If we want $\mathbb{E}_{i \sim [m]}\left[F(y_i H_T(\boldsymbol{x}_i))\right] \leqslant F(z^*)$, assuming wlog $F(z^*) \leqslant F_0$, then it suffices to iterate until:

$$
\sum_{t=1}^{T} \frac{1 - \pi_t^2}{\overline{W}_{2,t}(1 + \varepsilon_t)} \cdot \frac{\eta_t^2}{M_t^2} \geqslant 4(F_0 - F(z^*)). \tag{39}
$$

Remind that the edge $\eta_t$ is not normalized. We have defined a normalized edge,

$$
[-1, 1] \ni \tilde{\eta}_t \doteq \sum_i \frac{|w_{ti}|}{W_t} \cdot \tilde{y}_{ti} \cdot \frac{h_t(\boldsymbol{x}_i)}{M_t}, \tag{40}
$$

with $\tilde{y}_{ti} \doteq y_i \cdot \text{sign}(w_{ti})$ and $W_t \doteq \sum_i |w_{ti}| = \sum_i |\delta_{v_{(t-1)i}} F(\tilde{e}_{(t-1)i})|$. We have the simple relationship between $\eta_t$ and $\tilde{\eta}_t$:

$$
\begin{aligned}
\tilde{\eta}_t &= \sum_i \frac{|w_{ti}|}{W_t} \cdot (y_i \cdot \text{sign}(w_{ti})) \cdot \frac{h_t(\boldsymbol{x}_i)}{M_t} \\
&= \frac{1}{W_t M_t} \cdot \sum_i w_{ti} y_i h_t(\boldsymbol{x}_i) \\
&= \frac{m}{W_t M_t} \cdot \eta_t,
\end{aligned}
\tag{41}
$$

resulting in ($\forall t \geqslant 1$),

$$
\begin{aligned}
\frac{\eta_t^2}{M_t^2} &= \tilde{\eta}_t^2 \cdot \left(\frac{W_t}{m}\right)^2 \\
&= \tilde{\eta}_t^2 \cdot \left(\mathbb{E}_{i \sim [m]}\left[|\delta_{v_{(t-1)i}} F(\tilde{e}_{(t-1)i})|\right]\right)^2 \\
&\geqslant \tilde{\eta}_t^2 \cdot \left(|\mathbb{E}_{i \sim [m]}\left[\delta_{v_{(t-1)i}} F(\tilde{e}_{(t-1)i})\right]|\right)^2 \\
&= \tilde{\eta}_t^2 \cdot \overline{W}_{1,t}^2,
\end{aligned}
\tag{42}
$$

recalling $\overline{W}_{1,t} \doteq \left|\mathbb{E}_{i \sim D}\left[\delta_{v_{(t-1)i}} F(\tilde{e}_{(t-1)i})\right]\right|$. It comes from (42) that a sufficient condition for (39) to hold is:

$$
\sum_{t=1}^{T} \frac{\overline{W}_{1,t}^2(1 - \pi_t^2)}{\overline{W}_{2,t}(1 + \varepsilon_t)} \cdot \tilde{\eta}_t^2 \geqslant 4(F_0 - F(z^*)), \tag{43}
$$

which is the statement of Theorem 5.3.

## B.5 Proof of Lemma 5.7

We first observe that for any $a \in \mathbb{R}, b, c \in \mathbb{R}_*$,

$$
\begin{aligned}
|\delta_{\{b,c\}}F(a)| &= \frac{1}{|bc|} \cdot \left| \begin{array}{c} F(a+b+c) - F(a+c) - bF'(a+c) \\ -(F(a+b) - F(a) - bF'(a)) \\ +b(F'(a+c) - F'(a)) \end{array} \right| \\
&\leqslant \frac{1}{|bc|} \cdot \left( \begin{array}{c} |F(a+b+c) - F(a+c) - bF'(a+c)| \\ +|(F(a+b) - F(a) - bF'(a))| \\ +|b(F'(a+c) - F'(a))| \end{array} \right) \\
&\leqslant \frac{1}{|bc|} \cdot \left( \frac{\beta}{2} \cdot b^2 + \frac{\beta}{2} \cdot b^2 + \beta|bc| \right) = \beta + \beta \cdot \frac{b^2}{|bc|},
\end{aligned}
\tag{44}
$$

where we used the $\beta$-smoothness of $F$ and twice [12, Lemma 3.4]. We can also make a permutation in the expression of $\delta_{\{b,c\}}F(a)$ and instead write

$$
\begin{aligned}
|\delta_{\{b,c\}}F(a)| &= \frac{1}{|bc|} \cdot \left| \begin{array}{c} F(a+b+c) - F(a+b) - cF'(a+b) \\ -(F(a+c) - F(a) - cF'(a)) \\ +c(F'(a+b) - F'(a)) \end{array} \right| \\
&\leqslant \frac{1}{|bc|} \cdot \left( \begin{array}{c} |F(a+b+c) - F(a+b) - cF'(a+b)| \\ +|(F(a+c) - F(a) - cF'(a))| \\ +|c(F'(a+b) - F'(a))| \end{array} \right) \\
&\leqslant \frac{1}{|bc|} \cdot \left( \frac{\beta}{2} \cdot c^2 + \frac{\beta}{2} \cdot c^2 + \beta|bc| \right) = \beta + \beta \cdot \frac{c^2}{|bc|}.
\end{aligned}
\tag{45}
$$

We thus have

$$
|\delta_{\{b,c\}}F(a)| \leqslant \beta + \beta \cdot \left( \frac{\min\{|b|, |c|\}}{\sqrt{|bc|}} \right)^2
\tag{46}
$$
$$
\leqslant 2\beta,
$$

by the power mean inequality [13, Chapter III, Theorem 2]. Since $|h_t(\boldsymbol{x}_i)| \leqslant M_t$ by definition, we thus have

$$
\left| \mathbb{E}_{i \sim [m]} \left[ \delta_{\{e_{ti}, v_{(t-1)i}\}} F(\tilde{e}_{(t-1)i}) \cdot \left( \frac{h_t(\boldsymbol{x}_i)}{M_t} \right)^2 \right] \right| \leqslant 2\beta,
\tag{47}
$$

which allows us to fix $\overline{W}_{2,t} = 2\beta$ and completes the proof of Lemma 5.7.

**Remark B.3.** *Our result is optimal in the sense that if we make one offset (say $b$) go to zero, then the ratio in* (46) *goes to zero and we recover the condition on the $v$-derivative of the derivative, $|\delta_c F'(z)| \leqslant \beta$.*

## B.6 Proof of Theorem 5.8

We consider the upperbound::

$$
\begin{aligned}
\overline{W}_{2,t} &\\
\doteq &\left| \mathbb{E}_{i \sim [m]} \left[ \frac{h_t^2(\boldsymbol{x}_i)}{M_t^2} \cdot \delta_{\{e_{ti}, v_{(t-1)i}\}} F(\tilde{e}_{(t-1)i}) \right] \right| \\
= &\left| \mathbb{E}_{i \sim [m]} \left[ \frac{h_t^2(\boldsymbol{x}_i)}{M_t^2} \cdot \frac{1}{\tilde{e}_{ti}} \cdot \left( \frac{F(\tilde{e}_{ti} + v_{(t-1)i}) - F(\tilde{e}_{ti})}{v_{(t-1)i}} - \frac{F(\tilde{e}_{(t-1)i} + v_{(t-1)i}) - F(\tilde{e}_{(t-1)i})}{v_{(t-1)i}} \right) \right] \right| \\
= &\left| \frac{1}{\alpha_t} \cdot \mathbb{E}_{i \sim [m]} \left[ \frac{h_t(\boldsymbol{x}_i)}{y_i M_t^2} \cdot \left( \frac{F(\tilde{e}_{ti} + v_{(t-1)i}) - F(\tilde{e}_{ti})}{v_{(t-1)i}} - \frac{F(\tilde{e}_{(t-1)i} + v_{(t-1)i}) - F(\tilde{e}_{(t-1)i})}{v_{(t-1)i}} \right) \right] \right| \\
= &\left| \frac{1}{\alpha_t} \cdot \mathbb{E}_{i \sim [m]} \left[ \frac{y_i h_t(\boldsymbol{x}_i)}{M_t^2} \cdot \left( \delta_{v_{(t-1)i}} F(\tilde{e}_{ti}) - \delta_{v_{(t-1)i}} F(\tilde{e}_{(t-1)i}) \right) \right] \right|
\end{aligned}
\tag{48}
$$

(The last identity uses the fact that $y_i \in \{-1, 1\}$). Remark that we have extracted $\alpha_t$ from the denominator but it is still present in the arguments $\tilde{e}_{ti}$. For any classifier $h$, we introduce notation

$$\eta(\boldsymbol{w}, h) \doteq \mathbb{E}_{i \sim [m]} [w_i y_i h(\boldsymbol{x}_i)],$$

and so $\eta_t$ (Step 2.2 in SECBOOST) is also $\eta(\boldsymbol{w}_t, h_t)$, which is guaranteed to be non-zero by the Weak Learning Assumption (5.4). We want, for *some* $\varepsilon_t > 0, \pi_t \in [0, 1)$,

$$\alpha_t \in \frac{\eta_t}{2(1 + \varepsilon_t) M_t^2 \overline{W}_{2,t}} \cdot [1 - \pi_t, 1 + \pi_t]. \tag{49}$$

This says that the sign of $\alpha_t$ is the same as the sign of $\eta(\boldsymbol{w}_t, h_t) = \eta_t$. Since we know its sign, let us look for its absolute value:

$$|\alpha_t| \in \frac{|\eta_t|}{2(1 + \varepsilon_t) M_t^2 \overline{W}_{2,t}} \cdot [1 - \pi_t, 1 + \pi_t]. \tag{50}$$

From (9) (main file), we can in fact search $\alpha_t$ in the union of all such intervals for $\varepsilon_t > 0, \pi_t \in [0, 1)$, which amounts to find first:

$$|\alpha_t| \in \left(0, \frac{|\eta_t|}{M_t^2 \overline{W}_{2,t}}\right),$$

and then find any $\varepsilon_t > 0, \pi_t \in [0, 1)$ such that (50) holds. Using (48) and simplifying the external dependency on $\alpha_t$, we then need

$$1 \in \left(0, \frac{|\eta_t|}{\underbrace{|\mathbb{E}_{i \sim [m]} [y_i h_t(\boldsymbol{x}_i) \cdot (\delta_{v_{(t-1)i}} F(\alpha_t y_i h_t(\boldsymbol{x}_i) + \tilde{e}_{(t-1)i}) - \delta_{v_{(t-1)i}} F(\tilde{e}_{(t-1)i}))]|}_{\doteq B(\alpha_t)}}\right), \tag{51}$$

under the constraint that the sign of $\alpha_t$ be the same as that of $\eta_t$. But, using notation (17) (main file), we have

$$B(\alpha_t) = |\eta(\boldsymbol{w}_t, h_t) - \eta(\tilde{\boldsymbol{w}}_t(\alpha_t), h_t)|,$$

and so to get (51) satisfied, it is sufficient that

$$\frac{|\eta_t - \eta(\tilde{\boldsymbol{w}}_t(\alpha_t), h_t)|}{|\eta_t|} < 1, \tag{52}$$

which is Step 1 in SOLVE$_\alpha$. The Weak Learning Assumption (5.4) guarantees that the denominator is $\neq 0$ so this can always be evaluated. The continuity of $F$ in all $\tilde{e}_{(t-1)i}$ guarantees $\lim_{\alpha_t \to 0} \eta(\tilde{\boldsymbol{w}}_t(\alpha_t), h_t) = \eta_t$, and thus guarantees the existence of solutions to (52) for some $|\alpha_t| > 0$.

To summarize, finding $\alpha_t$ can be done in two steps, (i) solve

$$\frac{|\eta_t - \eta(\tilde{\boldsymbol{w}}_t(\text{sign}(\eta_t) \cdot a), h_t)|}{|\eta_t|} < 1$$

for some $a > 0$ and (ii) let $\alpha_t \doteq \text{sign}(\eta_t) \cdot a$. This is the output of SOLVE$_\alpha(S, \boldsymbol{w}_t, h_t)$, which ends the proof of Theorem 5.8.

## B.7 Implementation of the offset oracle: particular cases

Consider the "spring loss" that we define, for $[.]$ denoting the nearest integer, as:

$$F_{\text{SL}}(z) \doteq \log(1 + \exp(-z)) + 1 - \sqrt{1 - 4(z - [z])^2}. \tag{53}$$

Figure 6 plots this loss, which composes the logistic loss with a "U"-shaped term. This loss would escape all optimization algorithms of Table 1 (Appendix), yet there is a trivial implementation of our offset oracle, as explained in Figure 6:

1. if the interval $\mathbb{I}$ defined by $\tilde{e}_{(t-1)i}$ and $\tilde{e}_{ti}$ contains at least one peak, compute the tangence point $(z_t)$ at the closest local "U" that passes through $(\tilde{e}_{(t-1)i}, F(\tilde{e}_{(t-1)i}))$; then if $z_t \in \mathbb{I}$ then $v_{ti} \leftarrow z_t - \tilde{e}_{(t-1)i}$, else $v_{ti} \leftarrow \tilde{e}_{ti} - \tilde{e}_{(t-1)i}$;

2. otherwise $F$ in $\mathbb{I}$ is strictly convex and differentiable: a simple dichotomic search can retrieve a feasible $v_{ti}$ (see convex losses below);

Notice that one can alleviate the repetitive dichotomic search by pre-tabulating a feasible $v$ for a set of differences $|a - b|$ ($a, b$ belonging to the abscissae of the same "U") decreasing by a fixed factor, choosing $v_{ti} \leftarrow v$ of the largest tabulated $|a - b|$ no larger than $|\tilde{e}_{ti} - \tilde{e}_{(t-1)i}|$.

**Discontinuities** discontinuities do not represent issues if the argument $z$ of $\mathbb{I}_{ti}(z)$ is large enough, as shown from the following simple Lemma.

**Lemma B.4.** *Define the discontinuity of $F$ as:*

$$\mathrm{disc}(F) \doteq \max \left\{ \begin{array}{l} \sup_z |F(z) - \lim_{z^-} F(z)|, \\ \sup_z |F(z) - \lim_{z^+} F(z)| \end{array} \right\}. \tag{54}$$

*For any $z \geq 0$, if $\mathrm{disc}(F) \leq z$ then $\mathbb{I}_{ti}(z) \neq \varnothing, \forall t \geq 1, \forall i \in [m]$.*

Figure 4 (c) shows a case where the discontinuity is larger than $z$. In this case, an issue eventually happens for computing the next weight happens, only when the current edge is at the discontinuity. We note that as iterations increase and the weak learner finds it eventually more difficult to return weak hypotheses with $\eta$, large enough, the discontinuities may become an issue for SECBOOST to not stop at Step 2.5. Or one can always use a simple trick to avoid stopping and which relies on the leveraging coefficient $\alpha_t$: this is described in the Appendix, Section B.9.

**The case of convex losses** If $F$ is convex (not necessarily differentiable nor strictly convex), there is a simple way to find a valid output for the offset oracle, which relies on the following Lemma.

**Lemma B.5.** *Suppose $F$ convex. Then for any $z, z' \in \mathbb{R}, v \neq 0$,*

$$\{v > 0 : Q_F^*(z, z', v) = r\}$$
$$= \left\{v > 0 : D_F \left( z \,\middle\|\, \frac{F(z + v) - F(z)}{v} \right) = r \right\}. \tag{55}$$

(proof in Appendix, Section B.8) By definition, $\mathbb{I}_{ti}(z') \subseteq \mathbb{I}_{ti}(z)$ for any $z' \leq z$, so a simple way to implement the offset oracle's output $\mathrm{OO}(t, i, z)$ is, for some $0 < r < z$, to solve the Bregman identity in the RHS of (55) and then return any relevant $v$. If $F$ is strictly convex, there is just one choice.

If solving the Bregman identity is tedious but $F$ is strictly convex, there a simple dichotomic search that is guaranteed to find a feasible $v$. It exploits the fact that the abscissa maximizing the difference between any secant of $F$ and $F$ has a simple closed form (see [21, Supplement, Figure 13]) and so the OBI in (1) (Definition 4.6) has a closed form as well. In this case, it is enough, after taking a first non-zero guess for $v$ (either positive or negative), to divide it by a constant $> 1$ until the corresponding OBI is no larger than the $z$ in the query $\mathrm{OO}(t, i, z)$.

## B.8 Proof of Lemma B.5

$F$ being convex, we first want to compute the set

$$\mathbb{I}_{z,v,r} \doteq \{v > 0 : Q_F(z, z + v, z + v) = r\}, \tag{56}$$

where $r$ is supposed small enough for $\mathbb{I}_{z,v,r}$ to be non-empty. There is a simple graphical solution to this which, as Figure 7 explains, consists in finding $v$ solution of

$$\sup_t F(z + v) - \left( F(t) + \left( \frac{F(z + v) - F(z)}{v} \right) \cdot (z + v - t) \right) = r. \tag{57}$$

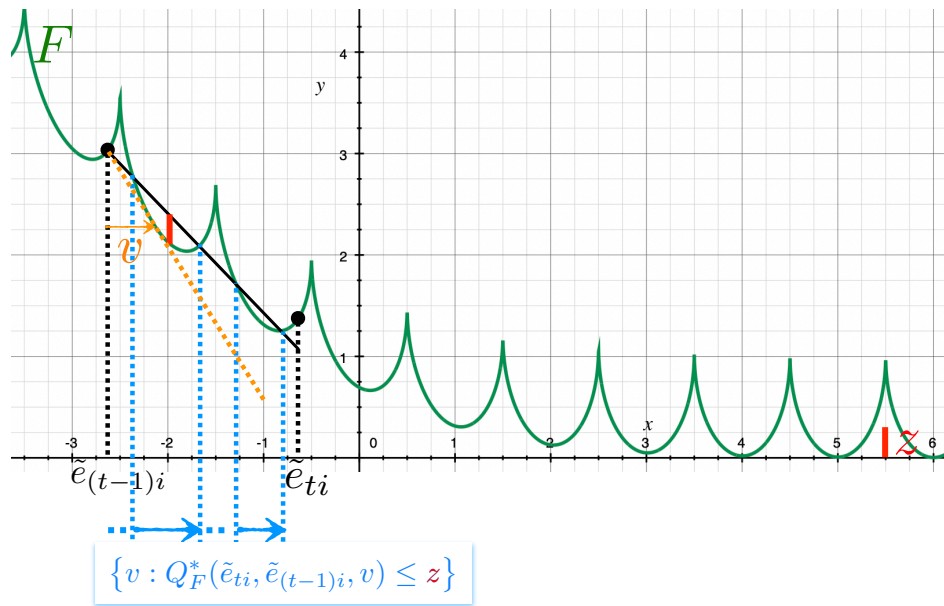

Figure 6: The spring loss in (53) is neither convex, nor Lipschitz or differentiable and has an infinite number of local minima. Yet, an implementation of the offset oracle is trivial as an output for OO can be obtained from the computation of a single tangent point (here, the orange $v$, see text; best viewed in color).

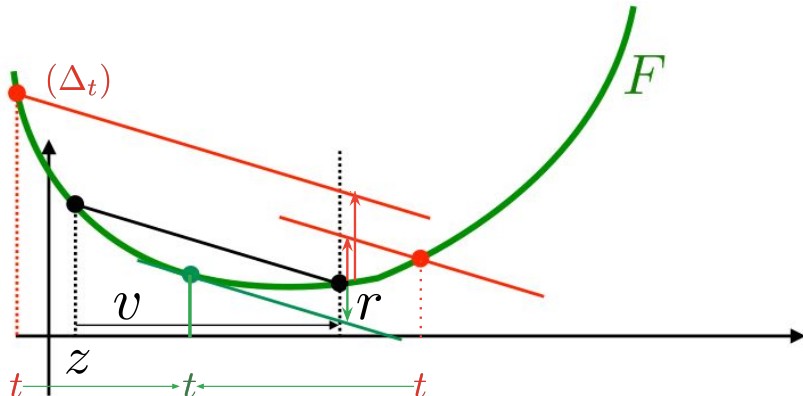

Figure 7: Computing the OBI $Q_F(z, z+v, z+v)$ for $F$ convex, $(z, v)$ being given and $v > 0$. We compute the line $(\Delta_t)$ crossing $F$ at any point $t$, with slope equal to the secant $[(z, F(z)), (z+v, F(z+v))]$ and then the difference between $F$ at $z+v$ and this line at $z+v$. We move $t$ so as to maximize this difference. The optimal $t$ (in green) gives the corresponding OBI. In (56) and 58, we are interested in finding $v$ given this difference, $r$. We also need to replicate this computation for $v < 0$.

The LHS simplifies:

$$\sup_t F(z+v) - \left( F(t) + \left( \frac{F(z+v) - F(z)}{v} \right) \cdot (z+v-t) \right)$$

$$= \frac{(z+v)F(z) - zF(z+v)}{v} + \sup_t \left\{ t \cdot \frac{F(z+v) - F(z)}{v} - F(t) \right\}$$

$$= \frac{(z+v)F(z) - zF(z+v)}{v} + F^\star \left( \frac{F(z+v) - F(z)}{v} \right)$$

$$= F(z) + F^\star \left( \frac{F(z+v) - F(z)}{v} \right) - z \cdot \frac{F(z+v) - F(z)}{v}$$

$$= D_F \left( z \left\| \frac{F(z+v) - F(z)}{v} \right. \right), \quad 23$$

so we end up with an equivalent but more readable definition for $\mathbb{I}_{z,v,r}$:

$$\mathbb{I}_{z,v,r} = \left\{ v > 0 : D_F\left( z \left\| \frac{F(z+v)-F(z)}{v} \right.\right) = r \right\}, \tag{58}$$

which yields the statement of the Lemma.

## B.9 Handling discontinuities in the offset oracle to prevent stopping in Step 2.5

Theorem 5.3 and Lemma 5.6 require to run SECBOOST for as many iterations are required. This implies not early stopping in Step 2.5. Lemma B.4 shows that early stopping can only be triggered by too large local discontinuities at the edges. This is a weak requirement on running SECBOOST, but there exists a weak assumption on the discontinuities of the loss itself that simply prevent any early stopping and does not degrade the boosting rates. The result exploits the freedom in choosing $\alpha_t$ in Step 2.3.

**Lemma B.6.** *Suppose $F$ is any function defined over $\mathbb{R}$ discontinuities of zero Lebesgue measure. Then Corollary 5.6 holds for boosting $F$ with its inequality strict while never triggering early stopping in Step 2.5 of* SECBOOST.

*Proof.* To show that we never trigger stopping in Step 2.5, it is sufficient to show that we can run SECBOOSTwhile ensuring $F$ is continuous in an open neighborhood around all edges $y_i H_t(\boldsymbol{x}_i), \forall i \in [m], \forall t \geqslant 0$ (by letting $H_0 \doteq h_0$). Remind that $\tilde{e}_{ti} \doteq \tilde{e}_{(t-1)i} + \alpha_t \cdot y_t h_t(\boldsymbol{x}_i)$, so changing $\alpha_t$ changes all edges. We just have to show that either computing $\alpha_t$ ensures such a continuity, or $\alpha_t$ can be slightly modified to do so. We have two ways to compute $\alpha_t$:

1. using a value for $\overline{W}_{2,t}$ that represents an "absolute" upperbound in the sense of (8) (*e.g.* Lemma 5.7) and then compute $\alpha_t$ as in Step 2.3 of SECBOOST;

2. using algorithm SOLVE$_\alpha$.

Because of the assumption on $F$, we can always ensure that $F$ is continuous in an open neighborhood of all edges (the basis of the induction amounts to a straightforward choice for $h_0$). This proves the Lemma for [2.].

If we rely on [1.] and the $\alpha_t$ computed leads to some discontinuities, then we have complete control to change $\alpha_t$: any continuous change of $\varepsilon_t$ induces a continuous change in $\alpha_t$ and thus a continuous change of all edges as well. So, starting from the initial $\varepsilon_t$ chosen in Step 2.3, we increase it to a value $\varepsilon_t^* > \varepsilon_t$, which we want to keep as small as possible. We can define for each $i \in [m]$ an open set $(a_i, b_i)$ which is the interval spanned by the new $\tilde{e}_{ti}(\varepsilon_t')$ using $\varepsilon_t' \in (\varepsilon_t, \varepsilon_t^*)$. Since there are only finitely many discontinuities on $F$, there exists a small $\varepsilon_t^* > \varepsilon_t$ such that

$$\forall i \in [m], \forall z \in (a_i, b_i), F \text{ is continuous on } z.$$

This means that $\forall \varepsilon_t' \in (\varepsilon_t, \varepsilon_t^*)$, we end up with a loss without any discontinuities on the new edges. Now comes the reason why we want $\varepsilon_t^* - \varepsilon_t$ small: we can check that there always exist a small enough $\varepsilon_t^* > \varepsilon_t$ such that for any $\varepsilon_t'$ we choose, the boosting rate in Corollary 5.6 is affected by at most 1 additional iteration. Indeed, while we slightly change parameter $\varepsilon_t$ to land all new edges outside of discontinuities of $F$, we *also* increase the contribution of the boosting iteration in the RHS of (15) by a quantity $\delta > 0$ which can be made as small as required — hence we can just replace the inequality in (15) by a strict inequality. This proves the statement of the Lemma if we rely on [1.] above.

This completes the proof of Lemma B.6. $\qquad\square$

## B.10 A boosting pattern that can "survive" above differentiability

Suppose $F$ is strictly convex and strictly decreasing as for classical convex surrogates (*e.g.* logistic loss). Assuming wlog all $\alpha_. > 0$ and example $i$ has both $y_i h_t(\boldsymbol{x}_i) > 0$ and $y_i h_{t-1}(\boldsymbol{x}_i) > 0$, as long as $z$ is small enough, we are guaranteed that any choice $v_{t-1} \in \mathbb{I}_{(t-1)i}(z)$ and $v_t \in \mathbb{I}_{ti}(z)$ results in $0 < w_{(t+1)i} < w_{ti}$, which follows the classical boosting pattern that examples receiving the right class by weak hypotheses have their weight decreased (See Figure 8). If $z = z'$ is large enough, then this does not hold anymore as seen from Figure 8.

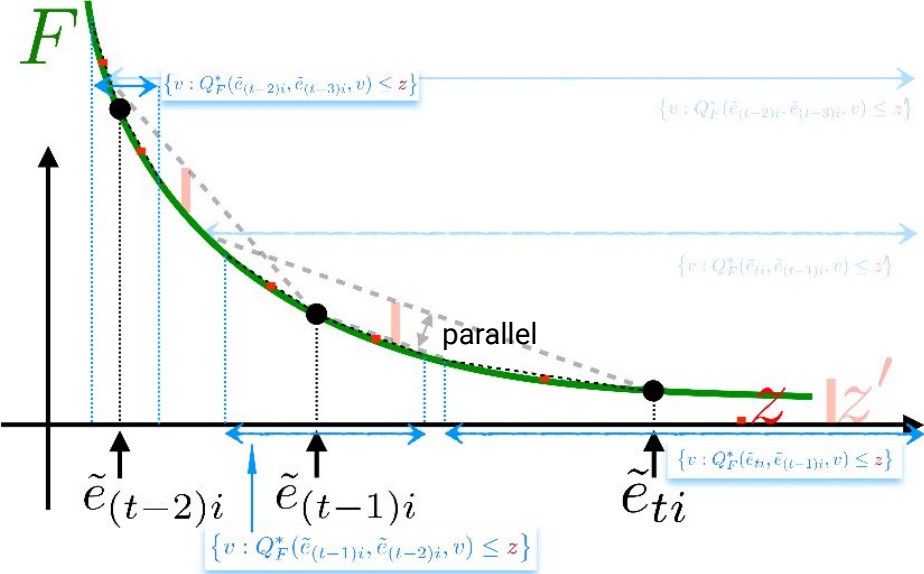

Figure 8: Case $F$ strictly convex, with two cases of limit OBI $z$ and $z'$ in $\mathbb{I}_{.i}(.)$. Example $i$ has $e_{ti} > 0$ and $e_{(t-1)i} > 0$ (**??**) large enough (hence, edges with respect to weak classifiers $h_t$ and $h_{t-1}$ large enough) so that $\mathbb{I}_{ti}(z) \cap \mathbb{I}_{(t-1)i}(z) = \mathbb{I}_{(t-1)i}(z) \cap \mathbb{I}_{(t-2)i}(z) = \mathbb{I}_{ti}(z) \cap \mathbb{I}_{(t-2)i}(z) = \varnothing$. In this case, regardless of the offsets chosen by OO, we are guaranteed that its weights satisfy $w_{(t+1)i} < w_{ti} < w_{(t-1)i}$, which follows the boosting pattern that examples receiving the right classification by weak classifiers have their weights decreasing. If however the limit OBI changes from $z$ to a larger $z'$, this is not guaranteed anymore: in this case, it may be the case that $w_{(t+1)i} > w_{ti}$.

## B.11  The case of piecewise constant losses for SOLVE$_\alpha$

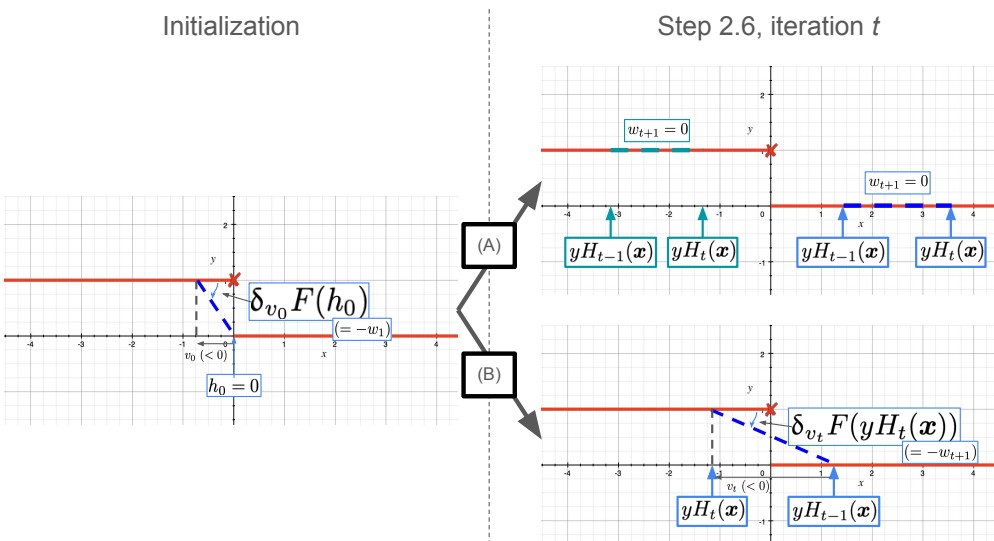

Figure 9: How our algorithm works with the 0/1 loss (in red): at the initialization stage, assuming we pick $h_0 = 0$ for simplicity and some $v_0 < 0$, all training examples get the same weight, given by negative the slope of the thick blue dashed line. All weights are thus $> 0$. At iteration $t$ when we update the weights (Step 2.6), one of two cases can happen on some training example $(\boldsymbol{x}, y)$. In **(A)**, the edge of the strong model remains the same: either both are positive (blue) or both negative (olive green) (the ordering of edges is not important). In this case, regardless of the offset, the new weight will be 0. In **(B)**, both edges have different sign (again, the ordering of edges is not important). In this case, the examples will keep non-zero weight over the next iteration. See text below for details.

Figure 9 schematizes a run of our algorithm when training loss = 0/1 loss. At the initialization, it is easy to get all examples to have non-zero weight. The weight update for example $(\boldsymbol{x}, y)$ of our algorithm in Step 2.3 is (negative) the slope of a secant that crosses the loss in two points, both being in between $yH_{t-1}(\boldsymbol{x})$ and $yH_t(\boldsymbol{x})$. Hence, if the predicted label does not change $(\mathrm{sign}(H_t(\boldsymbol{x})) = \mathrm{sign}(H_{t-1}(\boldsymbol{x})))$, then the next weight $(w_{t+1})$ of the example *will be zero* (Figure 9, case (A)). However, if the predicted label does change $(\mathrm{sign}(H_t(\boldsymbol{x})) \neq \mathrm{sign}(H_{t-1}(\boldsymbol{x})))$ then the example may get a non-zero weight depending on the offset chosen.

Hence, our generic implementation of Algorithms 3 and 4 may completely fail at providing non-zero weights for the next iteration, which makes the algorithm stop in step 2.7. And even when not all weights are zero, there may be just a too small subset of those, that would break the Weak Learning Assumption for boosting compliance of the next iteration (Assumption 5.5).

## C  Supplementary material on algorithms, implementation tricks and a toy experiment

### C.1  Algorithm and implementation of SOLVE$_\alpha$ and how to find parameters from Theorem 5.8

As Theorem 5.8 explains, SOLVE$_\alpha$ can easily get to not just the leveraging coefficient $\alpha_t$, but also other parameters that are necessary to implement SECBOOST: $\overline{W}_{2,t}$ and $\varepsilon_t$ (both used in Step 2.5). We now provide a simple pseudo code on how to implement SOLVE$_\alpha$ amnd get, on top of it, the two other parameters. We do not seek $\pi_t$ since it is useful only in the convergence analysis. Also, our proposal implementation is optimized for complexity (because of the geometric updating of $\delta, W$ in their respective loops) but much less so for for accuracy. Algorithm SOLVE_extended explains the overall procedure.

**Algorithm 3** SOLVE_extended$(S, \boldsymbol{w}, h, M)$

---

**Input** sample $S = \{(\boldsymbol{x}_i, y_i), i = 1, 2, ..., m\}$, $\boldsymbol{w} \in \mathbb{R}^m$, $h : \mathcal{X} \to \mathbb{R}$, $M \neq 0$.
    // in our case, $\boldsymbol{w} \leftarrow \boldsymbol{w}_t$; $h \leftarrow h_t$; $M \leftarrow M_t$ (current weights, weak hypothesis and max confidence, see Step 2.3 in SECBOOST and Assumption 5.1)
Step 1 :                                                                                    // all initializations

$$\eta_{\text{init}} \leftarrow \eta(\boldsymbol{w}, h); \tag{59}$$
$$\delta \leftarrow 1.0; \tag{60}$$
$$W_{\text{init}} \leftarrow 1.0; \tag{61}$$

Step 2 : **do**                                    // Step 2 computes the leveraging coefficient $\alpha_t$
    $\alpha \leftarrow \delta \cdot \text{sign}(\eta_{\text{init}})$;
    $\eta_{\text{new}} \leftarrow \eta(\tilde{\boldsymbol{w}}(\alpha), h)$;
    **if** $|\eta_{\text{new}} - \eta_{\text{init}}| < |\eta_{\text{init}}|$ **then** found_alpha $\leftarrow$ true **else** $\delta \leftarrow \delta/2$;
    **while** found_alpha = false;
Step 3 : $W \leftarrow$ Left Hand Side of (8) (main file)          // Step 3 computes $\overline{W}_{2,t}$
                                     // we can use (8) (main file) because we know $\alpha$
    **if** $W =_{\text{machine}} 0$ **then**
                   // the LHS of (8) is (machine) 0: just need to find $W$ such that (9) holds !
    $W \leftarrow W_{\text{init}}$;
    **while** $|\alpha| > |\eta_{\text{init}}|/(W \cdot M^2)$ **do** $W \leftarrow W/2$;
    **endif**
Step 4 : $b_{\text{sup}} \leftarrow |\eta_{\text{init}}|/(W \cdot M^2)$;                   // Step 4 computes $\varepsilon_t$
    $\varepsilon \leftarrow (b_{\text{sup}}/\alpha) - 1$;
**Return** $(\alpha, W, \varepsilon)$;

---

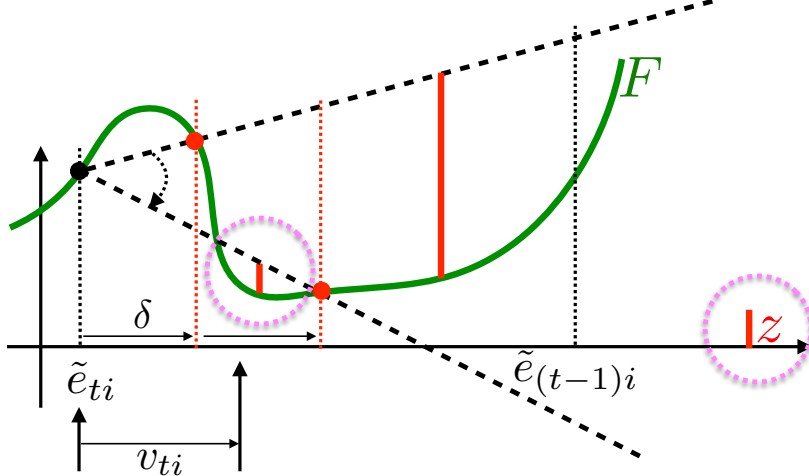

Figure 10: How to find some $v \in \mathbb{I}_{ti}(z)$: parse the interval $[\tilde{e}_{ti}, \tilde{e}_{(t-1)i}]$ with a regular step $\delta$, seek the secant with minimal slope (because $\tilde{e}_{ti} < \tilde{e}_{(t-1)i}$; otherwise, we would seek the secant with maximal slope). It is necessarily the one minimizing the OBI among all regularly spaced choices. If the OBI is still too large, decrease the step $\delta$ and start the search again.

## C.2   Algorithm and implementation of the offset oracle

There exists a very simple trick to get some adequate offset $v$ to satisfy (11) (main file), explained in Figure 10. In short, we seek the optimally bended secant and check that the OBI is no more than a required $z$. This can be done via parsing the interval $[\tilde{e}_{ti}, \tilde{e}_{(t-1)i}]$ using regularly spaced values. If the OBI is too large, we can start again with a smaller step size. Algorithm OO_simple details the key part of the search.

---

**Algorithm 4** OO_simple$(F, \tilde{e}_t, \tilde{e}_{t-1}, z, Z)$

---

**Input** loss $F$, two last edges $\tilde{e}_t, \tilde{e}_{t-1}$, maximal OBI $z$, precision $Z$.
$\qquad\qquad\quad$ // in our case, $\tilde{e}_t \leftarrow \tilde{e}_{ti}; \tilde{e}_{t-1} \leftarrow \tilde{e}_{(t-1)i}$; (for training example index $i \in [m]$)
Step 1 : $\qquad\qquad\qquad\qquad\qquad\qquad\qquad\qquad\qquad\qquad\qquad\qquad\qquad\quad$ // all initializations

$$\delta \leftarrow \frac{\tilde{e}_{t-1} - \tilde{e}_t}{Z}; \tag{62}$$

$$z_c \leftarrow \tilde{e}_t + \delta; \tag{63}$$

$$i \leftarrow 0; \tag{64}$$

Step 2 : **do**
$\qquad s_c \leftarrow \text{SLOPE}(F, \tilde{e}_t, z_c);$
$\qquad\qquad$ // returns the slope of the secant passing through $(\tilde{e}_t, F(\tilde{e}_t))$ and $(z_c, F(z_c))$
$\qquad$ **if** $(i = 0) \vee ((\delta > 0) \wedge (s_c < s_*)) \vee ((\delta < 0) \wedge (s_c > s_*))$ **then** $s_* \leftarrow s_c; z_* \leftarrow z_c$
$\qquad$ **endif**
$\qquad z_c \leftarrow z_c + \delta;$
$\qquad i \leftarrow i + 1;$
$\qquad$ **while** $(z_c - \tilde{e}_t) \cdot (z_c - \tilde{e}_{t-1}) < 0;$ $\qquad\qquad$ // checks that $z_c$ is still in the interval
**Return** $z_* - \tilde{e}_t;$ $\qquad\qquad\qquad\qquad\qquad\qquad\qquad\qquad$ // this is the offset $v$

---

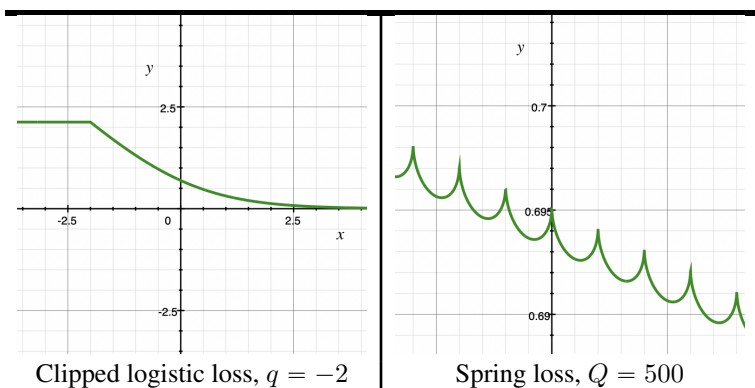

Clipped logistic loss, $q = -2$ $\qquad\qquad$ Spring loss, $Q = 500$

Figure 11: Crops of the two losses whose optimization has been experimentally tested with SEC-BOOST, in addition to the logistic loss. See text for details.

### C.3 A toy experiments

We provide here a few toy experiments using SECBOOST. These are just meant to display that a simple implementation of the algorithm, following the blueprints given above, can indeed manage to optimize various losses. These are not meant to explain how to pick the best hyperparameters (*e.g.* (60)) nor how to choose the best loss given a domain, a problem that is far beyond the scope of our paper.

In this implementation, the weak learner learns decision trees and we minimize Matushita's loss at the leaves of decision trees to learn fixed size trees, see [33] for the criterion and induction scheme, which is standard for decision trees. SECBOOST is implemented as is given in the paper, and so are the implementation of SOLVE$_\alpha$ and the offset oracle provided above. We have made no optimization whatsoever, with one exception: when numerical approximation errors lead to an offset that is machine 0, we replace it by a small random value to prevent the use of derivatives in SECBOOST.

We have investigated three losses. The first is the well known logistic loss:

$$F_{\text{LOG}}(z) \doteq \log(1 + \exp(-z)). \tag{65}$$

The other two are tweaks of the logistic loss. We have investigated a clipped version of the logistic loss,

$$F_{\text{CL},q}(z) \doteq \min\{\log(1 + \exp(-z)), \log(1 + \exp(-q))\}, \tag{66}$$

with $q \in \mathbb{R}$, which clips the logistic loss above a certain value. This loss is non-convex and non-differentiable, but it is Lipschitz. We have also investigated a generalization of the spring loss (main file):

$$F_{\text{SL},Q}(z) \doteq \log(1 + \exp(-z)) + \frac{1 - \sqrt{1 - 4(z_Q - [z_Q])^2}}{Q},$$

(67)

with $z_Q \doteq Qz - 1/2$ ([.] is the closest integer), which adds to the logistic loss regularly spaced peaks of variable width. This loss is non-convex, non-differentiable, non-Lipschitz. Figure 11 provides a crop of the clipped logistic loss and spring loss we have used in our test. Notice the "hardness" that the spring loss intuitively represents for ML.

We provide an experiment on public domain UCI `tictactoe` [23] (using a 10-fold stratified cross-validation to estimate test errors). In addition to the three losses, we have crossed them with several other variables: the size of the trees (either they have a single internal node = stumps or at most 20 nodes) and, to give one example of how changing a (key) hyperparameter can change the result, we have tested for a scale of changes on the initial value of $\delta$ in (60). Finally, we have crossed all these variables with the existence of symmetric label noise in the training data, following the setup of [37, 39]. We flip each label in the training sample with probability $\eta$. Table 12 summarizes the results obtained. One can see that SECBOOST manages to optimize all losses in pretty much all settings, with an eventual early stopping required for the spring loss if $\delta$ is too large. Note that the best initial value for $\delta$ depends on the loss optimized in these experiments: for $\delta = 0.1$, test error from the spring loss decreases much faster than for the other losses, yet we remind that the spring loss is just the logistic loss plus regularly spaced peaks. This could signal interesting avenues for the best possible implementation of SECBOOST, or a further understanding of the best formal ways to fix those paramaters, all of which are out of the scope of this paper.

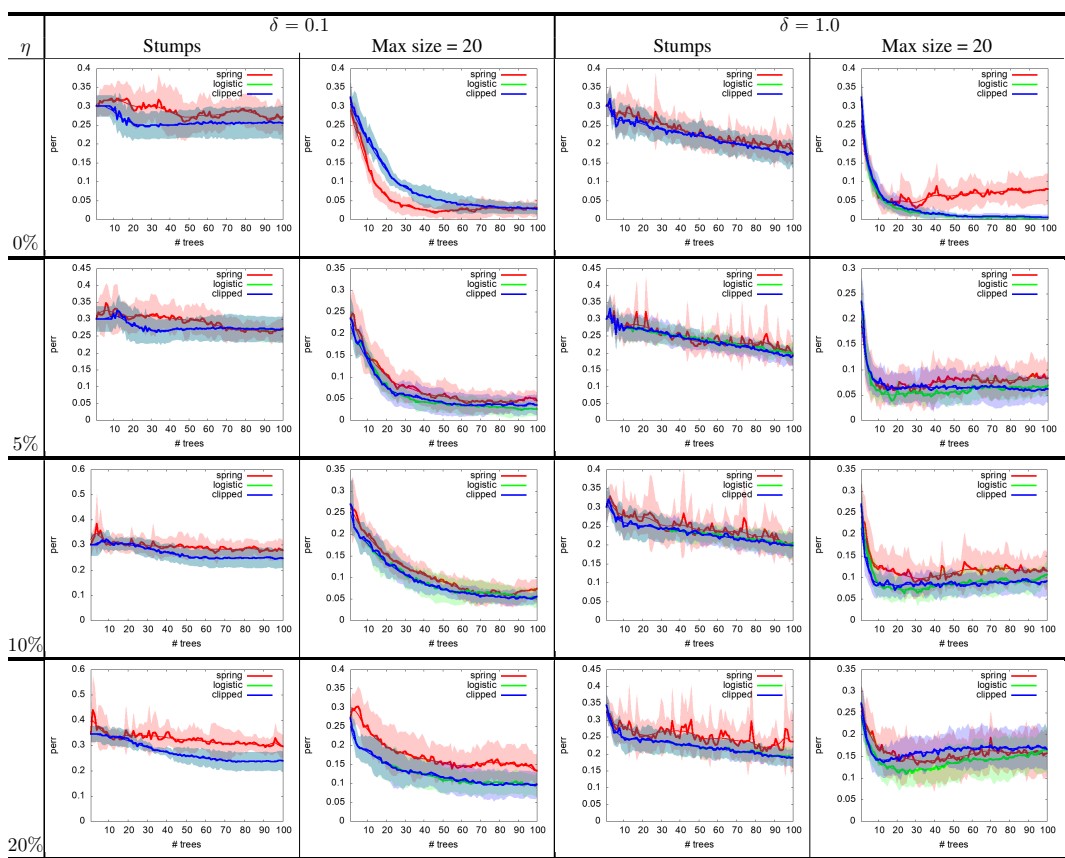

Figure 12: Experiments on UCI `tictactoe` showing estimated test errors after minimizing each of the three losses we consider, with varying training noise level $\eta$, max tree size and initial hyperparameter $\delta$ value in (60). See text.

