# OpenReview forum: "How to Boost Any Loss Function"
_NeurIPS.cc/2024/Conference — NeurIPS 2024 poster_

### Official Review · Reviewer_TGRx · 2024-06-18

**Soundness:** 2
**Presentation:** 3
**Contribution:** 2
**Rating:** 6
**Confidence:** 1

**Summary:**

The paper presents a framework for boosting any (bounded) loss function given access to a weak learner. The authors present how recent developments in boosting have involved turning the boosting problem into an optimization problem, where different combinations of assumptions on the loss such as convexity, lipschitcness, differential, etc. are assumed to be true to obtain bounds on the error. The authors then point to the origin of the boosting idea which was that one had access to a weak learner who given a weighted sample produced a hypothesis that was slightly better than guessing on the weighted sample and by using the access to query this weak learner produce a final classifier which performed arbitrarily well compared to just slightly better than guessing. I.e. there were no assumptions on the loss function itself. Thus the authors want to consider this more general setting where no extra information on the loss $F$ (other than a bound on the loss in the hypothesis returned by the weak learner on the sample is bounded). To construct boosting for such general $F$ they look to a zero'th order optimization (only access to losses) and find that they can't find any previous work on boosting using zero'th order optimization. The authors then present a new boosting algorithm SECBOOST that uses zero'th order optimization to boost any loss function. With the before mentioned boundedness assumption on $F$ they are able to show a in-sample error guarantee, which if SECBOOST is successful and run sufficiently long can be made arbitrarily close to the minimum of the loss function $\inf_{x\in R}F(x)$. Noted by the authors this guarantee holds when the number of iterations is large enough - they also comment on which problem can arrise in terms of SECBOOST stopping to early  - and how they can be alleviated.

**Strengths:**

Originality:

The authors couldn't find previous work on zero'th order optimzation and boosting so it is original in that sense. The authors also points out how SECBOOST may differ from traditional boosting algorithms by for instance, possibly chaning signs, and using emperical quantaties relating v-first and second order derivatives.

Quality:

 The main doesn't contain any proof so I will not comment on the theoretical analysis soundness and is also my reason for my confidence score. The authors seems to be honest about the limitations of SECBOOST and clearly state the assumptions made in the paper.

Clarity:
The main is very well written and the authors give an intuitive description of SECBOOST and the different concepts that it used - here also using figures to illustrate the concepts.

Significance:
 As the problem of combinding boosting and zero'th order optimization is presented as new it seems like a interesting problem to look at. The framework is presented very generally which makes it applicable in many places. Also, this new perspective on using zero'th order optimization in boosting may also inspire new ideas for boosting algorithms in problem-specific settings.

**Weaknesses:**

As also asked in the questions, to highlight the applicability of the framework it could be nice with and example of a loss $F$ where one could compare SECBOOST to the best-known algorithms for that specific loss in the boosting setting.

**Questions:**

What does $R_{\star}$ and $N_{\star}$ denote?

 Line 142 can you explain why the sign of any $v$ in $I_{ti}(z)$ is always the sign of $-y_{i}\alpha_{t}h_{t}(x_i)$?

 Figure 2, is the dotted lines $w_{t,1},\ldots,w_{t,m}$

 Corollary 5.4, i guess could also be stated with just the weak learning assumption? so with 5.1 and 5.4?

 Why does figure 3 and 4 come before figure 2? And why are the figures not on the pages where they are used to discripe the setup?

 Line 176 $\nu_{t}$ depend on $h_{t }$ thorough $M_t$ is this a problem, when we assume that the weak learner find the hypothesis $h_t$ such that $|\nu_{t}|\geq \gamma$. Why isn't it the case that the weak learner output the hypothesis $h_t$ such that $\nu_{t}\geq \gamma$(no absolute value)?

 What if the weak learning gaurantee was given in terms of the the loss funciton  $F$?

 Can you come up with a setting of $F$ where the algorithm gives a bound on the number of iterations need and compare it to the best known bound for that setting?

**Limitations:**

Yes.

---

> ### Author Rebuttal · Authors · 2024-08-03
>
> We thank the reviewer for highlighting that our " [...] main is very well written [...] " and a long strength section that indeed summarizes well some of our contributions. We hope this rebuttal answers the questions to strenghten further the review's polarity.
>
> > What does $R_*$ and $N_*$ denote ?
>
> These are the sets of reals / naturals without 0.
>
> > can you explain why the sign of any [...]
>
> because the secant is always taken in the interval defined by $\tilde{e}_{t-1}$ and ${\tilde{e}_t}$ (We propose to put more of this information in Fig. 1 right)
>
> > Figure 2, is the dotted lines [...]
>
> The *slopes* of the dotted lines (secants) provide the weights (the information needed for optimisation).
>
> > Corollary 5.4, i guess could also be stated with just the weak learning assumption?
>
> We assume the reviewer talks about Corollary 5.6 (there is no Corollary 5.4). No, we need all assumptions. All related parameters indeed appear in the bound, but there is also a rationale for their appearance, see [RA5.4E] [RA5.4.2].
>
> > Why does figure 3 and 4 come before figure 2?
>
> As far as we can tell, Fig. 2 is on page 7, Fig. 3 on page 8, Fig. 4 on page 9 so the statement does not hold. As for where they appear wrt citation, there is probably a bit of LaTeX optimisation that can be done to fix it. We are happy to do it.
>
> > Line 176 $\nu_t$ depend on $h_t$ through $M_t$ is this a problem.
>
> We do not use notation $\nu_t$ here. We assume the reviewer means $\tilde{\eta}_t$. Dependence on $M_t$ is a requirement: otherwise it would unfairly penalize the weak learner. Consider for example that a weak learner predicting *all* classes well but with very little confidence could in fact not satisfy the weak learning assumption if we drop the normalization by $M_t$. In fact, many papers *directly assume* that $h_t \in [-1,1]$, see for example [ssIB] (our reply to reviewer bfG5).
>
> > Why isn't it the case that the weak learner output the hypothesis $h_t$ such that $\nu_t \geq \gamma$ (no absolute value)
>
> (again, we consider that the reviewer talks about $\tilde{\eta}_t$) This is one beautiful feature of boosting: if the weak hypothesis is very bad, say $\tilde{\eta}_t < -\gamma$, then its negation is very good ($\tilde{\eta}_t > \gamma$), which is captured in a $<0$ leveraging coefficient !
>
> > What if the weak learning gaurantee was given in terms of the the loss funciton $F$ ?
>
> The weak learning assumption would then be "just about one loss" and thus very restricted. The adopted formulation is much more general.
>
> > Can you come up with a setting [...] compare it to the best known bound for that setting?
>
> We assume that the reviewers want to know how close we can come to an optimal, *loss-dependent* bound ?
>
> Can we be close to the best rates for *some* losses ? The answer is yes. Consider the logistic loss. It comes from [tAP] (additional bibliography above) that our rate is within a constant factor of the one shown for the logistic loss, which is then shown to be optimal in [tAP].
>
> Are we this good in the general case (i.e. for all losses) ? Certainly not: we do not exploit curvature  for strongly convex losses in the same way as Adaboosting does so we are as suboptimal from AdaBoost-type boosting for the exponential loss as logistic gradient boosting is [tAP]. Note that our remark hits at possible improvements of our algorithm to then capture a generalized notion of curvature for better boosting rates, but only for some losses.

---

> ### Comment · Reviewer_TGRx · 2024-08-08
> **Weak learning assumption 5.5.**
>
> The weak learning assumption I know is from "Boosting" Robert E. Schapire and Yoav Freund page 48.
> "Specifically, we say that the empirical $\gamma$-weak learning assumption holds if for any distribution $D$ on the indices $\{1, \ldots, m\}$ of the training examples, the weak learning algorithm $A$ is able to find a hypothesis $h$ with weighted training error at most $\frac{1}{2}-\gamma$ :
> $$
> P_{i \sim D}\left[h\left(x_i\right) \neq y_i\right] \leq \frac{1}{2}-\gamma .
> $$"
> or equivalently since $h\in \{-1,1\} $
> $$
> E_{i \sim D}\left[h\left(x_i\right)y_i\right] \geq 2\gamma.
> $$
> Assuming the above would lead to a $\gamma/M_t$ margin in assumption 5.5 as far as I can see, and in the setting presented in "Boosting" where $h\in \{-1,1\} $ recover that of "Boosting".
>
> Can you point to other literature of boosting where the assumption 5.5. is made or is this something that was introduced in the paper?
>
> Sorry if I made a blunder but I were not able to find the references [RWL3], [RDRF] and [mnwRC] can you point me to them, further I can not find [ssIB] in "(our reply to reviewer bfG5)".
>
>
> Furthermore, I would like to be sure that I understand correctly, that the techniques used in the paper of 0'th order optimization is "known"? (line 44-47) and the contribution of the paper is to introduce/make the connection to boosting? If this is correct is there a specific reference(s) that your work builds upon and which you think should be cited in the Related Work/ or added to line 44-47. If it is not correct please point out the novel new ideas you use in terms of 0'th order optimization.

---

> > ### Author Response · Authors · 2024-08-08
> > **On the weak learning assumption, some previous uses, and tools and techniques that we introduce not used in 0th optimisation.**
> >
> > > Assuming the above would lead to a $\gamma'/M_t$ margin in assumption 5.5
> >
> > It is in fact $2\gamma'/M_t$, but factor 2 is a detail: what is more important is that $M_t=1$ and so it simplifies to $|\tilde{\eta}_t| \geq 2\gamma$, and the right-hand side is of the same order as ours. Note that this also goes the other way as well: take some $h_t/M_t \in [-1,1]$ that satisfies Assumption 5.5. Classifier $h'_t = \mathrm{sign}(h_t/M_t)$ would then satisfy [fsAD]'s weak learning assumption with $\gamma' = \gamma/2$. Constant 2 just change the exponential's inner constant in (21) of [fsAD] as a function of our $\gamma$ (2 becomes 1/2) and does not change the convergence rate's order. Assumption 5.5 is thus equivalent to [fsAD]'s weak learning assumption when $h_t = -1,1$.
> >
> > In the general case, division by $M_t$ is crucial. If we don't do it, it would be intuitively hard to prove any sort of weak-to-strong boosting result: suppose that on a call, the weak hypothesis has huge positive $y_i h_t(x_i)$ on the example $i$ that has the smallest, minute order non-zero weight, and zero on all others. Without using the division by $M_t$, this hypothesis passes the weak learning assumption but it is obvious that it would be of no use to the ensemble. Dividing by $M_t$, it fails the weak learning assumption and thus cannot be returned by the weak learner.
> >
> > One can remark that the first use of ${\tilde{\eta}_t}$ is in [ssIB] (it is their $r_t$). Here are a sample of papers that previously used a weak learning assumption like ours:
> >
> > [mnwRC] Y. Mansour, R. Nock and R.C. Williamson. Random classification noise does not defeat all convex potential boosters irrespective of model choice. ICML 2023 (see their Theorem 1)
> >
> > [nawBW] R. Nock, E. Amid and M. Warmuth. Boosting with Tempered Exponential Measures. NeurIPS 2023 (their $\rho_t$ generalizes $\tilde{\eta}_t$)
> >
> > [msAT] I. Mukherjee and R. Schapire. A theory of muticlass boosting. NeurIPS 2010 (our formulation is a special case of theirs because their cost matrix is real valued and authorized to change at each iteration, so we can put $\pm h_t$ as cost and use sign($h_t$) as the class in the cost argument)
> >
> > [osOA] K Oono and T. Suzuki. Optimization and Generalization Analysis of Transduction through Gradient Boosting and Application to Multi-scale Graph Neural Networks. NeurIPS 2020 (Their proposition 6 in their appendix gives the equivalence with our formulation via its point 2. $\gamma$ can be found in (5))
> >
> > [sakmmnsxFW] A. Soen, I. Alabdulmohsin, S. Koyejo, Y. Mansour, N. Moroosi, R. Nock, K. Sun and L. Xie. Fair Wrapping for Black-box Predictions. NeurIPS 2022 (Point (i) of their Assumption 2)
> >
> >
> >
> >
> > > Sorry if I made a blunder but I were not able to find the references [RWL3], [RDRF] and [mnwRC] can you point me to them, further I can not find [ssIB] in "(our reply to reviewer bfG5)".
> >
> > It is for us to apologize, as we probably did not make this explicit enough: those references can be found in the webpage using the search tool of the browser. For example, on a Mac, select [RWL3] => command+C => command+F => command+V will display its four occurrences on the page, one of which is the sought reference in part 7/8 of our reply to reviewer bfG5.
> >
> > > Furthermore, I would like to be sure that I understand correctly, that the techniques used in the paper of 0'th order optimization is "known"? (line 44-47) and the contribution of the paper is to introduce/make the connection to boosting?
> >
> > No, the contribution of the paper also encompasses new tools that we introduce. *It is our fault if this was not clear enough from the paper*. L46 indeed says that some tools can be found in 0th order optimization: it is the secant. However, we should have made explicit after that we also introduce a new notion that seems to be crucial for our analysis, the higher-order v-derivative information with variable offsets. This notion is not even defined in our bedside book of quantum calculus. We refer to [RNT2] above for a more technical explanation of our contribution (we have not seen our technique replacing the classical Taylor expansion by a bound involving multiple order v-derivative information in 0th order optimisation).

---

> > ### Author Response · Authors · 2024-08-09
> > **Apologies for the invisibility of some "Official Comments" we asked you to check !**
> >
> > It indeed seems that some "Official Comments" we asked you to check via tags were in fact not visible from your browser.
> >
> > Please accept our sincere apologies for your time wasted in trying to find them. Hopefully, this is now fixed.

---

> ### Comment · Reviewer_TGRx · 2024-08-08
>
> Thanks for taking the time to reply my questions and correcting the $2\gamma$.
>
> Regarding: "$h_t/M_t \in [-1,1]$ that satisfies Assumption 5.5. Classifier $h'_t = \mathrm{sign}(h_t/M_t)$ would then satisfy [fsAD]'s weak learning assumption with $\gamma'=\gamma/2$"  could you show this derivation - thanks for pointing out the other implication it was insightful for understanding/drawing the connection of/the motivation of  Assumption 5.5. better for a person only knowing boosting from the point of view that is presented in "Boosting" Robert E. Schapire and Yoav Freund page 48. If you could also add some more intuition of why the labels in the assumption is allowed to change it would be good (from you comment it seems to not allowed in the setup of "Boosting" Robert E. Schapire and Yoav Freund where the surrogate lose is exp-loss - and if the learner is allowed to change them totally as pleased or what is allowed?
> .

---

> > ### Author Response · Authors · 2024-08-09
> > **On the relationship between some weak learning assumptions**
> >
> > Our comment on the equivalence of past weak learning assumptions (quoted by the reviewer) seems to only work provided more constraints are put on $h_t$. Apologies for making a general statement out of it.
> >
> > Here is a refined statement and proof sketch involving the min / max absolute values of $h$. Denote $h$ the weak hypothesis, $M$ its empirical max, $\textbf{w}$ the weights summing to 1 (normalized). Let $m$ denote its empirical non-zero min in absolute value. Since $yh = |h| y \mbox{sign}(h) = |h|(1 - 2 [y\neq h])$ ($[.]$ is the indicator variable), The WLA implies
> >
> > [1] $\gamma \leq \sum_i w_i y_i h(x_i) / M = \sum_i w_i |h(x_i)|/M - 2 \sum_i w_i (|h(x_i)|/M) [y_i\neq h(x_i)] \leq 1 - (2m/M) P$,
> >
> > $P$ being the empirical risk computed on $\textbf{w}$. In summary,
> >
> > [2] $P \leq \frac{M}{m} \cdot \left(\frac{1}{2} - \frac{\gamma}{2}\right)$
> >
> > Let (A) denote the assertion $(m/M) \geq 2/\left(1+\frac{1}{1-\gamma}\right)$.  If (A) is true, then we get $P \leq (1/2) - (\gamma/4)$, and we are done (for **$\gamma' = \gamma/4$**).
> >
> > It seems (A) can be further weakened by more sophisticated arguments but we (expectedly, in fact) do not get to a point where the implication holds however small would be $m/M$.
> >
> > Where those weak learning assumptions find their "equivalence class" is in the observation that for each of them, flipping a fair coin to decide the class would never satisfy the weak learning assumption and thus the weak learner has to effectively "learn" some dependence between observations and classes.
> >
> > Apologies (twice !), for eventually misleading the reviewer.

---

> > > ### Comment · Reviewer_TGRx · 2024-08-13
> > >
> > > Thanks for taking the time to reply to my questions and helping me understand your work better - I still have a hard time understanding how the complexity of the weak learner is changed(increased) when one can also ask for samples with opposite labels but I think this is out of the scope of this review. I don't have any further questions currently so I wanted to wish you good luck.

---

### Official Review · Reviewer_XK3t · 2024-07-11

**Soundness:** 3
**Presentation:** 3
**Contribution:** 3
**Rating:** 7
**Confidence:** 3

**Summary:**

The paper investigates the theoretical aspect of boosting algorithms in machine learning. The authors propose a new algorithm, SECBOOST, which aims to optimize any loss function using zeroth-order information. This approach dis different from traditional boosting methods that require first-order information such as gradients. By leveraging tools from quantum calculus, the paper claims to extend the applicability of boosting to loss functions that are not necessarily convex, differentiable, or even continuous. The core contribution is the demonstration that boosting can be effectively performed without relying on derivative information.

--

I have read the rebuttal and other reviews. Rating unchanged.

**Strengths:**

I like this paper which is somewhat different from the majority of boosting related work. Though the outbreak of deep learning methods generally comes with derivative info, a vast variety of problems don't. This paper provides theoretical contributions of designing boosting algorithm for any loss function whose set of discontinuities has zero Lebesgue measure which is a pretty general setting. It feels like a piece missing for boosting algorithm literatures and it's nice to have that eventually.

**Weaknesses:**

I would connect more to real-world applications. This doesn't necessarily mean to run experiments, but at least providing examples about such kind of loss functions and their importance in real world would be helpful.

In addition, sometimes the loss function is differentiable yet getting the derivative could be expensive. Some discussion around performance vs. cost will be very helpful. This part is optional but would make the paper much stronger.

Some of the material in appendix III are actually insightful and helpful. Suggest to move some to the main body.

**Questions:**

Please see the strength/weakness section.

---

> ### Author Rebuttal · Authors · 2024-08-03
>
> We thank the reviewer for the whole content of the strength section, which summarizes the key strengths of our approach.
>
> > I would connect more to real-world applications.
>
> Even when we deliberately formatted our paper for a theory report, we understand the reviewer's standpoint. We in fact ran experiments (see our supplementary information) but these were really just meant as a "trial of fire" for our theory, using eventually some very "nasty" losses (see the spring loss). We were please that it works but beyond that, the test of real-world applications is important but would deserve separate consideration, because of the potential that offers our algorithm to deal with very complex settings: consider for example adversarial learning. In this case a "robust" empirical loss is trained with the objective to yield good models on the "actual" domain. The mainstream consists in designing the empirical "robust" loss *using data modifications*. One could also think of adding the possibility (or replacing data modifications with) designing the *loss* itself to prevent bad outcomes on generalization (e.g. to prevent categories of large margins). This could be computationally much more efficient. Regardless of the loss' design, it could be used in our algorithm *as is*, which we believe shows a strong benefit of our approach.
>
> > In addition, sometimes the loss function is differentiable yet getting the derivative could be expensive.
>
> The reviewer is right. We suggest to put [RGRI] (answers to reviewer 8gCx) in the camera ready using the additional page.
>
> > Some of the material in appendix III are actually insightful and helpful
>
> We understand the reviewer would like some details (at least about implementation tricks) to be put in the main file. We would be happy to oblige using part of the +1 page camera ready.

---

> > ### Comment · Reviewer_XK3t · 2024-08-13
> >
> > Appreciate the feedback. I have also read other reviews and discussions and overall I think this is a solid work with minor limitations to be addressed in the next edit. I'll leave my score unchanged (7-accept).

---

### Official Review · Reviewer_8gCx · 2024-07-19

**Soundness:** 3
**Presentation:** 2
**Contribution:** 3
**Rating:** 6
**Confidence:** 2

**Summary:**

-	This paper discusses an alternative boost algorithm by using the zeroth-order optimization technique. The key benefit by using such technique is that it does not require the loss function to be convex, differentiable or Lipschitz. They provide theoretical results and validate them with experiments.

**Strengths:**

-	The contributions from this paper in terms of the generalizability are clear and interesting.
-	If the authors’ claims are proper and assumptions are not hard to satisfy, their contributions are definitely very important in extending the loss function class to be as general as we know in the loss properties.

**Weaknesses:**

-	I do not understand some assumptions well, which makes me a bit confused on the strength of the algorithms and subsequent theoretical contributions. And I think more descriptions related to the condition should be provided:

First, Assumption 5.4, is this easy to verify for some given learners? Since it is quite convoluted and from my perspective, it seems more like an intermediate result which requires some efforts to analyze; Furthermore, will some structure properties of the loss function depend on $\rho$. If the loss is super bad, would not the associated Assumption 5.4 fail? How restricted is such Assumption 5.4?

Then, Assumption 5.5, this is not the standard weak learning assumption in literature, since the standard weak learning condition is the unweighted version with taking expectation for the predictor and labels. And I am not sure why the condition is imposed for all $t$ and requires conditions for the subsequent $h_t$ beyond the initial weak learner. I am not sure whether it is the traditional weak learning assumption in boosting literature. If so, more reference and / or  related discussions should be provided.

-	There are some inconsistent notations and definitions that appear here and there in the paper, which causes readers confusion, e.g., 1-order and first-order, zero-order and zeroth-order. And in Page 2, both $F(S, h)$ and $F(\cdot)$ appear as well as its use subscript or superscript alternatively for different losses, which can be improved. Therefore, it is very confusing in Theorem 5.3 and Corollary 5.6 when both of them appear at the same time. Besides, there are some typos and grammar mistakes around the paper and the paper needs some rewritten to improve the writing.

**Questions:**

-	What are examples of the strict benefits of the boosting for any loss function beyond those requiring gradient information? That is, the true computational / generalizability benefit of SECBOOST over existing first-order boosting methods with function approximations.  I am not super familiar with all first-order boosting methods, however, I guess for those non-smooth loss objectives, we can smooth the original non-smooth loss objectives and do boosting for these surrogate loss. Please correct me if this is not case or quite nontrivial for some general losses.

**Limitations:**

The authors mention some limitations related to their Assumption 5.4 and 5.5. However, I still feel like more discussions related to the scope of these assumptions in terms of the loss function (like the weakness and question I pointed out) to increase the accessibility of the paper and help readers understand better.

---

> ### Author Rebuttal · Authors · 2024-08-03
>
> We thank the reviewer for writing that our contribution is "[...] potentially very important [...]" and hope to give here the arguments sought by the reviewer to enforce further this claim.
>
> > First, Assumption 5.4, is this easy to verify for some given learners? [...]
>
> [RA5.4E] This assumption is indeed technical and comes from the nature of the losses that we minimize (for another set of arguments to explain it, see [RA5.4.2]). There is in fact no "super bad" losses but only locations on the loss landscape that prevent further optimization. In classical non-convex optimization, these are just local minima, which explains that a measure of convergence is the expected gradient's norm in such works. The equivalent in our case is the numerator of $\rho_t$. If it is zero, there is barely anything we can do to get to a better solution and assumption 5.4 breaks. if however it is not the case and our numerator is strictly positive, then we can guarantee a >0 rate.
>
> Why do we need to include the denominator of $\rho_t$ ? It appears to be necessary because we consider v-derivatives: the variation of the function is thus not exactly local like for a derivative and we need to factor in the possibility for the loss to "jiggle a lot" locally, which blurs the information of the secants for convergence. The denominator of $\rho_t$ factors in second-order v-derivatives. The reviewer may think of classical curvature in differentiable optimisation. A small denominator prevents lots of jiggling where the current solution stands, and thus better rates. See also [RA5.4] above.
>
> Is this a limitation of the use of v-derivatives ? Quite the opposite in fact: our algorithm includes the possibility to escape local minima ! This feat somehow comes from the use of the v-derivatives (and their higher orders) that authorise "seeing past local minima" [RA5.4.2]. To get such a guaranteed feast would however necessitate to optimize further the offset oracle, which delivers the local "horizons" the strong learner can look for better solutions. Such would surely necessitate a work / paper of its own.
>
> > Then, Assumption 5.5, this is not the standard weak learning assumption [...]
>
> [RA5.5E] It is in fact the standard assumption, we refer to [RWL3] for a parallel discussion. The standard weak learning assumption includes the weights as a distribution and are thus normalized. If the reviewer means normalization for the hypotheses, then this became standard from [ssIB] since convergence does not depend anymore on the normalization constraint. See for example [mnwRC] (bibliographical references above).
>
> > And I am not sure why the condition is imposed for all $t$ and requires condition for the subsequent $h_t$ beyond the initial weak learner [...]
>
> [RA5.5F] There might be a misunderstanding here: the weak learner is not assumed to change. Only the weak hypotheses it returns can change. It is standard to assume that each of them needs to satisfy the weak learning assumption because otherwise the weak learning assumption is so weak that the weak learner could just return an unbiased coin as predictor, which would obviously be useless to improve the ensemble.
>
> We would be happy to add some additional content to make [RA5.5E] [RA5.5F] clear.
>
> > There are some inconsistent notations and definitions
>
> We indeed have overloaded some notations (like $F$, which is indeed used both for the pointwise loss and for the population loss) in the hope of limiting notational inflation. If the reviewer feels like it is not good for readability, we are happy to reverse the trend.
>
> > What are examples of the strict benefits of the boosting for any loss function beyond those requiring gradient information? [...]
>
> [RGRI] This is an excellent question. One answer, which we hope will have become more clear at this point thanks to [RA5.4E], is that our algorithm comes with the possibility to escape local minima, which cannot "naturally" be escaped for gradient approaches because a gradient just provides variation information at the exact point where the solution stands.
>
> Another answer is the one that has motivated the field of 0th order optimisation: computing gradients can be expensive compared to loss values (think "green AI"). This of course assumes in general that gradients are "estimated" (e.g. using autodiff), but even when they are not, remark that this then requires the computation of another function than the loss, which, regardless of how it is done, always require at some point some additional calculus / computation.
>
> A last answer comes from the one that has originally motivated the Ada-boosting field: non-differentiable losses are difficult to optimize (first and foremost because the gradient "trick" is not accessible) and so one can instead optimize a "nice" surrogate loss (the exponential loss in the case of AdaBoost). But this comes at a price, which is that we just do not optimize the "ideal" loss anymore *directly* (the 0/1 loss in the case of AdaBoost). There is thus less that can be told for the optimization of this "ideal" loss. Our algorithm offers the possibility to directly target the optimization of this ideal loss without resorting to surrogates, *with guaranteed rates*. Of course, as we write in L235-L237, this may come with additional design choices for the oracles.
>
> We hope the limitations section in the review is now adequately addressed.

---

> ### Comment · Reviewer_8gCx · 2024-08-09
>
> I acknowledge and thank the author for their response.
>
> Btw, do the authors forget to put [RWL3] [ssIB][RDRF] in their response to the Reviewer bfG5? I still cannot find them even searching them by command + F...
>
> And one rebuttal suggestion related to that is to put the parts of some common responses to the Global response instead of letting each reviewer search where it is. Thanks a lot!

---

> > ### Author Response · Authors · 2024-08-09
> > **Sincere apologies**
> >
> > We would like to thank the reviewer for sending these comments, which led us to realise that some comments we submitted were probably not visible (from any reviewer, it actually seem ?).
> >
> > We hopefully have fixed this issue. Sincere apologies for wasting your time !!

---

### Official Review · Reviewer_bfG5 · 2024-07-25

**Soundness:** 3
**Presentation:** 1
**Contribution:** 2
**Rating:** 5
**Confidence:** 1

**Summary:**

Boosting can be regarded as a general optimization problem, and most of the currently popular Boosting techniques tend to do so, and do it by using $1^\text{st}$ order (gradient) information to minimise a loss function.
This work proposes an algorithm to minimise an arbitrary loss function using only $0^\text{th}$ order information.
The authors prove that this method converges for essentially any loss function, which is unprecedented in the field.

**Strengths:**

1. The writing style is quite acceptable. I could understand the local meaning of pretty much every sentence.
2. The result covers a **very** wide class of functions. I agree with the authors that putting it as "essentially all functions" is a fair claim.
3. The use of quantum calculus is interesting and adds flavour to the work.
4. The authors are upfront about disregarding generalisation aspects.

**Weaknesses:**

### **Note on the choice of primary area:** This is much more of an *optimisation* than a *learning theory* paper
The choice of primary area for this paper (learning theory) was poor.
This work is far better placed within the area of optimisation than in learning theory.
This is quite clear throughout the technical parts of the paper, and the authors even write "We do not investigate the questions of generalisation, which would entail specific design choices about the loss at hand" (within the paragraph after line 74), so I expect the authors to agree with this assessment.
Given that, I honestly fail to see how the authors judged their choice of primary area to be the one that would lead their work toward the most appropriate reviewers.

This issue could have been significantly mitigated by an especially strong presentation of the work.
Instead, I found the presentation to be, at the very best, fair (I elaborate on this below).

Overall, this work turned out to be laborious to review.
It is not that the technical side is particularly complex, but there are many moving parts and nested concepts, making it laborious to keep track of everything without a clear roadmap.
Especially when coming from a learning theory background (which, again, seems to be the authors' target audience), overcoming the weak presentation requires an amount of effort that I am not sure is reasonable for the authors to demand from the reader.

Ultimately, this matter ended up lowering my confidence score since, for example, it is definitely "possible I am unfamiliar with some pieces of related work".
Still, within the resources I had at my disposal, I did my best to make my review useful to the authors and the community as a whole.


### A technical (reformulated) summary of the work's contribution

To help clarify my understanding of the work, I will start with an abstract of the paper in more technical terms than what I offered in the "summary" field above.
The goal is to make it easier for the authors to point out and correct potential misunderstandings.
(It is based on the definitions from Section 3 starting at line 74)

As mentioned in the note above, the authors completely disregard generalisation.
Thus, only the training set $S = \{(\mathbf{x}_1, y_1), \ldots, (\mathbf{x}_m, y_m)\}$ is relevant to the work.
In particular, given a hypothesis $h\colon \mathcal{X} \to \mathbb{R}$ (not the binary $\{-1, 1\}$-range as in the classical setting), we may simply consider
$$f_i \coloneqq h(\mathbf{x}_i),$$
defining a vector $\mathbf{f} = (f_1, \ldots, f_m) \in \mathbb{R}^m$,
since, again, **the work is oblivious to any evaluation of hypotheses outside of the $m$ points in the training set**.

Let $\mathcal{G}$ be the hypothesis class of the weak learner, i.e., the set of all hypotheses that the weak learner can output.
> NOTE 1: Not to be confused with the set $\mathcal{H}$ defined by the authors to be that of all classifiers attainable via linear combinations of hypotheses in $\mathcal{G}$.
I am highlighting this as it is a bit unusual in the context of classical boosting, where one typically argues in terms of the class of weak hypotheses $\mathcal{G}$ rather than the class of ensemble classifiers $\mathcal{H}$.

For simplicity, let us assume that $\mathcal{G}$ is finite:
Let $n = \lvert \mathcal{G} \rvert$ and, employing the notation above, consider the enumeration $\mathcal{G} = \{\mathbf{f}^{(1)}, \ldots, \mathbf{f}^{(n)}\}$.

Finally, consider a generic loss function $F\colon \mathbb{R} \to \mathbb{R}$.

The problem attacked by the paper then becomes

**Target Problem (reformulated):**
Given **fixed vectors** $\mathbf{f}^{(1)}, \ldots, \mathbf{f}^{(n)} \in \mathbb{R}^m$
and associated labels $y_1, \ldots, y_m \in \{-1, 1\}$,
and a loss function $F\colon \mathbb{R} \to \mathbb{R}$,
find coefficients $\alpha_1, \ldots, \alpha_n \in \mathbb{R}$ to minimise
$$\sum_{i=1}^m F\left(y_i \sum_{j=1}^n \alpha_j f^{(j)}_i\right).$$

> NOTE 2: In my first question below, I've explicitly asked the authors to confirm that this formulation is an accurate summary of the problem.

Naturally, the difficulty of the problem depends on the properties of $F$.
The central claim of the paper is to provide an algorithm to effectively find such a set of coefficients while requiring very little from $F$:
That its points of discontinuity form a set of measure zero.
Moreover, the authors provide guarantees on the sparsity of the solution as at most $T$ coefficients are non-zero, where $T$ is the number of boosting steps performed by the algorithm.

> Notice that I omitted the $\gamma$-weak learner from the reformulation above because I had some issues with the authors' definition.
I dedicated a subsection to this matter below.

### Weaknesses

> My goal is to be objective and direct, but I acknowledge this can give the text a bitter-like tone. I apologise if I sound too harsh in the review.


The most pervasive weakness of the work is that, while the writing style itself is acceptable, the overall presentation is poor when considering more global aspects.
All the following points stem from this issue to some extent.

1. The authors are not sufficiently clear and explicit in enumerating the claimed contributions.
    The closest to that would be the paragraph starting at line 38, however, one cannot conclude that the claimed contribution is restricted to the contents of that paragraph.

After reading the paper, I suspect that the claimed contribution boils down to (put roughly)
> We provide the first study on the convergence of $0^\text{th}$ order optimisation methods for boosting under a general loss function.
Moreover, we ensure convergence for the widest class of loss functions seen so far in the field of $0^\text{th}$ order optimisation in the ML setting.
Namely, we cover all functions whose points of discontinuity form a set of measure zero.

In particular, my understanding is that
- The authors do not claim any new techniques used to obtain their results.
    In fact, very little emphasis is given to the proofs, with all of them being left to the appendices and no sketches being provided.
- As the text does not provide explicit insights into the difficulty of the problem they are solving, the authors do not claim the problem is particularly challenging under the light of the usual frameworks used for this kind of problem.

This is a fair setting for a paper and I would lean towards acceptance provided the authors do a good job in
establishing the relevance of filling the gap in the literature that they claim to have filled.
To make this point clearer, an informal and exaggerated version of it would be "Totally convincing the reader that others weren't aware of the gap and ignored it out of disinterest" (I am not suggesting this was the case).
Alternatively, but preferably both, the authors could solidly establish the significance of the uniquely wide class of loss functions they cover.
I believe that the authors failed to do either, and this is the main reason why I recommend rejection.

While the following points are not minor, the authors can regard them as less critical than what I have discussed above.

2. Theorem 5.3 fails to provide a reasonably self-contained statement that leads to something close to what is claimed in the paragraph starting at line 38.
    Instead, the authors' phrasing of the statement resembles more that of a lemma, in that it requires multiple logical steps and the reference to other results to properly grasp its significance.
    I can see a good formal rephrasing of the statement can be hard to achieve, so the authors could consider adding a less formal version of result to the introduction.
    That would also help greatly with the former point.

3. The authors do not provide an exact statement for the optimisation problem at hand (something in the direction of the *Target problem* above).
    It is important to know early and with absolute certainty what the authors are attacking.

4. **From the perspective of learning theory**, some of the motivation provided for $0^\text{th}$ order methods seems misplaced:
    1. On the theoretical side, $0^\text{th}$ order optimal weak to strong learners are known to exist: see [1].
    2. On a more practical side, in general, the performance of boosting methods does not really come from the minimisation of the associated loss function.
        See, for example, the discussion in [2, Section 7.3: "Loss Minimization Cannot Explain Generalization"].
    3. I recognise that the authors explicitly dismiss matters of generalisation, but I still believe the remarks above are relevant in assessing the motivation for the work.
    The authors may consider bringing up these points in some form, perhaps by mentioning how the related works compare to their results in this regard.

5. I am confused by the authors' concept of what constitutes "traditional" boosting.
    To me, AdaBoost is the most prototypical and traditional boosting algorithm.
    However, for example, in line 131 it is clear that the authors believe that "traditional" boosting requires first-order information, which is not the case for AdaBoost.

6. The discussion around Assumption 5.4 is too loose to fully justify an assumption that seems to be quite relevant (see the role of $\rho_*$ in Corollary 5.6).
    I understand the point made just above the assumption, but I believe that would only suffice to fully justify some special treatment of arbitrary small $\rho_t\text{s}$.
    I expected a formal or at least deeper discussion of the significance of the assumption.


### References

[1]: Larsen, K. G. (2023, July). Bagging is an optimal PAC learner. In The Thirty Sixth Annual Conference on Learning Theory (pp. 450-468). PMLR.\
[2]: Schapire, R.E. and Freund, Y., 2013. Boosting: Foundations and algorithms. Kybernetes, 42(1), pp.164-166.


### Minor issues and suggestions

#### Issues with the definition of $\gamma$-weak learner

1. The authors, unfortunately, do not define it in Section 3, "Definitions and notations", delaying it to page 7 (Assumption 5.5 at line 175).
2. The definition is not self-contained, as it depends on $\tilde{\eta}_t$ which itself depends on other parameters (see Eq. 18).
3. Honestly, I do not recognise that definition as "the traditional" one (see line 174).
    To me, that would be a $\gamma$-weak learner that, for any distribution over the training set, outputs a (binary) hypothesis whose average error is at most $1/2 - \gamma$.
    Of course, the paper discusses non-binary hypotheses so that a definition in terms of edges is natural.
    Still, I do not think the equivalence/analogy is immediate enough, and, regardless, the authors should provide a self-contained and formal definition of $\gamma$-weak learner, ideally in Section 3.
    Step 2.1 of Algorithm 1 does not make it totally clear what are the inputs and outputs of the $\gamma$-weak learner.
4. Assumption 5.5 might be less "global" than one could expect.
    It depends on the definition of $\tilde{\eta}\_t$, which depends on $M\_t \coloneqq \max\_i \lvert h\_t(\mathbf{x}\_i) \rvert$, making Assumption 5.5 have a different strength at each round if one parses Eq. 18 as
    $$\tilde{\eta}\_t = \frac{1}{M\_t} \sum_{i=1}^m \frac{\lvert w\_{ti} \rvert}{\sum\_i \lvert w\_{ti} \rvert} \cdot \tilde{y}\_{ti} h\_t(\mathbf{x}\_i) = \frac{1}{M\_t} \cdot \text{``edge of $h_t$ relative to a normalised version of the weights $\mathbf{w}\_t$"}.$$

> NOTE 3: Despite the difficulties, I believe to have eventually understood what the authors meant by $\gamma$-weak learner. At Step 2.1,
- the notation $\lvert \mathbf{w}\_t \rvert$ refers to the vector $(\lvert w\_{t1} \rvert, \ldots, \lvert w\_{tm} \rvert)$ (I do not recall whether this was defined);
- the notation $⨉$ denotes the weak learner (why not something clear like $\mathrm{WeakLeaner}$?);
- the definition of $\mathbf{S}\_t$ in the comment (why there?) sets the pseudo-labels;
- the weak learner $⨉$ operates on the training points $\mathbf{x}\_i$ with the pseudo-labels $\tilde{y}\_{ti}$ and weights $w\_{ti}/\lVert \mathbf{w}\_t \rVert\_1$;
- $⨉$ provides a hypothesis $h_t$ with an edge of at least $\gamma$, where the concept of edge needs a normalisation to contemplate hypotheses with unbounded range; (The authors chose a normalisation factor that is a function of $h_t$ while in points 3 and 4 above I remark that a global factor could be more natural).

> My point is not that it is impossible to retrieve what the authors had in mind, but that they made it significantly harder to do so than what is necessary.
Also, one needs explicit formal definitions to fully appreciate contributions.


#### General minor issues and suggestions

- In the technical summary above, one could cover infinite $\mathcal{G}$ by considering a suitable measure and employing a (Lebesgue) integral in the objective function.
    I haven't thought much about this, but it's too much of a coincidence that the weak regularity assumption on $F$ that the authors found resembles the conditions for integrability so closely.

- The other theorem statements also suffer from the "lemma-like" issue I mentioned above

- Section 4 is somewhat representative of the general issue with the presentation.
    Ideally, a section like this should go along the lines of
    - What is the goal;
    - What are the original concepts;
    - Why they do not suit the current setting;
    - How you modified them;
    - How the change fixes the issue.

    Instead, the definitions introduced are left insufficiently motivated and, indeed, it is not even fully clear how novel the concepts introduced are.

- I suspect you require less from the hypotheses returned by the weak learner than it may appear at line 113 and the subsequent Assumption 5.1.
    As the next note highlights, you don't really need it to be non-zero.
    Finally, since the training set is finite, to have that $\forall i \in [m],\, \lvert h(\mathbf{x}_i) \rvert < \infty$ we only need $h$ to be a real-valued function defined in $\{\mathbf{x}_1, \ldots, \mathbf{x}_m, \ldots\}\subseteq \mathcal{X}$.
    Long story short, I think you only need to consider a weak learner returning functions $h\colon \mathcal{X} \to \mathbb{R}$, which is a very mild assumption.
    (The central point here is that real-valued functions are always bounded on finite sets. Unless one is working with the extended reals, something like $f(x) = 1/\lvert x \rvert$ is not defined at 0.)

- Consider stating more explicitly that the assumption of non-zero predictions (line 113) sacrifices no generality (I was only sure this was the case after checking Section II.2 from the Appendix)

- [Eq. (4)] Consider using $(F(a)F(b))_\alpha$

- Number only the equations that are referenced in the text

- Consider a version of Figure 1 with an extra dimension (a surface plot) as, after all, the Bregman Secant distortion behaves as taking 2 arguments

- Avoid starting sentences with mathematical symbols (e.g., line 93)

- [122] the reference to [47, Appendix, Section 4] has a bit too much packed in it. The authors should consider adding some further guidance or explanation. That should be feasible since the reference tags almost all equations in that section.

- Adding hyperlinks to the steps of the algorithm (e.g., Step 2.1) would make it meaningfully more convenient to navigate the text.

- Algorithm 1 can be made significantly tidier. Doing so would be a considerable improvement in the presentation.

**Questions:**

1. Up to the simplifying assumption of a finite weak hypothesis class, do you agree with the reformulation of the problem I provided above (see NOTE 2)? Naturally, conveying all nuances would require more space. Still, the goal was to provide a fair summary.
2. Why do you believe "optimisation" is not a more appropriate primary area for this work than "learning theory"?
3. What happens to Definition 4.1 when considering $v = 0$?
4. Sorry if I simply missed this one, but what is your actual claim about Assumption 5.4? Is it that you expect it to always hold in practice, that there is an easy way to overcome it, or something else?

**Limitations:**

I am not sure this really applies to the work, but adding something like what I suggested in weakness 4.3 should make some relevant limitations of the work significantly clearer to many readers.

---

> ### Author Rebuttal · Authors · 2024-08-03
>
> We thank reviewer bfG5 for a passionate review. We particularly appreciate that reviewer bfG5 put the key strength of our paper in **bold faces**. This is rarely seen in reviews in general and in our case for example, none of our 6-6-7 other reviews use bold faces to describe the strengths of our paper.
>
> We hope the length of the rebuttal does not discourage the reviewer to dig (it is split in several comments below) -- it should rather serve as a token of our appreciation of the review’s content, disregarding whether the statements made actually hold or our potentially divergent opinions. The review can be split in 3 parts, first discussing the positioning of our paper, then summarising our objective and technical “value” and finally digging into specifics. This rebuttal uses simple tags like [RTB] to point to the relevant parts located elsewhere via a browser’s search.
>
> Our reply contains several points that would add to the paper content. We made sure that this content, alongside the one proposed for the other reviews, would fit in the additional camera-ready page: [RDML][RNT2] [R4.3].
>
> This rebuttal part contains two sub-parts: a reply to the positioning of our paper and a reply to the questions asked in the review.
>
> ## On the the positioning of our paper.
>
> > The choice of primary area for this paper (learning theory) was poor. This work is far better placed within the area of optimisation than in learning theory […] the most appropriate reviewers […]  I did my best to make my review useful to the authors and the community as a whole.
>
> [ROLT] We believe we understand where the reviewer comes from, and we do not share the reviewer's opinion. One can argue that submission "strategies" do not have a clear path, in particular for targetting the humans in the review loop. It is not just about targeting reviewers, ideally the best and most dynamic ACs have the most influential role in the game and this year, some allocation algorithms were different. Factor in the number of submissions and the number submitted in each primary area (unknown in advance, of course) with the risk of "overflowing": the choice of the primary area is then far from being a best bet on getting the best "humans in the loop". So we sticked to a "logical" primary area [RTB]. We may remark that our choice of primary area at least did get us a dynamic reviewer clearly open to discussing our points of disagreements -- we do not know a strategy that grants this automatically :).
>
> ## Questions
>
> > Up to the simplifying assumption of a finite weak hypothesis class, do you agree with the reformulation [...]
>
> De do not, for reasons explained above [RDRF]. However, the reviewer's comments helped us realize a simple improvement that would hopefully alleviate misunderstandings and oversimplifications [RDML]
>
> > Why do you believe "optimisation" is not a more appropriate primary area for this work than "learning theory"?
>
> We explain it in [ROLT] and [RTB]. In one word: history.
>
> > What happens to Definition 4.1 [...]
>
> The v-derivative becomes a conventional derivative and our algorithm then becomes a more classical gradient boosting algorithm (albeit with explicit rates, which again is usually not stated in the state of the art)
>
> > Sorry if I simply missed this one, but what is your actual claim about Assumption 5.4?
>
> [RA5.4.2] It is a very legitimate question ! And perhaps we should have made this part more explicit or formulated in another way. What we mean is that since the rate essentially behaves as $1/\min_t \rho_t$ and is thus vacuous if $\rho_t=0$, we must ensure a strictly positive value for this minimal value, $\rho_*$ (however small it may be). Note that this is not restrictive: we explain in L165-L166 that $\rho_t=0$ means that the algorithm converges to what looks like a (local) minimum. We also explain in L217-L227 that the offset oracle has in fact the "ability" to have the strong learner move to a better minimum. We would like to emphasize that this cannot be achieved with gradient-based algorithms, since the gradient information just gives information at exactly where the algorithm stands and thus leaves it "trapped" if it is a local minimum. Note that we have not investigated optimizing the offset oracle for such tasks since it would probably require a following work/paper of its own. See also [RA5.4E].
>
> We hope the reviewer will find the answers to their questions and the information to fill the gaps they wanted to see filled in their comment "[...] I would lean towards acceptance provided the authors do a good job in establishing the relevance [...]".

---

> > ### Comment · Reviewer_bfG5 · 2024-08-11
> > **General reply to rebuttal (preliminary)**
> >
> > I thank the authors for their detailed response. I acknowledge that my review was likely to be met with resistance and that probably meant a long reply. It seems suitable to reiterate that my sole intention was to provide the community with the most useful feedback I could.
> >
> > I will proceed to address the points raised by the authors in their reply, using multiple comments. To avoid making the discussion even longer, I will not cover all the points, omitting replies to those that I have simply acknowledged.
> >
> > Finally, I ask for the comprehension of the reader for the poor quality of my text, in particular, its length. I wanted to reply as soon as possible and making the text shorter would take too much time.
> > Also, I apologise for the typos in the mathematical expressions in the review. I tried to fix those, but the TeX engine is inconsistently buggy. In particular, many of my curly braces disappeared.

---

> > > ### Comment · Reviewer_bfG5 · 2024-08-11
> > >
> > > I will wait a bit more to update any scores.
> > > For now, the only change in evaluation I am sure is in demand is to retract "Strength 1" from my original review. I clearly cannot "understand the local meaning of pretty much every sentence". I understand what the authors write, not what they mean. Reading their rebuttal made me confident that there is a non-negligible gap between those in their writing.
> > >
> > > I was unable to interact with all points today, but I managed to touch many, at least. This should allow the authors to already interact with them, if they wish so we can gain time. I will keep working on the rebuttal and will try hard to add useful replies with whatever time I am able to allocate to this task.
> > >
> > >
> > > Thanks!

---

> > ### Comment · Reviewer_bfG5 · 2024-08-11
> > **On the positioning of the paper**
> >
> > (This reply references [RTB] but is not my direct answer to that point. I elaborate on [RTB] in a reply to its associated "reply block".)
> >
> > I will start by, hopefully, taking a substantial portion of the discussion out of the way as part of the response to this point (also in [RTB]) seems misdirected. The authors argue (well) that their choice of primary area is **valid**, invoking, for example, historical reasons. This is unnecessary as I have not questioned it. I am well aware that there are no clear guidelines for that choice and, thus, the authors are free to choose any option. Appropriately, I would not let it directly affect my recommendation, and I tried to signal so by keeping that text as a separate subsection in the review rather than putting it as a true "weakness" (bullet) point. Also, see my careful wording in the subtitle of the section.
> >
> > To ensure this is fully clear and to not risk being unfair to the authors: I understand that the choice of primary area should not weigh in my grading. Thus, the authors are not obliged to further discuss the matter with me and are free to ignore what follows. I only kept those paragraphs because I was somewhat surprised by their lack of accountability for the problem (and out of naïve idealism).
> >
> > 11. Reading the authors' reply, especially [RTB], made me confident that they indeed "understand where the reviewer comes from", as they put in [ROLT]. They know of the existence of a "learning theory" *phylum* and a "statistics" *phylum* within boosting (notice that my choice of terms here is derived directly from the authors' text in [RTB]), and they are aware of the divergence in expertise between the two groups. Finally, I suspect that their target audience is mainly the second group or, at least, that they would recognise that experts closer to the second group are likely to have an easier time reviewing their work than those closer to the first.
> >
> > 12. Considering the information available to it, I cannot blame "the system" (algorithms, the chairs, etc) for being "fooled" by the authors' choice of primary area because it fooled me.
> >     Normally, I do not even notice the primary area of the submissions as the abstract usually carries enough information to guide my bidding.
> >     However, for this specific work, I was left thinking about the matter discussed in the paragraph above (11) after reading the abstract. Only then I properly considered the authors' choice, understanding that it meant that the authors judged that their paper was better suited for learning theorists to evaluate than any other community (among the many options offered by the venue).
> >
> > 13. After having a first pass on the paper, it was clear that I had been misguided, so proper reviewing here would be expensive.
> >     Frankly, I considered dropping the review, but ultimately decided to put in the work to go through with it out of care for the community (including the authors, the other reviewers, and, especially, the meta-reviewers).
> >     That's because I immediately conjectured that
> >     > This work was going get assigned mostly to theoreticians which would likely give it low-confidence weak-accepts. Moreover, this would cause the work to need more reviews than usual.
> >
> >     Reading the other reviews largely confirmed my guess and my main point is that I do not think this was a coincidence (I get that the authors disagree with that).
> >
> > 14. The paragraphs above already help to motivate my "lucky guess" (the conjecture mentioned in paragraph 13). Additionally, the reasoning is simple:
> >     1. The authors' choice made it more likely for the work to be assigned to reviewers with a suboptimal match in expertise.
> >     2. That already increases the expected "cost" of the review process (it surely increased mine) and decreases the expected confidence of the reviewers.
> >     3. This could be mitigated by an exceptional presentation, but I did not see that. For some substantiation, notice that the main text of the paper does not even discuss proofs, which tend to be the most demanding portions of the text. So, all that the reviewers are left to evaluate is the presentation of the ideas, their contextualisation in the literature, their relevance, and similar matters. Still, we see low confidence scores.
> >     4. When confused, and lacking the long time required fix that, many reviewers "fail fairly": They assume the authors to be right and give them weak accepts (in this venue, a 6 would be the most likely outcome since reviewers are explicitly discouraged from using the 5 "borderline" score). This is especially true for young researchers, who not only are more susceptible to the issues above but are also understandably less comfortable opposing the opinion of more experienced colleagues. "Defaulting" to positive scores also tends to be much "cheaper", see, e.g., my situation for the opposite example: resisting is "costly".
> >
> > (continues with paragraph 15)

---

> > > ### Comment · Reviewer_bfG5 · 2024-08-11
> > >
> > > (continued from paragraph 14)
> > >
> > > 15. Finally, the last paragraph (14) suggests that the authors' choice could give them some unfair advantage in the process. So, I wish to make clear that **I do not think the authors were malicious with their choice of primary area**. I believe it was just unfortunate. Yet, I maintain my opinion that it was a poor choice as I believe it is demanding significantly more than necessary from everyone involved (in particular, the reviewers are trying extra hard to deliver) and it may, nonetheless, be reducing the quality of the review process to some extent.

---

> > ### Comment · Reviewer_bfG5 · 2024-08-11
> > **Answers to Questions [minor things]**
> >
> > > [Reply to Question 2] We explain it in [ROLT] and [RTB]. In one word: history.
> >
> > I address those points separately in other replies.
> >
> >
> > > [Reply to Question 3] The v-derivative becomes a conventional derivative [...]
> >
> > It does not. I understand that the authors may elsewhere implicitly indicate it does, but I am referring to what is actually written in the definition. As is, Definition 4.1 simply breaks for $v = 0$.
> > (This is not the first time I have gotten the impression that the authors have a hyperreal mental model for numbers. This means little, but I found it curious)
> > Regardless, I realised that this was actually a minor issue that was more suitable for the "Minor issues and suggestions" section. I apologise for the confusion. I only brought this point so that the authors know that it needs a patch.

---

> > ### Comment · Reviewer_bfG5 · 2024-08-11
> > **[RDRF] (As a reply to Question 1)**
> >
> > > The issue with the formulation [...]
> >
> > Which formulation? The authors mentioned two "formulation"s in the previous sentence, making it ambiguous. I will assume it is the one I asked about.
> >
> > > [...] the analysis is then implicitly carried knowing the weak classifiers that are going to be chosen by the weak learner [...]
> >
> > No classifier is "chosen" in the reformulation I mentioned. All of them show up in the sum.
> >
> > > (because the weak learner may well learn in a set of unbounded size !)
> >
> > In general, it can. But, crucially, it cannot in a reply to a question starting with "Up to the simplifying assumption of a **finite weak hypothesis class**".
> >
> > > This forgets that in our case boosting has to be nested [...]
> >
> > What follows in the authors reply are points about the specific solution in the paper. I fail to see why details of one solution could invalidate an attempt to describe the problem itself.

---

> ### Author Response · Authors · 2024-08-03
> **[part 1/8] On the technical reformulation**
>
> [RDRF] The formulation proposed by the reviewer is reminiscent of the game theory formulation for boosting [fsGT] where the full information available for training fits in a matrix or a set of predefined fixed vectors. The issue with the formulation in our case is that the *analysis* is then implicitly carried *knowing* the weak classifiers that are going to be chosen by the weak learner (because the weak learner may well learn in a set of unbounded size !). This forgets that in our case boosting has to be nested also with (i) the computation of the leveraging coefficients (second option Step 2.3) and (ii) the interaction with the offset oracle (Step 2.5) and would render a clean convergence proof definitely a lot more challenging.
>
> [RDML] This being said, the comment helped us realize that we perhaps forgot to complete section 3 by adding a standard ML part at the end ! This part would read as follows (after L81).
>
> "*Our ML setting consists in having primary access to a weak learner WL that when called, provides so-called *weak* hypotheses, weak because barely anything is assumed in terms of classification performance for each of them separately. The goal is to devise a so-called "boosting" algorithm that can take *any loss* $F$ as input and training sample $S$ and a target loss value $F_0$ and after some $T$ calls to WL returning classifiers $h_1, h_2, ..., h_T$, returns classifier $H_T = \sum_t {\alpha_t} h_t$ with $F(H_T,S) \leq F_0$, where the leveraging coefficients are computed by the algorithm. Notice that this is substantially more general than the classical boosting formulation where the loss would be fixed or belong to a restricted subset of functions.*"
>
> (We comment on the $\gamma$-weak learner below [RWL1][RWL2][RWL3])

---

> ### Author Response · Authors · 2024-08-03
> **[part 2/8] (Mis)understanding of our contributions; on our technical material, part 1/2**
>
> > In particular, my understanding is that [...] The authors do not claim any new techniques used to obtain their results […] the authors do not claim the problem is particularly challenging under the light of the usual frameworks used for this kind of problem.
>
> [RNT] It takes a full reading of the review to see that the rest of the review contradicts these statements in places -- and we disagree with them. We however attribute these statements to several things:
>
> the reviewer is clearly knowledgeable about AdaBoost but “reduces boosting” to AdaBoost-ing and never mentions nor discusses our contributions with respect to the gradient boosting current, which is a lot more productive nowadays [RTB]. By definition, gradient boosting exclusively relies on computing *gradients* (and in fact, some versions of AdaBoost does also rely on such computations [ssIB]). From the standpoint of gradient boosting, it is clear that our paper needs to rely on different tools to achieve boosting. Since most of the gradient boosting literature does not have convergence proofs (even less so with explicit rates, even less so in the weak-strong framework) and neither do the sparse boosting results on non-differentiable / non-convex losses (see our L66-L68), it would make sense that original tools **needed to be used** in our case. In fact, it is quite implicit from our L90: to achieve our goal, we need higher-order v-derivatives that are not even defined in a bedside book of quantum calculus ! We also relied on the inference that since the state of the art 0th optimisation relies up to some extent on additional assumption about the loss, there is either something we make possible with boosting that is not yet known to be possible in the classical 0th order setting OR (non exclusive) we pulled new tools to achieve our objectives. Given the sheer amount of work in 0th order optimisation, it would somehow be conceited to claim the former, but we surely can claim the latter, and we believe the reviewer in fact agrees with us (see the points below)
>
> - at this point, we hope the reviewer agrees that there no such thing as a “usual framework” for “this kind of problem”… because there was in fact no framework at all yet defined to properly analyse such problems, because no such result existed before ours. We are the first to provide one, and in fact the reviewers surely agrees with us when they press us to comment on this new parameter $\rho$ that perfectly captures the rate in our case [RA5.4]
>
> (continued below)

---

> > ### Comment · Reviewer_bfG5 · 2024-08-11
> > **Reply to [RNT] 1**
> >
> > > It takes a full reading of the review to see that the rest of the review contradicts these statements in places -- (continues)
> >
> > I eventually understood that the authors were referring to [RA5.4]. I politely ask the authors to try to be explicit whenever feasible in their text since doing so would make their argument easier to follow.
> >
> > > -- (continued) and we disagree with them.
> >
> > I am not so sure the authors do. I am confused by the structure of the argument here. For example, part of one of my two claims discussed here is
> >
> > > [...] the authors do not claim the problem is particularly challenging [...].
> >
> > The authors claim to disagree with that, but by the end of the discussion, in the paragraph starting with
> >
> > > We understand the reviewer would like us to claim that the problem is challenging (last paragraph of *[part 3/8] (Mis)understanding of our contributions; on our technical material, part 2/2*),
> >
> > I understood that the authors are **agreeing** with me, even providing an eloquent explanation for why they do not claim what I said that they did not claim.

---

> > ### Comment · Reviewer_bfG5 · 2024-08-11
> > **Reply to [RNT] 2**
> >
> > My other point under discussion (within [RNT]) is that
> >
> > > The authors do not claim any new techniques used to obtain their results [...]
> >
> > I am also somewhat confused. It seems that the authors misinterpreted the sentence to some extent. The sentence was deliberately crafted to mean what it says: "the authors do not **claim** new techniques". In particular, the authors could develop and use multiple new methods, but if they did not say that explicitly my statement would still hold.
> > The main point here is that it should be much easier to identify the technical contributions of the paper. Among other advantages, very explicit contributions help in the review process. To the extent that it is heuristically recommended by many venues, including the present one (see around L589). In fact, something explicit can be very useful here.
> >
> > **Request**: Could the authors provide a very explicit bullet list of their contributions put concisely?
> >
> > To further clarify my point, consider that the authors reply that
> >
> > > [...] it would make sense that original tools needed to be used in our case.
> >
> > I agree! I was expecting some very clear highlight of those nice tools, discussions about their applicability to other problems, and alike. I thought it would surely be very easy to identify them, but I remain unsure.
> >
> > Part of the reason seems to related to the following.
> >
> > > [...] it is quite implicit from our L90: to achieve our goal, we need higher-order v-derivatives that are not even defined in a bedside book of quantum calculus !
> >
> > There is a small confusion here. From L89:
> >
> > > Higher order $v$-derivatives can be defined [31]
> >
> > The text is ambiguous once more: is it that [31] defines higher-order $v$-derivatives or that the source simply says that those could be defined?
> > Fortunately, this does not matter much since defining the first-order $v$-derivative already defines the higher-order ones by the means of composition. Much like Real Analysis texts simply introduce notations for (the usual) higher-order derivatives without ceremony; it is not really something new.
> >
> > However, indeed, immediately after we read
> >
> > > though we shall need a more general definition that accommodates for variable offsets.
> >
> > This makes it clear that the authors are referring to $\mathcal{V}$-derivatives, where $\mathcal{V}$ is a multiset. That is, a higher order $v$-derivative where the offsets are not necessarily all the same. Of course, I already new that, and my small digression here was an attempt to explain why I got confused. I did not recognise that the authors were claiming that definition as a main technical contribution. This is much clearer in their rebuttal (to me and to other reviewers). I suggest they write it (explicitly!) in the text.

---

> > ### Comment · Reviewer_bfG5 · 2024-08-11
> >
> > > the reviewer [...] “reduces boosting” to AdaBoost-ing
> >
> > The quotes make it impossible to be sure what the authors meant. Just in case, I remark that I do not reduce boosting to AdaBoost. While I do not do research in the "gradient boosting current", I am well aware of its existence. In fact, the existence of those other "currents" of boosting is the base of my argument about the choice of primary area, for example. Moreover, as a learning theorist, if I was as oblivious to the "gradient boosting current" as the authors suggest, that would serve as validation for that argument (the one on the primary area).

---

> ### Author Response · Authors · 2024-08-03
> **[part 3/8] (Mis)understanding of our contributions; on our technical material, part 2/2**
>
> (continued from above)
>
> from the very first comment of the review (**Note on the choice of primary area**), we in fact suspect that the main reason for [RNT] is that we did not format our paper like a “conventional” (if there exist one) learning theory paper, where the introduction or a special section right afterwards usually frames the technical contribution per se in terms of *tools* used and key results achieved, eventually disregarding the problem solved in fine. From here they perhaps concluded that there was nothing worth mentioning and got to the conclusion above that we rebute here. Perhaps they got additionally confused by the statement we make in L46 that “some [tools from quantum calculus] appear to be standard in the analysis of 0th order optimisation”. The “some” mentioned is in fact the basic secant information (we have *never seen the higher-order v-derivative information used; in fact, it has never been defined in quantum calculus either, see above*).
>
> If this is the reason for the reviewer’s comments, then we deeply apologise. We in fact chose not to present our paper this way for space reasons and left the numerous implicit implications of our claims (see the first point above) to “speak for themselves". Since there is one additional page in the camera-ready, should our paper be accepted, it would be trivial to include the following statement in the second part of the introduction (would replace “some of which” and what follows in L45):
>
>  “[RNT2] *Our proof technique builds on a classical boosting technique for convex functions that relies on an order-one Taylor expansion to bound the progress between iterations [nwLO]. In this technique, boosting weights depend on the gradient and thus the sample expectation of Taylor’s remainder becomes a function of the weak learner’s advantage over random guessing, which is guaranteed to have strictly positive value by virtue of the weak learning assumption, leading to a strict decrease of the loss. In our case, we change the Taylor expansion by a more general bounding involving v-derivatives and a quantity related to a generalisation of the Bregman information [bmdgCW]. Getting the boosting rate involves the classical weak learning assumption’s advantage and a new parameter bounding the ratio of the expected weights (squared) over a generalised notion of curvature relying on second-order v-derivatives, quantifying the local potential “jiggling” of the loss (an uncertainty measure, smaller being better). Our algorithm, which learns a linear combination of weak classifiers, introduces notable generalisations compared to the AdaBoost / gradient boosting lineages, chief among which the computation of acceptable ranges for the v-derivatives used to compute boosting weights (which are always zero for classical gradient boosting).*”
>
> At this point, we hope to have clarified our claimed contributions, the technical nature of our work and the novelty of some tools we use *and the fact that there is indeed no such thing as a "usual framework" for our problem*. We also hope that after reading [RA5.4.2], the reviewer will be convinced that our approach brings substantially more than "just" solving a technical problem.
>
> We understand the reviewer would like us to claim that the problem is challenging: such is a matter of personal perception and understanding; claiming it could be seen as somehow pretentious. We are happy to let the reviewer conclude based on the above but we think it is worth mentioning that a problem should just be worth solving, regardless of its technical nature. If the statement were true indeed, then an influential boosting paper would probably not have appeared — or at least not in its form, [ssIB], whose empirical convergence proofs are arguably elementary (given or not the back-then state of the art) but have been instrumental in both the design and convergence proofs of boosting algorithms for numerous settings.

---

> > ### Comment · Reviewer_bfG5 · 2024-08-11
> >
> > > If this is the reason for the reviewer’s comments, then we deeply apologise.
> >
> > The word "this" is replacing a lot here, so I am not entirely sure what the authors mean. If they mean the fact that they do not present their contributions clearly, then, yes, that was the reason. Crucially, there is no need to present those in any predefined format, so long as it is clear. Also, yes, the statement in L46 is confusing, but only because of the overall quality of the writing. The separation between what the authors mean and that they actually mean is substantial. They also have a tendency to keep things implicit without any need, which makes things even worse.

---

> > ### Comment · Reviewer_bfG5 · 2024-08-11
> > **(Mind the **Request** in this reply)**
> >
> > > We [...] left the numerous implicit implications of our claims to “speak for themselves"
> >
> > 1. That explains a lot.
> > 2. Although sometimes contributions do "speak for themselves", it is usually much better to stay humble and assume that does not need to be the case.
> > 3. Making things very explicit here would be helpful.
> >
> > **Request**: Could the authors provide a very explicit bullet list of those "numerous implicit implications of their claims" put concisely?

---

> > ### Comment · Reviewer_bfG5 · 2024-08-11
> >
> > > We understand the reviewer would like us to claim that the problem is challenging
> >
> > I do not. Simple contributions are my favourite, even more so when I am reviewing. I re-read my review and could not locate where I said something that means what the authors seem to think I said. It appears to be yet another case of the authors reading something that was not written. What I implied was that if they established that their result was challenging to achieve (think something like "a centuries old conjecture") it would help us size their contribution. But there are other ways to achieve that, of course.

---

> ### Author Response · Authors · 2024-08-03
> **[part 4/8]  Points 2-4 before references**
>
> > 2. Theorem 5.3 fails to provide a reasonably self-contained statement [...]
>
> What we propose to put in the introduction [RNT2] and at the end of Section 3 [RDML] is also aimed at clarifying this result. If the reviewer also means that it also takes a re-read of the manuscript to grasp some parameters in (20), we are happy to make this summary of what the key parameters are just before the theorem's statement, following a custom often seen in theory papers.
>
> > 3. The authors do not provide an exact statement for the optimisation problem [...]
>
> We conjecture that this comes from the lacking ML part at the end of Section 3 [RDML]
>
> > 4.1 On the theoretical side 0th order optimal weak to strong learners are known to exist: see [1]
>
> [R4.1] This reference is irrelevant for two reasons: [1] (which needs to be combined with [lrOW]) leads to *sample* optimal “AdAboost-ing”. Sample optimality is not our focus. Second, a 0th order optimisation algorithm takes a loss as input and has to work for *large sets of of losses*, not just 1 or a few. All references we put in Table A1 operate on large sets of losses, and the set is even wider for our algorithm, as the reviewer accurately noticed. AdaBoost does not qualify (AdaBoost optimises directly the exponential loss, indirectly the 0/1 loss).
>
> > 4.2 On a more practical side, in general, the performance of boosting methods does not really come from the minimisation of the associated loss function
>
> [R4.2] We wholeheartedly disagree: this statement overgeneralises from [2]’s Section 7.3 and puts it out of its *very specific context*: performance = 0/1 loss, associated loss = exponential loss (the “surrogate” loss).
>
> [2]’s Section 7.3’s “*raison d’être*” is the fact that there is more to just minimising the exponential loss that brings good performance on the 0/1 loss: margin maximisation (AdaBoost’s “dynamics”) is crucial to get there. In fact, in this very specific case, it is not even enough to get good performance: early stopping for AdaBoost is crucial to get statistical consistency [btAI]. Not early stopping would grant boosting statistical consistency *if* the loss is Lipschitz [tBW], which is not the case for the exponential loss. Hence, one can get rid of this early stopping design constraint by just *choosing a different loss* (but then, on an orthogonal performance measure, one faces slower rates [tBW,wvTS]).
>
> Could we remove this *additional* margin maximisation property for the associated loss minimisation to lead to good performance on the 0/1 loss ? It is indeed possible, and simple: just clip the exponential loss by replacing it by $F(z) = \exp(-\max\{z,u\})$ with $u>0$. It is trivial to show that any algorithm substantially beating the trivial max loss $\exp(-u)$ on average would in fact guarantee large margins — and thus good generalisation on the 0/1 loss via standard large margin classification results, e.g. [sfblBT]. Note however that minimisation would have to be carried out using a 0th order algorithm — or our approach !
>
> > 4.3 I recognise that the authors explicitly dismiss matters of generalisation, but […] the authors may consider bringing up these points in some form.
>
> [R4.3] We dismissed generalisation for 2 reasons: (1) it was hard to discuss in the page limit but more importantly, (2) classical matters relevant to generalisation would amount to *restricting* the set of losses OR putting additional constraints on our algorithm, and we chose to stick to the most general setting, thus only focusing on the empirical boosting of any loss. As examples, to get statistical consistency, we would “just” have to restrict the set of losses to Lipschitz losses [tBW]; to get statistical consistency with a strongly convex loss, we would have to consider early stopping; to learn in an adversarial setting in the simplest way with our algorithm, we would probably *not* consider Lipschitz losses because of the then “easiness" of an adversary to play against the learner [cmnowMB]. In all these settings, there would be no fundamental modification to our algorithm. We could put a few lines using the additional page to discuss such matters informally.

---

> > ### Comment · Reviewer_bfG5 · 2024-08-11
> > **Weakness 4**
> >
> > Recall that this point reads
> >
> > > **From the perspective of learning theory**, some of the motivation provided for $0^\text{th}$ order methods seems misplaced
> >
> > The other three points are sub-points of this global one.
> >
> > I am confused by the authors' arguments once more. I felt like they were replying to other text, which only resembles mine.
> >
> > ### Weakness 4.1
> >
> > Alongside 4.3 (the conclusion of point 4), I mean that "**From the perspective of learning theory**, it seems worth mentioning that **there exists an optimal boosting algorithm that does not make use of gradients**". Again, 4.3 starts with
> >
> > > 4.3 I recognise that the authors explicitly dismiss matters of generalisation, but I still believe the remarks above are relevant in assessing the motivation for the work.
> >
> > I maintain that opinion and, honestly, I am surprised that the authors disagree. I am saying that **learning theorists**, which are (somehow) the target audience of the authors, could value being reminded of that fact and to see some discussion about it. Do the authors really disagree?
> >
> > I made it clear in my text that this "optimal $0^\text{th}$ algorithm" does not compete with the contribution of the authors (see 4.3 again), I am saying that "the authors may consider bringing this point in some form".
> >
> > ### Weakness 4.2
> >
> > > We wholeheartedly disagree
> >
> > Again, I suspect the authors do not. That is because I do not see how
> >
> > > [...] this statement overgeneralises from [2]’s Section 7.3 [...]
> >
> > It is meant to be just one "very specific context" to illustrate that going from loss minimisation to generalisation performance is not given (aand the authors' rebuttal makes that case, here). Again, my point is within a "**From the perspective of learning theory**".
> >
> >
> > Overall, I think it is worth mentioning that foreseeing important ways in which the target audience may be confused is part of an excellent presentation.

---

> ### Author Response · Authors · 2024-08-03
> **[part 5/8]  Point 5 before references**
>
> > 5. I am confused by the authors' concept of what constitutes "traditional" boosting. To me, AdaBoost is the most prototypical and traditional boosting algorithm.
>
> [RTB] We would be happy to agree with the reviewer as this would simplify a lot our arguments, but this is unfortunately factually untrue, as e.g. recently debated in PNAS [nnTP]. AdaBoost [fsAD] was the (learning theory’s) first boosting algorithm in the sense of Kearns/Valiant’s weak/strong learning model. After AdaBoost, a sizeable current in *statistics* started its own boosting “phylum”: *gradient boosting*, with the works of Jerome Friedman (et al.), remarking that Adaboost could be framed as a *gradient* optimiser and then “generalising" gradient AdaBoosting to any *differentiable loss*.
>
> To grasp the importance of this statistical current, consider that Friedman’s founding paper [fGF] is now cited more than twice as much each year compared to Freund and Schapire’s founding paper [fsAD] (source: Google Scholar).
>
> Now, two key differences between the statistical current and learning theory’s AdaBoost are (i) statistics broadened the scope of boosting to any *differentiable* loss,  **but** (ii) very few of such works have convergence proofs (even less so convergence *rates*, even less so in the weak-strong model). This is where our work finds its place and justification: we considerably broaden the applicable losses of the statistical approach to any loss while proposing explicit convergence rates in AdaBoost’s weak/strong learning setting in all cases.
>
> This also explains why we ultimately picked learning theory instead of optimisation as primary area: this field grounds the rich history of boosting and roots boosting’s “phylogenetic tree” [nnTP]; progress in optimisation has been orthogonal to this history. This is not a criticism: a lot of our references (all of Table A1) are on 0th order optimisation, which has been hugely productive on this topic while learning theory’s boosting has been mostly “deaf” to these advances. This is not surprising: the weak-strong learning setting does not explicitly call to the use of the loss’ derivatives to learn (unlike, of course, gradient descent). Given Friedman’s (et al.) take on boosting that “reduces it” to gradient descent, it was expectable and justified to try to alleviate the gradient dependence, which is our contribution. It came as a pleasant surprise that we had to make no functional assumptions to get there (unlike the state of the art 0th order optimisation).

---

> > ### Comment · Reviewer_bfG5 · 2024-08-11
> > **Weakness 5**
> >
> > Again, I have a hard time following the argument here. The authors start by saying that they strongly disagree with me and then follow it with a discussion that, to me, seems to largely validate my points.
> >
> > > [...] this is unfortunately factually untrue [...]
> >
> > What does "this" mean here? Which of my sentences are factually untrue: "**I** am confused [...]" or "To **me**, [...]"? Since I doubt I was "recently debated in PNAS", I assume the authors are referring to "AdaBoost is the most prototypical and traditional boosting algorithm" (Sorry for the joke. I was trying to lighten the mood. Still, the point is minor but valid: Try to avoid putting **any** unnecessary cognitive load on the reader, however small.)
> >
> > The "prototypical" part is not so significant and not discussed, so I will only focus on the "traditional". It seemed to me that the authors were attacking a point that I never made. I know that AdaBoost is not the most "popular" boosting algorithm and I did not suggest it had better numbers on Google Scholar than any other method. I mean what **the authors said** in more detail,
> >
> > > AdaBoost [fsAD] was the (learning theory’s) first boosting algorithm in the sense of Kearns/Valiant’s weak/strong learning model,
> >
> > or something generic like what you can find on the Wikipage for "Boosting (machine learning)".

---

> ### Author Response · Authors · 2024-08-03
> **[part 6/8]  Point 6 before references**
>
> > 6. The discussion around Assumption 5.4 is too loose […]
>
> [RA5.4] We understand the reviewer would like a discussion about the significance of the assumption. The discussion that grounds the assumption, in L165-L171, is meant to explain why such an assumption is in fact necessary in our setting. Our work is the first on boosting exploiting 0th order information about the loss and we have not seen before high-order v-derivatives with different offsets being used so we assume it is the first time $\rho$ indeed appears. To strengthen the discussion, we shall make a parallel with stochastic gradient descent (SGD) on strongly convex losses. We are happy to push some of what follows in the camera-ready, should our paper be accepted.
>
> When investigated on general strongly convex differentiable losses, the rate of SGD depends on some real that quantifies the “niceness” of the loss. A prominent such real is the *condition number* $\kappa$, the ratio between the largest and smallest eigenvalues of the Hessian (*a second order loss parameter*) of the loss [bsSO]. The convergence rate of SGD can be summarised as $\mathrm{Loss}(H_T) - \mathrm{Opt} \leq O (\kappa / T)$. This makes sense: the smaller $\kappa$, the more the loss resembles a 1D strongly convex curve rotated around a revolution axis and so any gradient step has to point to a large extent towards the global optimum (Slide 29 in [bsSO]). Hence, a rough “nice picture” for a loss to grant fast SGD convergence is that of a paraboloid of revolution.
>
> Consider our case: what is the nice picture when the loss can be arbitrary for boosting ? This picture becomes more complicated and there are reasons to believe that an additional "degree of complexity" is necessary:
> - Our weights depend on a quantity that generalises the first order derivative (Step 2.6) and pretty much like in ordinary boosting, large weights in absolute value point to examples for which substantial loss variation is possible via the weak learner. We write “loss variation” and not “loss decrease” because *labels can be flipped* (compared with AdaBoost for example, Step 2.1) so having a good weak learner is not sufficient anymore for good convergence. It makes sense thus that convergence would include an aggregator of “how nice are the weights”. Perhaps surprisingly, a relevant aggregator is trivial: it is a quadratic function of the average weight, the numerator of $\rho$. In the loss space, it is just the quadratic expectation of secants’ slopes (19) ! If this expectation is large in absolute value, there is leeway for better models. This picture is particularly clear in the convex case with the exponential loss for example, as in this case it just means that we are far from the minima of the loss.
>
> - But unfortunately in the general case of an arbitrary loss, it just gives a partial view of what is sufficient for good convergence ! Indeed, we could be optimising a loss function that jiggles a lot locally (consider Griewank’s function as an example). In this case, all slopes could be located around different basins, with different local minima nearby, and the information of a large $|\overline{W}_{1,t}|$ would then be not sufficient to ensure a better overall loss afterwards. Read: while in SGD first-order information is intuitively not enough for the best characterisation of convergence, just *ensuring* convergence in our case requires higher order v-derivative information (than just the secants’).
>
> We were pleased to realise that order-two v-derivative information is in fact sufficient, and this fits in the denominator of $\rho$. From a parallel with the Hessian in SGD, one can see that a small denominator yields smaller local “curvature” (i.e. less potential for local "jiggling") and with a large enough numerator, sufficient information is then collected in a *single real* to grant good, guaranteed boosting rate. We are sure the reviewer grasps at this point the subtleties of our approach that surely do not follow from any state of the art boosting analysis [RNT].

---

> ### Author Response · Authors · 2024-08-03
> **[part 7/8] After references, minor issues and suggestions -- $\gamma$ weak learner**
>
> > 1. The authors, unfortunately, do not define it in Section 3 [...]
>
> [RWL1] Doing so would imply defining the notion of edge, normalized edge; it seems however that it would indeed find a legitimate place there after our proposal [RDML].
>
> > 2. The definition is not self-contained [...]
>
> [RWL2] after [RDML] [RWL1], it would be straightforward to make it so, directly in Section 3
>
> > 3. Honestly, I do not recognise that definition as "the traditional" one [...] To me, that would be a  $\gamma$-weak learner [...] whose average error is at most $1/2-\gamma$.
>
> [RWL3] This is the original definition [fsAD]. However it rapidly got even weaker and generalized: the paper [fhtAL] shows that we can in fact require that the average error be slightly *different* from 1/2 instead of slightly smaller (polarity-reversal argument after their equation (20): if the weak classifier $h$ does worse than 1/2 accuracy then $-h$ does better than 1/2 accuracy). This is however still for $f = -1,1$. The paper [ssIB] generalizes the measure to $f \in [-1,1]$: their $r_t$ (Corollary 1) is our $\tilde{\eta}_t$ and the discussion relating it to the error is right after (page 303, par. 1). From here many papers started to adopt the edge / margin notion directly in the weak learning assumption, see for example [mnwRC] for a definition that looks just like ours.
>
> > Assumption 5.5 might be less "global" than one could expect [...]
>
> [RWL4] Misunderstanding: it would not be a good idea to present the assumption the way proposed by the reviewer because then we risk losing sight of the fact that $h$ is normalized by *its* maximal empirical value. Our definition in fact coincides with [ssIB] (Corollary 1).
>
> The proposition made later in the bullet points to instead divide by a global term (we assume it is a term computed over all weak hypotheses) could in fact break the purpose of the weak learning framework: suppose that one $h$ has a huge $M^* = 10000$ while the others have $M=1$. Then satisfying the weak learning assumption by all those hypotheses imposes, instead of having our $|\tilde{\eta}| \geq \gamma$, to have $|\tilde{\eta}| \geq \gamma \cdot M^*/M = 10 000 \gamma$, imposing the weak classifiers to be in fact strong in disguise.
>
> At this point, we hope we have clarified the misunderstandings and our simple rewriting proposal would make it even easier to grasp the idea of the $\gamma$-weak learner.
>
> > Bullet points
>
> $|\textbf{w}|$ is indeed the componentwise absolute value (we will put it in Section 3)
>
> notation X is a bug -- it should have been noted $\textbf{wl}$. We shall fix it.
>
> $\mathcal{S}_t$ is the set of examples that our algorithm uses. We have defined it in the algorithm itself.
>
> See [RWL4] for the last bullet point.

---

> > ### Comment · Reviewer_bfG5 · 2024-08-11
> >
> > > This is the original definition [fsAD]. However it rapidly got even weaker and generalized [...]
> >
> > Again, it seems that the authors are agreeing with me. There is a traditional definition, then others modify it (obtaining non-traditional ones).

---

> > ### Comment · Reviewer_bfG5 · 2024-08-11
> >
> > > $\mathcal{S}_t$ is the set of examples that our algorithm uses. We have defined it in the algorithm itself.
> >
> > I am not sure to what the authors are replying. To my quick "(Why there?)"? If so, I remain unconvinced that it is the best choice to only define a variable in a comment.

---

> ### Author Response · Authors · 2024-08-03
> **[part 8/8] general minor issues and suggestions**
>
> ## General minor issues and suggestions
>
> We proceed through bullet order
>
> > In the technical summary above [...]
>
> The suggestion seems to suggest a continuous version of boosting in the vein of [awWW] for continuous EG. It is a very interesting question !
>
> > (2 following bullets) [...] it is not even fully clear how novel the concepts introduced are.
>
> Section 4 introduces concepts related to v-derivatives, with 4 definitions, 4.1, 4.2, 4.3, 4.4, 4.6. It is absolutely explicit in the text that 4.1 comes from another work, 4.2 is new, 4.3 is new, 4.4 comes from another work and 4.6 is new. For all new concepts, we link them to their closest relative published.
>
> > I suspect you require less from the hypotheses returned by the weak learner [...]
>
> The reviewer is right but somehow this would risk "hiding" the fact that on training, an hypothesis needs to have finite values (or we just cannot compute the objective). Our formulation perhaps look less general but was formulated as is on purpose.
>
> > Consider stating more explicitly [...]
>
> We will do.
>
> > [Eq. (4)] Consider using [...]
>
> Excellent suggestion.
>
> > Number only the equations that are referenced in the text
>
> We will do.
>
> > Consider a version of Figure 1
>
> Just to confirm, we see it as a Figure where dim1 would e.g. be the difference between $z$ and $z'$ (say $z$ is fixed) and dim2 would be the offset ? That would be easy and a good idea.
>
> > Avoid starting sentences with mathematical symbols
>
> Agreed, though L93 does not start like this (typo ?)
>
> > [122] the reference to [47, Appendix, Section 4] has a bit too much packed
>
> Agreed. Note that from [RNT2], part of the "unpacking" would directly start from the introduction
>
> > Adding hyperlinks to the steps of the algorithm
>
> Easy. We will do.
>
> > Algorithm 1 can be made significantly tidier.
>
> We believe we can simplify Step 2.5 and replace the table in Step 2.3 by a more conventional algorithmic convention

---

> > ### Comment · Reviewer_bfG5 · 2024-08-11
> >
> > > It is absolutely explicit in the text that [...]
> >
> > It is not. The authors are relying on the convention that "providing no citations means it is original", right? That leaves originality implicit (definitely not "absolutely explicit"). Also, many authors do not follow this convention strictly (I find that a pitty). So, what I wrote holds, specially since it is a weaker statement than what the authors seem to notice: "not even fully clear" includes, for example, "just clear". Still, I was truly more confused than usual there, but unfortunately, I am not quite sure why.

---

> > ### Comment · Reviewer_bfG5 · 2024-08-11
> >
> > > Agreed, though L93 does not start like this (typo ?)
> >
> > This is a good example of the communication issue here. It is not a typo (I checked). What I wrote means what it says.

---

> ### Author Response · Authors · 2024-08-03
> **Additional references used therein**
>
> [awWW] E. Ahmid and M. Warmuth. Winnowing with gradient descent. COLT 2020
>
> [bsSO] F. Bach and S. Sra. Stochastic optimization: Beyond stochastic gradients and convexity. Tutorial at NeurIPS 2016
>
> [bmdgCW] A. Banerjee, S. Merugu, I. Dhillon and J. Ghosh. Clustering with Bergman divergences. JMLR 2005
>
> [btAI] P. L. Bartlett and M. Traskin. AdaBoost is consistent. JMLR 2007
>
> [cmnowMB] Cranko, Menon, Nock, Ong and Walder. Monge blunts Bayes: Hardness Results for Adversarial Training. ICML 2019.
>
> [fGF] J. Friedman. Greedy function approximation: a *gradient* boosting machine. Annals of Statistics 2001 (emphasis ours)
>
> [fhtAL] J. Friedman, T. Hastie and R. Tibshirani. Additive logistic regression: a statistical view of boosting. AoS 2000
>
> [fsAD] Y. Freund and R. Schapire. A decision-theoretic generalization of on-line learning and an application to boosting. JCSS 1997 (early version in EuroCOLT 1995).
>
> [fsGT] Y. Freund and R. Schapire. Game theory, on-line prediction and boosting. COLT 1996
>
> [ksNO] M. Kearns and S. Singh. Near-optimal reinforcement learning in polynomial time. ICML 1998.
>
> [lrOW] K. G. Larsen and M. Ritzert. Optimal weak to strong learning. NeurIPS 2022
>
> [mnwRC] Y. Mansour, R. Nock and R.C. Williamson. Random classification noise does not defeat all convex potential boosters irrespective of model choice. ICML 2023
>
> [nnTP] R. Nock and F. Nielsen. The Phylogenetic Tree of Boosting has a Bushy Carriage but a Single Trunk. PNAS 2020.
>
> [nwLO] R. Nock and R.C. Williamson. Lossless or quantised boosting with integer arithmetic. ICML 2019..
>
> [sfblBT] R. Schapire, Y. Freund, P. Bartlett and W.S. Lee. Boosting the margin: a new explanation for the effectiveness of voting methods. ICML 1997.
>
> [ssIB] R. Schapire and Y. Singer. Improved boosting algorithms using confidence-rated predictions. MLJ 1999.
>
> [tAP] M. Telgarsky. A primal-dual convergence analysis of boosting. JMLR 2012
>
> [tBW] M. Telgarsky. Boosting with the logistic loss is consistent. COLT 2013
>
> [wvTS] M.K. Warmuth and S.V.N. Vishwanathan, Survey of boosting from an optimization perspective. Tutorial at ICML 2009

---

### Author Response · Authors · 2024-08-09
**Sincere apologies: some of our "Official Comments" were not visible to reviewers (should be fixed now) !**

After comments from reviewer 8gCx and TGRx saying that some tags could not be found in our list of comments to reviewer bfG5, we checked the visibility of our "Official Comments" (8 + references) to reviewer bfG5 and **it appears that the checkbox for reviewers was unchecked -- it thus seems that no reviewer could access them !!**

We have hopefully fixed the problem for our official comments to bfG5 and will have a quick look at other "Official Comments" made to other reviewers, to make sure they properly appear in your browser.

We can only sincerely apologize for the blunder we caused and the time wasted in trying to find those hidden tags (special apologies to reviewer TGRx whose comments should have led us to the issue hours ago) !!

Just to confirm now, if for example you look for tag [RWL3], you will find that it points to an "Official Comment" to reviewer bfG5 titled: "**[part 7/8] After references, minor issues and suggestions -- weak learner**"

---

### Comment · Area_Chair_kTAK · 2024-08-12
**regarding the discussion between reviewer bfG5 and the authors**

[I am sending this as a comment to all reviewers so it doesn't get lost in the particular thread]

Dear reviewer bfG5,

Thank you for the obvious and honest effort you are putting into your review, well beyond the call of duty. But at this point I must intervene, as the amount of work required of the authors to reply, and for the other reviewers and myself to follow the conversation, is excessive. Could you write a very succint summary of what your points are for and against the paper, that we can refer to as your final review? And keeping to objective, technical matters as much as possible. The positioning of a paper is quite subjective, and the authors are within their right to frame their paper as they wish. That succint summary will simplify our job. The fundamental goals of the process are to make a decision about the paper and to give helpful suggestions to the authors to improve the paper.

Also, purely as a reminder, the NeurIPS policy is that "for accepted papers, and for rejected papers that opt-in, reviews and notes you send to the reviewers will be made public after notification". So both authors and reviewers can leave diverging comments about the paper for the world to see.

Thanks,

Your AC

---

> ### Comment · Reviewer_bfG5 · 2024-08-12
>
> Of course. I will also update my scores as that is sure to remove much burden from the authors.
>
> In short, my position on this work is
> > Looks like a nice contribution, but the presentation did not allow me to fully understand how nice it is. I will assume it is an OK contribution, but I am confident it was poorly presented (not just due to the mismatch of expertise). That amounts to a score of 5 (borderline accept). Moreover, I know that I could not detect if this contribution was some sort of sham. Among other reasons, this motivates me to lower my confidence score to 1 as "The submission is not in (my) area or the submission was difficult to understand".
>
> On noteworthy technical matters
> > We did not quite finish much. For example, I attempted to reframe the problem to better contextualize it, further shaving any learning theory taste in it. However, I felt that the authors' reply was not very helpful, so I give up on that.
>
> ---
> As a last quick note about the AC's reminder that this discussion will become public (it could help the authors understand my position)
> > This is what is giving me energy. To me, I am fighting FOR the authors (there might even be (talented!) students among them). I do believe they are skilled researchers and, thus, it would not be worth for them to risk tarnishing their reputation *even if the risk was small*. Again, I do not think they are malicious, as I mentioned, but my personal opinion is that others could suspect.
>
> Thank you.

---

### Decision · Program_Chairs · 2024-09-25

**Decision:**

Accept (poster)

**Comment:**

After a long discussion, all reviewers suggested acceptance, although overall they found the paper hard to understand and their confidence was very low. Personally, I like to see this new approach to boosting, an area that has seen so much work in past decades. Based on all this, I suggest acceptance. I advice the authors to take the reviewers' comments into consideration in order to make the paper more clear.